# Local Convergence Analysis of Gradient Descent Ascent with Finite Timescale Separation

**Tanner Fiez & Lillian J. Ratliff**
Department of Electrical and Computer Engineering
University Washington
Seattle, WA 98195
{fiezt, ratliffl}@uw.edu

## Abstract

We study the role that a finite timescale separation parameter $\tau$ has on gradient descent-ascent in non-convex, non-concave zero-sum games where the learning rate of player 1 is denoted by $\gamma_1$ and the learning rate of player 2 is defined to be $\gamma_2 = \tau\gamma_1$. We provide a non-asymptotic construction of the finite timescale separation parameter $\tau^*$ such that gradient descent-ascent locally converges to $x^*$ for all $\tau \in (\tau^*, \infty)$ if and only if it is a strict local minmax equilibrium. Moreover, we provide explicit local convergence rates given the finite timescale separation. The convergence results we present are complemented by a non-convergence result: given a critical point $x^*$ that is not a strict local minmax equilibrium, we present a non-asymptotic construction of a finite timescale separation $\tau_0$ such that gradient descent-ascent with timescale separation $\tau \in (\tau_0, \infty)$ does not converge to $x^*$. Finally, we extend the results to gradient penalty regularization methods for generative adversarial networks and empirically demonstrate on CIFAR-10 and CelebA the significant impact timescale separation has on training performance.

## 1 Introduction

In this paper we study learning in zero-sum games of the form

$$\min_{x_1 \in X_1} \max_{x_2 \in X_2} f(x_1, x_2)$$

where the objective function of the game $f$ is assumed to be sufficiently smooth and potentially non-convex and non-concave in the strategy spaces $X_1$ and $X_2$ respectively with each $X_i$ a precompact subset of $\mathbb{R}^{n_i}$. This general problem formulation has long been fundamental in game theory (Başar & Olsder, 1998) and recently it has become central to machine learning with applications in generative adversarial networks (Goodfellow et al., 2014), robust supervised learning (Madry et al., 2018; Sinha et al., 2018), reinforcement and multi-agent reinforcement learning (Rajeswaran et al., 2020; Zhang et al., 2019), imitation learning (Ho & Ermon, 2016), constrained optimization (Cherukuri et al., 2017), and hyperparameter optimization (Lorraine et al., 2020; MacKay et al., 2019).

The gradient descent-ascent learning dynamics are widely studied as a potential method for efficiently computing equilibria in game formulations. However, in zero-sum games, a number of past works highlight problems with this learning dynamic including both non-convergence to meaningful critical points as well as convergence to critical points devoid of game theoretic meaning, where common notions of 'meaningful' equilibria include the local Nash and local minmax (Stackelberg) concepts. For instance, in bilinear games, gradient descent-ascent avoids local Nash and Stackelberg equilibria due to the inherent instability of the update rule for this class. Fortunately, in this class of games, regularization or gradient-based learning dynamics that employ different numerical discretization schemes (as compared to forward Euler for gradient descent-ascent) are known to alleviate this issue (Daskalakis et al., 2018; Mertikopoulos et al., 2019; Zhang & Yu, 2020). For the more general nonlinear nonconvex-nonconcave class of games, it has been shown gradient descent-ascent with a shared learning rate is prone to reaching critical points that are neither local Nash equilibria nor local Stackelberg equilibria (Daskalakis & Panageas, 2018; Jin et al., 2020; Mazumdar et al., 2020). While an important negative result, it does not rule out the prospect that gradient

descent-ascent may be able to guarantee equilibrium convergence as it fails to account for a key structural parameter of the dynamics, namely the ratio of learning rates between the players.

Motivated by the observation that the order of play between players is fundamental to the definition of the game, the role of *timescale separation* in gradient descent-ascent has recently been explored theoretically (Chasnov et al., 2019; Heusel et al., 2017; Jin et al., 2020). On the empirical side, it has been widely demonstrated that timescale separation in gradient descent-ascent is crucial to improving the solution quality when training generative adversarial networks (Arjovsky et al., 2017; Goodfellow et al., 2014; Heusel et al., 2017). Denoting $\gamma_1$ as the learning rate of the player 1, the learning rate of player 2 can be redefined as $\gamma_2 = \tau\gamma_1$ where $\tau = \gamma_2/\gamma_1 > 0$ is the learning rate ratio. Toward understanding the effect of timescale separation, Jin et al. (2020) show the locally stable critical points of gradient descent-ascent coincide with the set of strict local minmax/Stackelberg equilibrium across the spectrum of sufficiently smooth zero-sum games as $\tau \to \infty$. In other words, all 'bad critical points' (critical points lacking game-theoretic meaning) become unstable and all 'good critical points' (game-theoretically meaningful equilibria) remain or become locally exponentially stable (cf. Definition 3) as $\tau \to \infty$. While a promising theoretical development, gradient descent-ascent with a timescale separation approaching infinity does not lead to a practical learning rule and the analysis of it does not necessarily provide insights into the common usage of a reasonable finite timescale separation. An important observation is that choosing $\tau$ arbitrarily large with the goal of ensuring local equilibrium convergence can lead to numerically ill-conditioned problems. This highlights the significance of understanding the exact range of learning rate ratios that guarantee local stability. Moreover, our experiments in Section 5 (Dirac-GAN) and in Appendix K show that modest values of $\tau$ are typically sufficient to guarantee stability of only equilibria which allows for larger choices of $\gamma_1$ and results in faster convergence to an equilibrium.

**Contributions.** We show gradient descent-ascent locally converges to a critical point for a range of finite learning rate ratios if and only if the critical point is a strict local Stackelberg equilibria (Theorem 1).[1] This result is constructive in the sense that we explicitly characterize the exact range of learning rate ratios for which the guarantee holds. Furthermore, we show all other critical points are unstable for a range of finite learning rate ratios that we explicitly construct (Theorem 2). To our knowledge, the aforementioned guarantees are the first of their kind in nonconvex-nonconcave zero-sum games for an implementable first-order method. Moreover, the technical results in this work rely on tools that have not appeared in the machine learning and optimization communities analyzing games. Finally, we extend these results to gradient penalty regularization methods in generative adversarial networks (Theorem 3), thereby providing theoretical guarantees for a common combination of heuristics used in practice, and empirically demonstrate the benefits and trade-offs of regularization and timescale separation on the Dirac-GAN along with image datasets.

## 2 PRELIMINARIES

A two–player zero-sum continuous game is defined by a collection of costs $(f_1, f_2)$ where $f_1 \equiv f$ and $f_2 \equiv -f$ with $f \in C^r(X, \mathbb{R})$ for some $r \geq 2$ and where $X = X_1 \times X_2$ with each $X_i$ a precompact subset of $\mathbb{R}^{n_i}$ for $i \in \{1, 2\}$ and $n = n_1 + n_2$. Each player $i \in \mathcal{I}$ seeks to minimize their cost $f_i(x_i, x_{-i})$ with respect to their choice variable $x_i$ where $x_{-i}$ is the vector of all other actions $x_j$ with $j \neq i$. We denote $D_i f_i$ as the derivative of $f_i$ with respect to $x_i$, $D_{ij} f_i$ as the partial derivative of $D_i f_i$ with respect to $x_j$, and $D_i^2 f_i$ as the partial derivative of $D_i f_i$ with respect to $x_i$.

**Mathematical Notation.** Given a matrix $A \in \mathbb{R}^{n_1 \times n_2}$, let $\text{vec}(A) \in \mathbb{R}^{n_1 n_2}$ be its vectorization such that $\text{vec}(A)$ takes rows $a_i$ of $A$, transposes them and stacks them vertically in order of their index. Let $\otimes$ and $\oplus$ denote the Kronecker product and sum respectively, where $A \oplus B = A \otimes I + I \otimes B$. Moreover, $\boxplus$ is an operator that generates an $\frac{1}{2}n(n+1) \times \frac{1}{2}n(n+1)$ matrix from a matrix $A \in \mathbb{R}^{n \times n}$ such that $A \boxplus A = H_n^+(A \oplus A)H_n$ where $H_n^+ = (H_n^\top H_n)^{-1}H_n^\top$ is the (left) pseudo-inverse of $H_n$, a full column rank duplication matrix. Let $\lambda_{\max}^+(\cdot)$ be the largest positive real root of its argument if it exists and zero otherwise. See Lancaster & Tismenetsky (1985) and Appendix B for more detail.

**Equilibrium.** There are natural equilibrium concepts depending on the order of play: the (local) Nash equilibrium concept in the case of simultaneous play and the (local) Stackelberg (equivalently minmax in zero-sum games) equilibrium concept in the case of hierarchical play (Başar & Olsder,

---

[1]Following Fiez et al. (2020), we refer to strict local Stackelberg as differential Stackelberg throughout.

1998). Formal local equilibrium definitions are provided in Appendix B, while here we characterize the different equilibrium notions in terms of sufficient conditions on player costs as is typical in the machine learning and optimization literature (see, e.g., Berard et al. 2020; Daskalakis & Panageas 2018; Fiez et al. 2020; Goodfellow 2016; Jin et al. 2020; Mazumdar et al. 2020; Wang et al. 2020).

The following definition is characterized by sufficient conditions for a local Nash equilibrium.

**Definition 1** (Differential Nash Equilibrium, Ratliff et al. 2013). *The joint strategy $x \in X$ is a differential Nash equilibrium if $D_1 f(x) = 0$, $-D_2 f(x) = 0$, $D_1^2 f(x) > 0$, and $D_2^2 f(x) < 0$.*

The Jacobian of the vector of individual gradients $g(x) = (D_1 f(x), -D_2 f(x))$ is defined by

$$J(x) = \begin{bmatrix} D_1^2 f(x) & D_{12} f(x) \\ -D_{12}^\top f(x) & -D_2^2 f(x) \end{bmatrix}. \tag{1}$$

Let $\mathtt{S}_1(\cdot)$ denote the Schur complement of $(\cdot)$ with respect to the $n_2 \times n_2$ block in $(\cdot)$. The following definition is characterized by sufficient conditions for a local Stackelberg equilibrium.

**Definition 2** (Differential Stackelberg Equilibrium, Fiez et al. 2020). *The joint strategy $x \in X$ is a differential Stackelberg equilibrium if $D_1 f(x) = 0$, $-D_2 f(x) = 0$, $\mathtt{S}_1(J(x)) > 0$, $D_2^2 f(x) < 0$.*

**Learning Dynamics**. We study agents seeking equilibria of the game via a learning algorithm and consider arguably the most natural learning rule in zero-sum continuous games: gradient descent-ascent (GDA). Moreover, we investigate this learning rule with *timescale separation* between the players. Let $\tau = \gamma_2/\gamma_1$ be the *learning rate ratio* and define $\Lambda_\tau = \mathrm{blockdiag}(I_{n_1}, \tau I_{n_2})$ where $I_{n_i}$ is a $n_i \times n_i$ identity matrix. The $\tau$-GDA dynamics with $g(x) = (D_1 f(x), -D_2 f(x))$ are given by

$$x_{k+1} = x_k - \gamma_1 \Lambda_\tau g(x_k). \tag{2}$$

## 3 STABILITY OF CONTINUOUS TIME GDA WITH TIMESCALE SEPARATION

To characterize the convergence of $\tau$-GDA, we begin by studying its continuous time limiting system

$$\dot{x} = -\Lambda_\tau g(x). \tag{3}$$

The Jacobian of the system from (3) is given by $J_\tau(x) = \Lambda_\tau J(x)$ where $J(x)$ is defined in (1). Observe that critical points ($x^*$ such that $g(x^*) = 0$) are shared between $\tau$-GDA and (3). Thus, by analyzing the stability of the continuous time system around critical points as a function of the timescale separation $\tau$ using the Jacobian $J_\tau(x^*)$, we can draw conclusions about the stability and convergence of the discrete time system $\tau$-GDA. It well known that a critical point is locally (exponentially) stable when the spectrum of $-J_\tau(x^*)$ is in the open left-half complex plane $\mathbb{C}_-^\circ$ (cf. Theorem B.1, Appendix B). Throughout, we use the broader term "stable" to mean the following.

**Definition 3.** *A critical point $x^*$ is locally exponentially stable for $\dot{x} = -\Lambda_\tau g(x)$ if an only if $\mathrm{spec}(-J_\tau(x^*)) \subset \mathbb{C}_-^\circ$ (or, equivalently, $\mathrm{spec}(J_\tau(x^*)) \subset \mathbb{C}_+^\circ$) where $\mathbb{C}_-^\circ$ and $\mathbb{C}_+^\circ$ denote the open left-half and right-half complex plane, respectively.*

We now show differential Stackelberg equilibria are the only critical points that are stable for a range of finite learning rate ratios,[2] whereas the remainder of critical points are unstable for a range of finite learning rate ratios. Importantly, we characterize the learning rate ratios for which the results hold.

### 3.1 NECESSARY AND SUFFICIENT CONDITIONS FOR STABILITY

To motivate our main stability result, the following example shows the existence of a differential Stackelberg which is unstable for $\tau = 1$, but is stable all for $\tau \in (\tau^*, \infty)$ where $\tau^*$ is finite.

**Example 1.** *Consider the quadratic zero-sum game defined by the cost*

$$f(x_1, x_2) = \tfrac{v}{2}(-x_{11}^2 + \tfrac{1}{2}x_{12}^2 - 2x_{11}x_{21} - \tfrac{1}{2}x_{21}^2 + x_{12}x_{22} - x_{22}^2)$$

*where $v > 0$ and $x_1, x_2 \in \mathbb{R}^2$. The unique critical point $x^* = (0, 0)$ is a differential Stackelberg equilibrium since $g(x^*) = 0$, $\mathtt{S}_1(J(x^*)) = \mathrm{diag}(v, \tfrac{v}{4}) > 0$, and $D_2^2 f(x^*) = -\mathrm{diag}(\tfrac{v}{2}, v) < 0$.*

---

[2]Note that differential Nash are a subset of differential Stackelberg (Fiez et al., 2020; Jin et al., 2020).

*Moreover,* $\mathrm{spec}(-J_\tau(x^*)) = \{\frac{-v}{4}(2\tau+1\pm\sqrt{4\tau^2 - 8\tau + 1}), \frac{-v}{4}(\tau-2\pm\sqrt{\tau^2 - 12\tau + 4})\}$. *Observe that for any* $v > 0$, $x^*$ *is unstable for* $\tau = 1$ *since* $\mathrm{spec}(-J_\tau(x^*)) \not\subset \mathbb{C}^\circ_-$, *but* $x^*$ *is stable for a range of learning rates since* $\mathrm{spec}(-J_\tau(x^*)) \subset \mathbb{C}^\circ_-$ *for all* $\tau \in (2,\infty)$.

In other words, GDA fails to converge to the equilibrium but a finite timescale separation is sufficient to remedy this problem. We now fully characterize this phenomenon. To provide some background, we remark it is known that the spectrum of $-J_\tau(x^*)$ asymptotically splits as $\tau \to \infty$ such that $n_1$ eigenvalues tend to fixed positions defined by the eigenvalues of $-\mathtt{S}_1(J(x^*))$, while the remaining $n_2$ eigenvalues tend to infinity at a linear rate $\tau$ along asymptotes defined by the eigenvalues of $D_2^2 f(x^*)$. This result is known from Klimushchev & Krasovskii (1961) and further discussion can be found in Appendix I as well as from Kokotovic et al. (1986, Chap. 2, Thm. 3.1). The previous fact is specialized from the class of singularly perturbed linear systems to $\tau$-GDA by Jin et al. (2020) which directly results in the connection between critical points of $\infty$–GDA and differential Stackelberg equilibrium. Specifically, the result of Jin et al. (2020) is showing that for the class of all sufficiently smooth games the stable critical points of $\infty$-GDA are exactly the strict local minmax. As a corollary of this fact, there exists a $\tau_1 < \infty$, such that $\tau$-GDA is stable for all $\tau > \tau_1$ (Kokotovic et al., 1986, Chap. 2, Cor. 3.1); this can be inferred from the proof of Theorem 28 in Jin et al. (2020) as well. Indeed, Jin et al. (2020) gives an asymptotic expansion showing that $n_1$ eigenvalues of $-J_\tau(x^*)$ are in $\mathrm{spec}(-\mathtt{S}_1(J(x^*))) + O(\tau^{-1})$ and the remaining $n_2$ eigenvalues are in $\tau(\mathrm{spec}(D_2^2 f(x^*)) + O(\tau^{-1}))$. Using the limit definition for the asymptotic expansion, for any fixed game and a strict local minmax $x^*$, one can show that there exists a finite $\tau$ such that $x^*$ is stable. We provide a detailed discussion of the relationship between the results of Jin et al. (2020) and Kokotovic et al. (1986) in Appendices A, I, and J. Unfortunately, the finite $\tau_1$ obtainable from the asymptotic expansion method can be arbitrarily large. From a practical perspective, this poses significant problems for the implementation and performance of $\tau$-GDA. Indeed, the *eigenvalue gap* between $\mathrm{spec}(-\mathtt{S}_1(J(x^*)))$ and $\mathrm{spec}(\tau D_2^2 f(x^*))$ has a linear dependence on $\tau$ and, in turn, the problem may become highly ill-conditioned from a numerical perspective as $\tau$ becomes large (Kokotovic, 1975). In contrast, we determine exactly the range of $\tau$ such that the spectrum of $-J_\tau(x)$ remains in $\mathbb{C}^\circ_-$, and hence, remedy this problem.

For the statement of the following theorem on the non-asymptotic construction of $\tau^*$, we define the following matrices: for a critical point $x^*$, let $S_1 = \mathtt{S}_1(-J_\tau(x^*)) = A_{11} - A_{12}A_{22}^{-1}A_{12}^\top$ and

$$-J_\tau(x^*) = \begin{bmatrix} -D_1^2 f(x^*) & -D_{12}f(x^*) \\ \tau D_{12}^\top f(x^*) & \tau D_2^2 f(x^*) \end{bmatrix} = \begin{bmatrix} A_{11} & A_{12} \\ -\tau A_{12}^\top & \tau A_{22} \end{bmatrix}.$$

**Theorem 1** (Non-Asymptotic Construction of Necessary and Sufficient Conditions for Stability). *Consider a zero-sum game* $(f_1, f_2) = (f, -f)$ *defined by* $f \in C^r(X, \mathbb{R})$ *for some* $r \geq 2$. *Suppose that* $x^*$ *is such that* $g(x^*) = 0$ *and* $\det(D_2^2 f_2(x^*)) \neq 0$. *There exists a* $\tau^* \in [0, \infty)$ *such that* $\mathrm{spec}(-J_\tau(x^*)) \subset \mathbb{C}^\circ_-$ *for all* $\tau \in (\tau^*, \infty)$ *if and only if* $x^*$ *is a differential Stackelberg equilibrium. Moreover,* $\tau^* = \lambda^+_{\max}(Q)$ *where*

$$Q = 2\left[(A_{12} \otimes A_{22}^{-1})H_{n_2} \quad (I_{n_1} \otimes A_{22}^{-1}A_{12}^\top)H_{n_1}\right] \begin{bmatrix} \bar{A}_{22}^{-1} H^+_{n_2}(A_{12}^\top \otimes I_{n_2}) \\ -\bar{S}_1^{-1} H^+_{n_1}(S_1 \otimes A_{12}A_{22}^{-1}) \end{bmatrix} - (A_{11} \otimes A_{22}^{-1})$$

*with* $\bar{A}_{22} = A_{22} \boxplus A_{22}$ *and* $\bar{S}_1 = S_1 \boxplus S_1$.

While at first glance $Q$ may appear difficult to understand, it is efficiently computable and can be used to understand the typical value for important classes of games. Indeed, many problems like generative adversarial networks have specific structure for the individual Hessians of each player and the interaction matrix $D_{12}f$ (cf. Assumption 1, Section 3.3) and are in a sense subject to design via network architecture and loss function selection. This result opens up an interesting future direction of research on understanding and potentially designing the structure of $Q$. To take a step in this direction, we explore a number of games in Section 5 and Appendix K where we compute $\tau^*$ by the construction and validate it is tight empirically. Along the way, we discover that $\tau^*$ is typically a reasonable value that is amenable to practical implementations.

As a direct consequence of Theorem 1, $\tau$-GDA converges locally asymptotically for any sufficiently small $\gamma(\tau)$ and for all $\tau \in (\tau^*, \infty)$ if and only if $x^*$ is a differential Stackelberg equilibrium; for a formal statement see Corollary C.1 in Appendix C.

*Proof Sketch of Theorem 1.* The full proof is contained in Appendix C. The key tools used in this proof are a combination of Lyapunov stability and the notion of a *guard map* (Saydy et al., 1990),

a new tool to the learning community. Recall that a matrix is exponentially stable if and only if there exists a symmetric positive definite $P = P^\top > 0$ such that $PJ_\tau(x^*) + J_\tau^\top(x^*)P > 0$ (cf. Theorem B.1, Appendix B). Hence, given a positive definite $Q = Q^\top > 0$, $-J_\tau(x^*)$ is stable if and only if there exists a unique solution $P = P^\top$ to

$$((J_\tau^\top(x^*) \otimes I) + (I \otimes J_\tau^\top(x^*)))\mathrm{vec}(P) = (J_\tau^\top(x^*) \oplus J_\tau^\top(x^*))\mathrm{vec}(P) = \mathrm{vec}(Q) \qquad (4)$$

where $\otimes$ and $\oplus$ denote the Kronecker product and Kronecker sum, respectively.[3] The existence of a unique solution $P$ occurs if and only if $J_\tau^\top$ and $-J_\tau^\top$ have no eigenvalues in common. Hence, using the fact that eigenvalues vary continuously, if we vary $\tau$ and examine the eigenvalues of the map $J_\tau^\top(x^*) \oplus J_\tau^\top(x^*)$, this tells us the range of $\tau$ for which $\mathrm{spec}(-J_\tau(x^*))$ remains in $\mathbb{C}_-^\circ$. This method of varying parameters and determining when the roots of a polynomial (or correspondingly, the eigenvalues of a map) cross the boundary of a domain uses a *guard map*; it provides a certificate that the roots of a polynomial lie in a particular guarded domain for a range of parameter values.

Formally, let $\mathcal{X}$ be the set of all $n \times n$ real matrices or the set of all polynomials of degree $n$ with real coefficients. Consider $\mathcal{S}$ an open subset of $\mathcal{X}$ with closure $\bar{\mathcal{S}}$ and boundary $\partial\mathcal{S}$. The map $\nu : \mathcal{X} \to \mathbb{C}$ is said to be a guardian map for $\mathcal{S}$ if for all $x \in \bar{\mathcal{S}}$, $\nu(x) = 0 \iff x \in \partial\mathcal{S}$. Elements of $\mathcal{S}(\mathbb{C}_-^\circ) = \{A \in \mathbb{R}^{n \times n} : \mathrm{spec}(A) \subset \mathbb{C}_-^\circ\}$ are (Hurwitz) stable. Given a pathwise connected set $U \subseteq \mathbb{R}$, the parameterized family $\{A(\tau) : \tau \in U\}$ is stable if and only if $(i)$ it is nominally stable—meaning $A(\tau_1) \in \mathcal{S}(\mathbb{C}_-^\circ)$ for some $\tau_1 \in U$—and $(ii)$ $\nu(A(\tau)) \neq 0$ for all $\tau \in U$ (Saydy et al., 1990, Prop. 1). The map $\nu(\tau) = \det(2(-J_\tau(x^*) \odot I)) = \det(-(J_\tau(x^*) \oplus J_\tau(x^*)))$ guards $\mathcal{S}(\mathbb{C}_-^\circ)$ where $\odot$ is the *bialternate product* and is defined by $A \odot B = \frac{1}{2}(A \oplus B)$ for matrices $A$ and $B$ (Govaerts, 2000, Sec. 4.4.4). For intuition, consider the case where each $x_1, x_2 \in \mathbb{R}$ so that

$$J_\tau(x^*) = \begin{bmatrix} a & b \\ -\tau b & \tau d \end{bmatrix} \in \mathbb{R}^{2 \times 2}.$$

It is known that $\mathrm{spec}(-J_\tau(x^*)) \subset \mathbb{C}_-^\circ$ if $\det(-J_\tau(x^*)) > 0$ and $\mathrm{tr}(-J_\tau(x^*)) < 0$ so that $\nu(\tau) = \det(-J_\tau(x^*))\,\mathrm{tr}(-J_\tau(x^*))$ is a guard map for the $2 \times 2$ stable matrices $\mathcal{S}(\mathbb{C}_-^\circ)$. Since the bialternate product generalizes the trace operator and $\det(-J_\tau(x^*)) = \tau^{n_2}\det(D_2^2 f(x^*))\det(-\mathsf{S}_1(J(x^*))) \neq 0$ for $\tau \neq 0$ by the facts $(\det(\mathsf{S}_1(J(x^*))) \neq 0$ and $\det(D_2^2 f(x^*)) \neq 0)$ for a differential Stackelberg equilibrium $x^*$, a guard map in the general $n \times n$ case is $\nu(\tau) = \det(-(J_\tau(x^*) \oplus J_\tau(x^*)))$.

This guard map in $\tau$ is closely related to the vectorization in (4): for any symmetric positive definite $Q = Q^\top > 0$, there will be a symmetric positive definite solution $P = P^\top > 0$ of $-(J_\tau^\top(x^*) \oplus J_\tau^\top(x^*))\mathrm{vec}(P) = \mathrm{vec}(-Q)$ if and only if $\det(-(J_\tau(x^*) \oplus J_\tau(x^*))) \neq 0$. Hence, to find the range of $\tau$ for which, given any $Q = Q^\top > 0$, the solution $P = P^\top$ is no longer positive definite, we need to find the value of $\tau$ such that $\nu(\tau) = \det(-(J_\tau(x^*) \oplus J_\tau(x^*))) = 0$—that is, where it hits the boundary $\partial\mathcal{S}(\mathbb{C}_-^\circ)$. Through algebraic manipulation, this problem reduces to an eigenvalue problem in $\tau$, giving rise to an explicit construction of $\tau^*$. $\qquad\square$

## 3.2 SUFFICIENT CONDITIONS FOR INSTABILITY

To motivate our main instability result, the following example shows a non-equilibrium critical point that is stable for $\tau = 1$, but is unstable for all $\tau \in (\tau_0, \infty)$ where $\tau_0$ is finite.

**Example 2.** *Consider the quadratic zero-sum game defined by the cost*

$$f(x_1, x_2) = \tfrac{v}{4}(x_{11}^2 - \tfrac{1}{2}x_{12}^2 + 2x_{11}x_{21} + \tfrac{1}{2}x_{21}^2 + 2x_{12}x_{22} - x_{22}^2)$$

*where $x_1, x_2 \in \mathbb{R}^2$ and $v > 0$. The unique critical point $x^* = (0, 0)$ is not a differential Stackelberg (nor Nash) equilibrium since $D_1^2 f(x^*) = \mathrm{diag}(v/2, -v/4) \not> 0$, $D_2^2 f(x^*) = \mathrm{diag}(v/4, -v/2) \not< 0$. Moreover, $\mathrm{spec}(-J_\tau(x^*)) = \{\tfrac{-v}{8}(2\tau - 1 \pm \sqrt{4\tau^2 - 12\tau + 1}), \tfrac{-v}{8}(2 - \tau \pm \sqrt{\tau^2 - 12\tau + 4})\}$. Observe that for any $v > 0$, $x^*$ is stable for $\tau = 1$ since $\mathrm{spec}(-J_\tau(x^*)) \subset \mathbb{C}_-^\circ$, but $x^*$ is unstable for a range of learning rates since $\mathrm{spec}(-J_\tau(x^*)) \not\subset \mathbb{C}_-^\circ$ for all $\tau \in (2, \infty)$. This is not an artifact of the quadratic example: games can be constructed in which stable critical points lacking game-theoretic meaning become unstable for all $\tau > \tau_0$ even in the presence of multiple equilibria.*

---

[3]See Lancaster & Tismenetsky (1985); Magnus (1988) for more detail on the definition and properties of these mathematical operators, and Appendix C for more detail directly related to their use in this paper.

This example demonstrates a finite timescale separation can prevent convergence to critical points lacking game-theoretic meaning. We now characterize this behavior generally. Note that Theorem 1 implies that for any critical point which is not a differential Stackelberg equilibrium, there is no finite $\tau^*$ such that $\mathrm{spec}(-J_\tau(x^*)) \subset \mathbb{C}_-^\circ$ for all $\tau \in (\tau^*, \infty)$. In particular, there exists at least one finite, positive value of $\tau$ such that $\mathrm{spec}(-J_\tau(x^*)) \not\subset \mathbb{C}_-^\circ$. We can extend this result to answer the question of whether there exists a finite learning rate ratio $\tau_0$ such that $-J_\tau(x^*)$ has at least one eigenvalue with strictly positive real part for all $\tau \in (\tau_0, \infty)$, thereby implying that $x^*$ is unstable.

**Theorem 2** (Non-Asymptotic Construction of Sufficient Condition for Instability.). *Consider a zero-sum game $(f_1, f_2) = (f, -f)$ defined by $f \in C^r(X, \mathbb{R})$ for some $r \geq 2$. Suppose that $x^*$ is such that $g(x^*) = 0$, $\det(D_2^2 f_2(x^*) \neq 0$, and $x^*$ is not a differential Stackelberg equilibrium. Then $\mathrm{spec}(-J_\tau(x^*)) \not\subset \mathbb{C}_-^\circ$ for all $\tau \in (\tau_0, \infty)$ with*

$$\tau_0 = \lambda_{\max}^+(Q_2^{-1}((P_1 D_{12} f(x^*) + \mathtt{S}_1(-J(x^*))L_0^\top P_2)^\top Q_1^{-1}(P_1 D_{12} f(x^*)$$
$$+ \mathtt{S}_1(-J(x^*))L_0^\top P_2) - P_2 L_0 D_{12} f(x^*) - (P_2 L_0 D_{12} f(x^*))^\top)).$$

*where $P_1, P_2, Q_1, Q_2$ are any non-singular Hermitian matrices such that (a) $Q_i > 0$ for each $i = 1, 2$, (b) $\mathtt{S}_1(-J(x^*))P_1 + P_1 \mathtt{S}_1(-J(x^*)) = Q_1$ and $D_2^2 f(x^*)P_2 + P_2 D_2^2 f(x^*) = Q_2$, and (c) the following matrix pairs have the same inertia: $(P_1, \mathtt{S}_1(-J(x^*)))$ and $(P_2, D_2^2 f(x^*))$.*

*Proof Sketch.* The full proof is provided in Appendix D. The key idea is to leverage the Lyapunov equation and Lemma B.3 to show that $-J_\tau(x^*)$ has at least one eigenvalue with strictly positive real part. Indeed, Lemma B.3 states that if $\mathtt{S}_1(-J(x^*))$ has no zero eigenvalues, then there exists matrices $P_1 = P_1^\top$ and $Q_1 = Q_1^\top > 0$ such that $P_1 \mathtt{S}_1(-J(x^*)) + \mathtt{S}_1(-J(x^*))P_1 = Q_1$ where $P_1$ and $\mathtt{S}_1(-J(x^*))$ have the same *inertia*—that is, the number of eigenvalues with positive, negative and zero real parts, respectively, are the same. An analogous statement applies to $-D_2^2 f(x^*)$ with some $P_2$ and $Q_2$. Since $x^*$ is a non-equilibrium critical point, without loss of generality, let $\mathtt{S}_1(-J(x^*))$ have at least one strictly positive eigenvalue so that $P_1$ does as well. Next, we construct a matrix $P$ that is *congruent* to $\mathrm{blockdiag}(P_1, P_2)$ and a matrix $Q_\tau$ such that $-P J_\tau(x^*) - J_\tau^\top(x^*)P = Q_\tau$. Since $P$ and $\mathrm{blockdiag}(P_1, P_2)$ are congruent, Sylvester's law of inertia implies that they have the same number of eigenvalues with positive, negative, and zero real parts, respectively. Hence, $P$ has at least one eigenvalue with strictly positive real part. We then construct $\tau_0$ via an eigenvalue problem such that for all $\tau > \tau_0$, $Q_\tau > 0$. Applying Lemma B.3 again, for any $\tau > \tau_0$, $-J_\tau(x^*)$ has at least one eigenvalue with strictly positive real part so that $\mathrm{spec}(-J_\tau(x^*)) \not\subset \mathbb{C}_-^\circ$. □

### 3.3 Regularization with Applications to Adversarial Learning

In this section, we focus on generative adversarial networks with regularization and using the theory developed so far extend the results to provide a stability guarantee for a range of regularization parameters and learning rate ratios. Consider the training objective

$$f(\theta, \omega) = \mathbb{E}_{p(z)}[\ell(\mathrm{D}(\mathrm{G}(z; \theta); \omega))] + \mathbb{E}_{p_\mathcal{D}(x)}[\ell(-\mathrm{D}(x; \omega))] \tag{5}$$

where $\mathrm{D}_\omega(x)$ and $\mathrm{G}_\theta(z)$ are discriminator and generator networks, $p_\mathcal{D}(x)$ is the data distribution while $p(z)$ is the latent distribution, and $\ell \in C^2(\mathbb{R})$ is some real-value function.[4] Nagarajan & Kolter (2017) show, under suitable assumptions, that gradient-based methods for training generative adversarial networks are locally convergent assuming the data distributions are absolutely continuous. However, as observed by Mescheder et al. (2018), such assumptions not only may not be satisfied by many practical generative adversarial network training scenarios such as natural images, but often the data distribution is concentrated on a lower dimensional manifold. The latter characteristic leads to highly ill-conditioned problems and nearly purely imaginary eigenvalues.

Gradient penalties ensure that the discriminator cannot create a non-zero gradient which is orthogonal to the data manifold without suffering a loss. Introduced by Roth et al. (2017) and refined in Mescheder et al. (2018), we consider training generative adversarial networks with one of two fairly natural gradient-penalties used to regularize the discriminator:

$$R_1(\theta, \omega) = \frac{\mu}{2}\mathbb{E}_{p_\mathcal{D}(x)}[\|\nabla_x \mathrm{D}(x; \omega)\|^2] \quad \text{and} \quad R_2(\theta, \omega) = \frac{\mu}{2}\mathbb{E}_{p_\theta(x)}[\|\nabla_x \mathrm{D}(x; \omega)\|^2],$$

---

[4]For example, $\ell(x) = -\log(1 + \exp(-x))$ gives the original formulation of Goodfellow et al. (2014).

where, by a slight abuse of notation, $\nabla_x(\cdot)$ denotes the partial gradient with respect to $x$ of the argument $(\cdot)$ when the argument is the discriminator $D(\cdot; \omega)$ in order prevent any conflation between the notation $D(\cdot)$ elsewhere for derivatives. Let $h_1(\theta) = \mathbb{E}_{p_\theta(x)}[\nabla_\omega D(x; \omega)|_{\omega=\omega^*}]$ and $h_2(\omega) = \mathbb{E}_{p_\mathcal{D}(x)}[|D(x; \omega)|^2 + \|\nabla_x D(x; \omega)\|^2]$. Define *reparameterization manifolds* $\mathcal{M}_G = \{\theta : p_\theta = p_\mathcal{D}\}$ and $\mathcal{M}_D = \{\omega : h_2(\omega) = 0\}$ and let $T_{\theta^*}\mathcal{M}_G$ and $T_{\omega^*}\mathcal{M}_D$ denote their respective tangent spaces at $\theta^*$ and $\omega^*$. As in Mescheder et al. (2018), we make the following assumption.

**Assumption 1.** *Consider a zero-sum game of the form given in* (5) *where* $f \in C^2(\mathbb{R}^{n_1} \times \mathbb{R}^{n_2}, \mathbb{R})$ *and* $G(\cdot; \theta)$ *and* $D(\cdot; \omega)$ *are the generator and discriminator networks, respectively, and* $x = (\theta, \omega) \in \mathbb{R}^{n_1} \times \mathbb{R}^{n_2}$. *Suppose that* $x^* = (\theta^*, \omega^*)$ *is an equilibrium. Then, (a) at* $(\theta^*, \omega^*)$, $p_{\theta^*} = p_\mathcal{D}$ *and* $D(x; \omega^*) = 0$ *in some neighborhood of* $\mathrm{supp}(p_\mathcal{D})$, *(b) the function* $\ell \in C^2(\mathbb{R})$ *satisfies* $\ell'(0) \neq 0$ *and* $\ell''(0) < 0$, *(c) there are* $\epsilon$–*balls* $B_\epsilon(\theta^*)$ *and* $B_\epsilon(\omega^*)$ *centered around* $\theta^*$ *and* $\omega^*$, *respectively, so that* $\mathcal{M}_G \cap B_\epsilon(\theta^*)$ *and* $\mathcal{M}_D \cap B_\epsilon(\omega^*)$ *define* $C^1$-*manifolds. Moreover, (i) if* $w \notin T_{\theta^*}\mathcal{M}_G$, *then* $w^\top \nabla_w h_1(\theta^*) w \neq 0$, *and (ii) if* $v \notin T_{\omega^*}\mathcal{M}_D$, *then* $v^\top \nabla_\omega^2 h_2(\omega^*) v \neq 0$.

We note that as explained by Mescheder et al. (2018), Assumption 1.c(i) implies that the discriminator is capable of detecting deviations from the generator distribution in equilibrium, and Assumption 1.c(ii) implies that the manifold $\mathcal{M}_D$ is sufficiently regular and, in particular, its (local) geometry is captured by the second (directional) derivative of $h_2$.

**Theorem 3.** *Consider training a generative adversarial network via a zero-sum game with generator network* $G_\theta$, *discriminator network* $D_\omega$, *and loss* $f(\theta, \omega)$ *with regularization* $R_j(\theta, \omega)$ *(for either* $j = 1$ *or* $j = 2$) *and any regularization parameter* $\mu \in (0, \infty)$ *such that Assumption 1 is satisfied for an equilibrium* $x^* = (\theta^*, \omega^*)$ *of the regularized dynamics. Then,* $x^* = (\theta^*, \omega^*)$ *is a differential Stackelberg equilibrium. Furthermore, for any* $\tau \in (0, \infty)$, $\mathrm{spec}(-J_{(\tau, \mu)}(x^*)) \subset \mathbb{C}_-^\circ$.

# 4    PROVABLE CONVERGENCE OF GDA WITH TIMESCALE SEPARATION

In this section, we characterize the asymptotic convergence rate for $\tau$-GDA to differential Stackelberg equilibria, and provide a finite time guarantee for convergence to an $\varepsilon$–approximate equilibrium. The asymptotic convergence rate result uses Theorem 1 to construct a finite $\tau^* \in (0, \infty)$ such that $x^*$ is stable, meaning $\mathrm{spec}(-J_\tau(x^*)) \subset \mathbb{C}_-^\circ$, and then for any $\tau \in (\tau^*, \infty)$, the two key lemmas—namely, Lemmas F.1 and F.2 in Appendix F—imply a local asymptotic convergence rate.

**Theorem 4.** *Consider a zero-sum game* $(f_1, f_2) = (f, -f)$ *defined by* $f \in C^r(X, \mathbb{R})$ *for* $r \geq 2$ *and let* $x^*$ *be a differential Stackelberg equilibrium of the game. There exists a* $\tau^* \in (0, \infty)$ *such that for any* $\tau \in (\tau^*, \infty)$ *and* $\alpha \in (0, \gamma)$, $\tau$-GDA *with learning rate* $\gamma_1 = \gamma - \alpha$ *converges locally asymptotically at a rate of* $O((1 - \alpha/(4\beta))^{k/2})$ *where* $\gamma = \min_{\lambda \in \mathrm{spec}(J_\tau(x^*))} 2\mathrm{Re}(\lambda)/|\lambda|^2$, $\lambda_\mathtt{m} = \arg\min_{\lambda \in \mathrm{spec}(J_\tau(x^*))} 2\mathrm{Re}(\lambda)/|\lambda|^2$, *and* $\beta = (2\mathrm{Re}(\lambda_\mathtt{m}) - \alpha|\lambda_\mathtt{m}|^2)^{-1}$. *Moreover, if* $x^*$ *is a differential Nash equilibrium,* $\tau^* = 0$ *so that for any* $\tau \in (0, \infty)$ *and* $\alpha \in (0, \gamma)$, $\tau$-GDA *with* $\gamma_1 = \gamma - \alpha$ *converges with a rate* $O((1 - \alpha/(4\beta))^{k/2})$.

To build some intuition, consider a differential Stackelberg equilibrium $x^*$ and its corresponding $\tau^*$ obtained via Theorem 1 so that for any fixed $\tau \in (\tau^*, \infty)$, $\mathrm{spec}(-J_\tau(x^*)) \subset \mathbb{C}_-^\circ$. For the discrete time system $x_{k+1} = x_k - \gamma_1 \Lambda_\tau g(x_k)$, if $\gamma_1$ is chosen such that the spectral radius of the local linearization of the discrete time map is a contraction, then $x_k$ locally (exponentially) converges to $x^*$ (cf. Proposition B.1). With this in mind, we formulate an optimization problem to find the upper bound $\gamma$ on the learning rate $\gamma_1$ such that for all $\gamma_1 \in (0, \gamma)$, $\rho(I - \gamma_1 J_\tau(x^*)) < 1$; indeed, let $\gamma = \min_{\gamma > 0} \{\gamma : \max_{\lambda \in \mathrm{spec}(J_\tau(x^*))} |1 - \gamma\lambda| \leq 1\}$. The intuition is as follows. The inner maximization problem is over a finite set $\mathrm{spec}(J_\tau(x^*)) = \{\lambda_1, \ldots, \lambda_n\}$ where $J_\tau(x^*) \in \mathbb{R}^{n \times n}$. As $\gamma$ increases away from zero, each $|1 - \gamma\lambda_i|$ shrinks in magnitude. The last $\lambda_i$ such that $1 - \gamma\lambda_i$ hits the boundary of the unit circle in the complex plane gives us the optimal $\gamma$ and the $\lambda_\mathtt{m} \in \mathrm{spec}(J_\tau(x^*))$ that achieves it. Examining the constraint, we have that for each $\lambda_i$, $\gamma(\gamma|\lambda_i|^2 - 2\mathrm{Re}(\lambda_i)) \leq 0$ for any $\gamma > 0$. As noted this constraint will be tight for one of the $\lambda$, in which case $\gamma = 2\mathrm{Re}(\lambda)/|\lambda|^2$ since $\gamma > 0$. Hence, by selecting $\gamma = \min_{\lambda \in \mathrm{spec}(J_\tau(x^*))} 2\mathrm{Re}(\lambda)/|\lambda|^2$, we have that $|1 - \gamma_1\lambda| < 1$ for all $\lambda \in \mathrm{spec}(J_\tau(x^*))$ and any $\gamma_1 \in (0, \gamma)$. From here, one can use standard arguments from numerical analysis to show that for the choice of $\alpha$ and $\beta$, the claimed asymptotic rate holds.

Theorem 4 directly implies a finite time convergence guarantee for obtaining an $\varepsilon$-differential Stackelberg equilibrium, that is, a point with an $\varepsilon$-ball around a differential Stackelberg equilibrium $x^*$.

**Corollary 1.** *Given $\varepsilon > 0$, under the assumptions of Theorem 4, $\tau$-GDA obtains an $\varepsilon$–differential Stackleberg equilibrium in $\lceil (4\beta/\alpha) \log(\|x_0 - x^*\|/\varepsilon) \rceil$ iterations for any $x_0 \in B_\delta(x^*)$ with $\delta = \alpha/(4L\beta)$ where $L$ is the local Lipschitz constant of $I - \gamma J_\tau(x^*)$.*

Moreover, the convergence rates and finite time guarantees extend to the gradient penalty regularized generative adversarial network described in the preceeding section.

**Corollary 2.** *Under the assumptions of Theorems 3 and 4, for any fixed $\mu \in (0, \infty)$ and $\tau \in (0, \infty)$, $\tau$-GDA converges locally asymptotically at a rate of $O((1 - \alpha/(4\beta))^{k/2})$, and achieves an $\varepsilon$-equilibrium in $\lceil (4\beta/\alpha) \log(\|x_0 - x^*\|/\varepsilon) \rceil$ iterations for any $x_0 \in B_\delta(x^*)$.*

In Appendix H, we extend the convergence analysis to the stochastic setting.

## 5 EXPERIMENTS

We now present numerical experiments and Appendix K contains further simulations and details.

**Dirac-GAN: Regularization, Timescale Separation, and Convergence Rate.** The Dirac-GAN (Mescheder et al., 2018) consists of a univariate generator distribution $p_\theta = \delta_\theta$ and a linear discriminator $D(x; \omega) = \omega x$, where the real data distribution $p_\mathcal{D}$ is given by a Dirac-distribution concentrated at zero. The resulting zero-sum game is defined by the cost $f(\theta, \omega) = \ell(\theta\omega) + \ell(0)$ and the unique critical point $(\theta^*, \omega^*) = (0, 0)$ is a local Nash equilibrium. However, the eigenvalues of the Jacobian are purely imaginary regardless of the choice of timescale separation so that $\tau$-GDA oscillates and fails to converge. This behavior is expected since the equilibrium is not hyperbolic and corresponds to neither a differential Nash equilibrium nor a differential Stackelberg equilibrium but it is undesirable nonetheless. The zero-sum game corresponding to the Dirac-GAN with regularization can be defined by the cost $f(\theta, \omega) = \ell(\theta\omega) + \ell(0) - \frac{\mu}{2}\omega^2$. The unique critical point remains unchanged, but for all $\tau \in (0, \infty)$ and $\mu \in (0, \infty)$ the equilibrium of the unregularized game is stable and corresponds to a differential Stackelberg equilibrium of the regularized game.

From Figures 1a and 1f, we observe that the impact of timescale separation with regularization $\mu = 0.3$ is that the trajectory is not as oscillatory since it moves faster to the zero line of $-D_2 f(\theta, \omega)$ and then follows along that line until reaching the equilibrium. We further see from Figure 1b that with regularization $\mu = 0.3$, $\tau$-GDA with $\tau = 8$ converges faster to the equilibrium than $\tau$-GDA with $\tau = 16$, despite the fact that the former exhibits some cyclic behavior in the dynamics while the

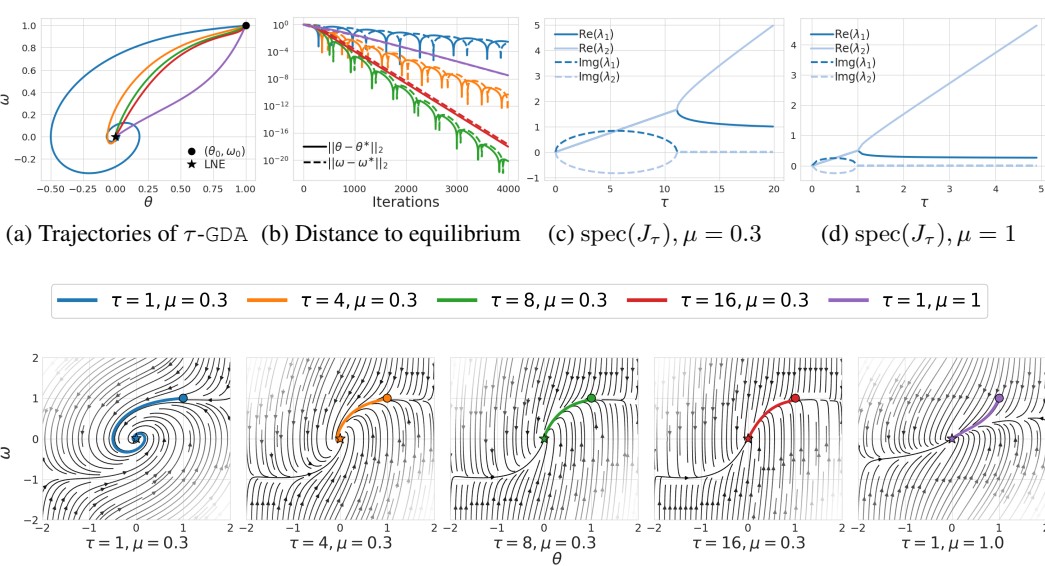

(a) Trajectories of $\tau$-GDA  (b) Distance to equilibrium  (c) $\mathrm{spec}(J_\tau), \mu = 0.3$  (d) $\mathrm{spec}(J_\tau), \mu = 1$

(f) Trajectories of $\tau$-GDA overlayed on vector fields generated by choices of $\tau$ and $\mu$.

Figure 1: Experimental results for the Dirac-GAN game of Section 5.

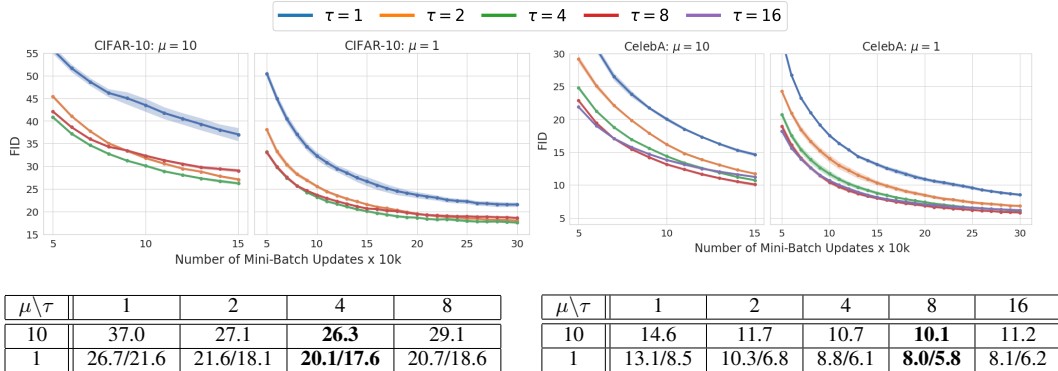

| $\mu\backslash\tau$ | 1 | 2 | 4 | 8 |
|---|---|---|---|---|
| 10 | 37.0 | 27.1 | **26.3** | 29.1 |
| 1 | 26.7/21.6 | 21.6/18.1 | **20.1/17.6** | 20.7/18.6 |

| $\mu\backslash\tau$ | 1 | 2 | 4 | 8 | 16 |
|---|---|---|---|---|---|
| 10 | 14.6 | 11.7 | 10.7 | **10.1** | 11.2 |
| 1 | 13.1/8.5 | 10.3/6.8 | 8.8/6.1 | **8.0/5.8** | 8.1/6.2 |

Figure 2: CIFAR-10 FID  Figure 3: CelebA FID

latter does not. The eigenvalues of the Jacobian with regularization $\mu = 0.3$ presented in Figure 1c explains this behavior since the imaginary parts are non-zero with $\tau = 8$ and zero with $\tau = 16$, while the eigenvalue with the minimum real part is greater at $\tau = 8$ than at $\tau = 16$. This highlights that some oscillatory behavior in the dynamics is not always harmful for convergence. For $\mu = 1$ and $\tau = 1$, Figures 1a and 1b show that even though $\tau$-GDA does not cycle since the eigenvalues of the Jacobian are purely real, the trajectory converges slowly to the equilibrium. Indeed, for each regularization parameter, the eigenvalues of $J_\tau(\theta^*, \omega^*)$ split after becoming purely real and then converge toward the eigenvalues of $\mathrm{S}_1(J(\theta^*, \omega^*))$ and $-\tau D_2^2 f(\theta^*, \omega^*)$. Since $\mathrm{S}_1(J(\theta^*, \omega^*)) \propto 1/\mu$ and $-\tau D_2^2 f(\theta^*, \omega^*) \propto \tau\mu$, there is a trade-off between the choice of regularization $\mu$ and the timescale separation $\tau$ on the conditioning of the Jacobian matrix that dictates the convergence rate.

**Generative Adversarial Networks: Image Datasets.** We build on the implementations of Mescheder et al. (2018) and train with the non-saturating objective and the $R_1$ gradient penalty. The network architectures are both ResNet based. We fix the initial learning rate for the generator to be $\gamma_1 = 0.0001$ with CIFAR-10 and $\gamma_1 = 0.00005$ for CelebA. The learning rates are decayed so that $\gamma_{1,k} = \gamma_1/(1 + \nu)^k$ and $\gamma_{2,k} = \tau\gamma_{1,k}$ are the generator and discriminator learning rates at update $k$ where $\nu = 0.005$. The batch size is 64, the latent data is drawn from a standard normal of dimension 256, and the resolution of the images is $32 \times 32 \times 3$. We run RMSprop with parameter $\alpha = 0.99$ and retain an exponential moving average of the generator parameters for evaluation with parameter $\beta = 0.9999$. We remark that RMSprop is an adaptive method that builds on GDA and is commonly used in training for image datasets. It is adopted here to explore the interplay with timescale separation and to determine if similar observations emerge compared to our extensive experiments with $\tau$-GDA (see Appendix K). The FID scores (Heusel et al., 2017) along the learning path and in numeric form at 150k/300k mini-batch updates for CIFAR-10 and CelebA with regularization parameters $\mu = 10$ and $\mu = 1$ are presented in Figures 2 and 3, respectively. The experiments were each repeated with 3 random seeds which yielded similar results and the mean scores are reported. The choices of $\tau = 4$ and $\tau = 8$ converge fastest with each regularization parameter for CIFAR-10 and CelebA, respectively. The performance with regularization $\mu = 1$ is superior to that with $\mu = 10$, which highlights the interplay between timescale separation and regularization. Moreover, we see that timescale separation improves convergence until hitting a limiting value. These conclusions agree with the insights from the simple Dirac-GAN experiment. Finally, it is worth reiterating there is a coupling between $\tau$ and $\gamma_1$: $\tau$ must be selected so that the continuous-time system is stable and then $\gamma_1$ must be chosen so that the discrete-time update is both stable and numerically well-conditioned for the choice of $\tau$.

## 6 CONCLUSION

We prove gradient descent-ascent locally converges to a critical point for a range of finite learning rate ratios if and only if the critical point is a differential Stackelberg equilibrium. This answers a standing open question about the local convergence of first order methods to local minimax equilibria. A key component of the proof is the construction of a (tight) finite lower bound on the learning rate ratio $\tau$ for which stability is guaranteed, and hence local asymptotic convergence of $\tau$-GDA.

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

APPENDIX

Below we provide a table of contents as a guide to the appendix.

TABLE OF CONTENTS

Section D contains the proof of our second main result on the existence of a finite range of values of $\tau$ such that non-equilibrium critical points which are stable for some $\tau$, become unstable for all $\tau$ in a range of $\tau$'s lower bounded by a finite value.

Section E contains the proof of Theorem 3 which is an application of the main results to the important adversarial learning application of training generative adversarial networks. Within the section, we also provide necessary conditions on the sizes of the network architectures for the discriminator and generator network for stability.

Section F contains the statement and proofs of two core technical lemmas on the convergence rate of $\tau$-GDA. We then show how to invoke them together with the stability results to obtain Theorem 4.

Section H contains an extension of the deterministic convergence guarantees to the stochastic setting in which agents have access only to an unbiased estimator of their gradient. We provide convergence guarantees for $\tau$-GDA in this setting as well under the least restrictive, to our knowledge, assumptions on stochastic approximation updates to date.

Section I contains a proof of the result for the limiting case of $\tau \to \infty$. While this result appeared recently in Jin et al. (2020), an analogous result has been known for some time in the theory of singularly perturbed dynamical systems. Since the proof techniques are particularly illuminating

and expose a new set of tools to this community, we provide this alternative proof from the singular perturbation theory perspective.

## J   Further Details on Related Work                                                                                    42

Section J contains an extended discussion on the related work by Jin et al. (2020). Specifically, we expand on the relationship between Proposition 27 in Jin et al. (2020), which provides examples of games such that the stable critical points of $\tau$-GDA are not a subset of local minmax and vice versa, and our main results. We show that the examples in these results, while illustrative of the fact intended to be shown by the proposition stated, do not conflict with out findings.

## K   Experiments Supplement                                                                                             44

Section K contains extended experimental results including applications to illustrative games (such as those examples included in the main body) as well as extended GAN experiments. Our results show that we are able to leverage $\tau$ and improve training results. These results are promising as they suggest that understanding the relationship between the three key hyperparameters—timescale separation $\tau$, regularization $\mu$, and exponential moving averaging weight $\beta$—can improve first order training (and hence, scalable) algorithms.

## L   Alternative Proof of Theorem 3 via $\tau^*$ Construction from Theorem 1                                            60

In order to highlight the utility of Theorem 1 along with future directions of obtaining values of $\tau^*$ for structured games, we revisit the proof of Theorem 3 and derive the result directly from the construction. The purpose of this section is to illustrate that the structure of equilibria considered in Theorem 3 can be exploited to obtain the value of $\tau^*$ for the entire class of games using properties of the Kronecker product and sum.

## A   RELATED WORK

In this section, we provide a review of related work at the intersection machine learning and game theory, as well as connections to dynamical systems theory and control.

### A.1   MACHINE LEARNING AND LEARNING IN GAMES

Given the extensive work on the topic of learning in games in machine learning that has gone on over the last several years, we cannot cover all of it and instead focus our attention on only the most relevant to this paper. We begin by reviewing solution concepts developed for the class of games under consideration and then discuss some learning dynamics studied in the literature beyond gradient descent-ascent. Following this, we delineate the related work studying gradient descent-ascent in non-convex, non-concave zero-sum games and finish by making note of the literature on bilinear and non-convex, concave zero-sum games.

**Solution Concepts.**   Owing to the numerous applications in machine learning, a significant portion of the modern work on learning in games has focused on the zero-sum formulation with non-convex, non-concave cost functions. Most recently, Daskalakis et al. (2021) tout the importance and significance of this class of games in a paper on the complexity of finding equilibria (in particular, in the constrained setting) in such games. Consequently, local solution concepts have been broadly adopted. Compared to the standard game-theoretic notions of equilibrium that characterize player's incentive to deviate given the game and information structure, local equilibrium concepts restrict the deviation search space to a suitable local neighborhood. Following the standard game-theoretic viewpoint, a vast number of works in machine learning study the local Nash equilibrium concept and critical points satisfying gradient-based sufficient conditions for the equilibrium, which are often referred to as differential Nash equilibria (Ratliff et al., 2013; 2014; Ratliff et al., 2016). Based on the observation that in non-convex, non-concave zero-sum games the order of play is fundamental in the definition of the game, there has been a push toward considering local notions of the Stackelberg equilibrium concept, which is the usual game-theoretic equilibrium when there is an explicit order of play between players. In the zero-sum formulation, Stackelberg equilibrium are often referred to

as minmax equilibria. Similar to as for the Nash equilibrium, gradient-based sufficient conditions for local minmax/Stackelberg equilibrium have been given (Fiez et al., 2020; Jin et al., 2020) and such critical points have been referred to as differential Stackelberg equilibria (Fiez et al., 2020). We remark that it has been shown that local/differential Nash equilibria are a subset of local/differential Stackelberg equilibria (Fiez et al., 2020; Jin et al., 2020). Following past works, we adopt the terminology of differential Nash equilibrium and differential Stackelberg equilibrium in this paper as the meaning of strict local Nash equilibrium and strict local minmax/Stackelberg equilibrium, respectively. It is also worth mentioning the proximal equilibria proposed by Farnia & Ozdaglar (2020), which we do not consider in this work, that depending on a regularization parameter can interpolate between the local Nash and local Stackelberg equilibrium notions. Finally, Zhang et al. (2020a) propose a local robust point as an equilibrium definition that contains both local minmax and local maxmin equilibrium and Vlatakis-Gkaragkounis et al. (2019) consider an equilibrium concept in what they deem hidden bilinear games.

**Learning Dynamics.**    Given that the focus of this work is on gradient descent-ascent, we center our coverage of related work on papers analyzing its behavior. Nonetheless, we mention that a significant number of learning dynamics for zero-sum games have been developed in the past few years, in some cases motivated by the shortcomings of gradient descent-ascent without timescale separation. The methods include optimistic and extra-gradient algorithms (Daskalakis et al., 2018; Gidel et al., 2019a; Mertikopoulos et al., 2019), negative momentum (Gidel et al., 2019b), gradient adjustments (Balduzzi et al., 2018; Letcher et al., 2019a; Mescheder et al., 2017), and opponent modeling methods (Foerster et al., 2018; Letcher et al., 2019b; Metz et al., 2017; Schäfer & Anandkumar, 2019; Zhang & Lesser, 2010), among others. While the aforementioned learning dynamics possess some desirable characteristics, they cannot guarantee that the set of stable critical points coincide with a set of local equilibria for the class of games under consideration. However, there have been a select few learning dynamics proposed that can guarantee the stable critical points coincide with either the set of differential Nash equilibria (Adolphs et al., 2019; Mazumdar et al., 2019) or the set of differential Stackelberg equilibria (Fiez et al., 2020; Wang et al., 2020; Zhang et al., 2020b)— effectively solving the problem of guaranteeing local convergence to only a class of local equilibria. However, since each of the algorithms achieving the equilibrium stability guarantee require solving a linear equation in each update step, they are not efficient and can potentially suffer from degeneracies along the learning path in applications such as generative adversarial networks. These practical shortcomings motivate either proving that existing learning dynamics using only first-order gradient feedback achieve analogous theoretical guarantees or developing novel computationally efficient learning dynamics that can match the theoretical guarantee of interest.

**Gradient Descent-Ascent.**    Gradient descent-ascent has been studied extensively in non-convex, non-concave zero-sum games since it is a natural analogue to gradient descent from optimization, is computationally efficient, and has been shown to be effective in practice for applications of interest when combined with common heuristics. A prevailing approach toward gaining understanding of the convergence characteristics of gradient descent-ascent has been to analyze the local stability around critical points of the continuous time limiting dynamical system. The majority of this work has not considered the impact of timescale separation. Numerous papers have pointed out that the stable critical points of gradient descent-ascent without timescale separation may not be game-theoretically meaningful. In particular, it has been shown that there can exist stable critical points that are not differential Nash equilibrium (Daskalakis & Panageas, 2018; Mazumdar et al., 2020). Furthermore, it is known that there can exist stable critical points that are not differential Stackelberg equilibria (Jin et al., 2020). The aforementioned results rule out the possibility that gradient descent-ascent without timescale separation can guarantee equilibrium convergence. In terms of the stability of equilibria, it is known that differential Nash equilibrium are stable for gradient descent-ascent without timescale separation (Daskalakis & Panageas, 2018; Mazumdar et al., 2020), but that there can exist differential Stackelberg equilibria which are not stable with respect to gradient descent-ascent without timescale separation.

The work of Jin et al. (2020) is the most relevant exploring how the aforementioned stability properties of gradient descent-ascent change with timescale separation. In particular, Jin et al. (2020) investigate whether the desirable stability characteristics (stability of differential Nash equilibria) and undesirable stability characteristics (stability of non-equilibrium critical points and instability of differential Stackelberg equilibria) of gradient descent without timescale separation are main-

tained and remedied, respectively with timescale separation. In terms of the former query, extending the examples shown in Mazumdar et al. (2020) and Daskalakis & Panageas (2018), Jin et al. (2020) show that differential Nash equilibrium are stable for gradient descent-ascent with any amount of timescale separation.

On the other hand, for the latter query, Jin et al. (2020) shows (in Proposition 27) two interesting examples: (a) for an a priori fixed $\tau$, there exists a game with a differential Stackelberg equilibrium that is not stable and (b) for an a priori fixed $\tau$, there exists a game with a stable critical point that is not a differential Stackelberg equilibrium. However, (a) does not imply that for the constructed game, there does not exist another (finite) $\tau$—independent of the game parameters—such the differential Stackelebrg equilibrium is stable for all larger $\tau$. In simple language, the result summarized in (a) says the following: *if a bad timescale separation is chosen, then convergence may not be guaranteed*. Similarly, (b) does not imply that there is no $\tau$ such that for all larger $\tau$ for the constructed game instance, the critical point becomes unstable. Again, in simple language, the result summarized in (b) says the following: *if a bad timescale separation is chosen, then non-game theoretically meaningful equilibria may persist*. While at first glance this set of results may appear to indicate that the undesirable stability characteristics of gradient descent without timescale separation cannot be averted by any finite timescale separation, it is important to emphasize that these results *do not* answer the questions of whether there (a) exists a game with a critical point that is not a differential Stackelberg equilibrium which is stable with respect to gradient descent-ascent without timescale separation and remains stable for all finite timescale separation ratios or (b) exists a game with a differential Stackelberg equilibrium that is not stable for all finite timescale separation ratios. The preceding questions are left open from previous work and are exactly the focus of this paper. In Appendix J, we go into greater detail on the comparison between Proposition 27 of Jin et al. (2020) as we believe this to be an important point of distinction between Theorem 1 and 2 in this paper.

Finally, Jin et al. (2020) study `GDA` with a timescale separation approaching infinity and show that the stable critical points of gradient descent-ascent coincide with the set of differential Stackelberg equilibria in this regime. This result effectively shows that gradient descent-ascent can guarantee only equilibrium convergence with timescale separation, albeit only when it becomes arbitrarily large. We remark that an equivalent result in the context of general singularly perturbed systems has been known in the literature (Kokotovic et al., 1986, Chap. 2) as we discuss further in Section I. Finally, we point out that since a timescale separation approaching infinity does not result in an implementable algorithm, fully understanding the behavior with a finite timescale separation is of fundamental importance and the motivation for our work.

Beyond the work of Jin et al. (2020) considering timescale separation in gradient descent-ascent, it is worth mentioning the work of Chasnov et al. (2019) and Heusel et al. (2017). Chasnov et al. (2019) study the impact of timescale separation on gradient descent-ascent, but focus on the convergence rate as a function of it given an initialization around a differential Nash equilibrium and do not consider the stability questions examined in this paper. Heusel et al. (2017) study stochastic gradient descent-ascent with timescale separation and invoke the results of Borkar (2008) for analysis. The stochastic approximation results the claims rely on guarantee the convergence of the system locally to a stable critical point. Consequently, the claim of convergence to differential Nash equilibria of stochastic gradient descent-ascent given by Heusel et al. (2017) only holds given an initialization in a local neighborhood around a differential Nash equilibrium. In this regard, the issue of the local stability of the types of critical point is effectively assumed away and not considered. In contrast, we are able to combine our stability results for gradient descent-ascent with timescale separation together with the stochastic approximation theory of Borkar (2008) to guarantee local convergence to a differential Stackelberg equilibrium in Section H. We remark that Heusel et al. (2017) empirically demonstrate that timescale separation can significantly improve the performance of gradient descent-ascent when training generative adversarial networks.

The final relevant line of work studying gradient descent-ascent is specific to generative adversarial networks. The results from this literature develop assumptions relevant to generative adversarial networks and then analyze the stability and convergence properties of gradient descent-ascent under them (see, e.g., works by Daskalakis et al. (2018); Goodfellow et al. (2014); Mescheder et al. (2018); Metz et al. (2017); Nagarajan & Kolter (2017)). Within this body of work, there has been a significant amount of effort focusing on how the stability (and, hence, convergence properties) of gradient descent-ascent in generative adversarial networks can be enhanced with regularization methods. Nagarajan & Kolter (2017) show, under suitable assumptions, that gradient-based methods

for training generative adversarial networks are locally convergent assuming the data distributions are absolutely continuous. However, as observed by Mescheder et al. (2018), such assumptions not only may not be satisfied by many practical generative adversarial network training scenarios such as natural images, but it can often be the case that the data distribution is concentrated on a lower dimensional manifold. The latter characteristic leads to nearly purely imaginary eigenvalues and highly ill-condition problems. Mescheder et al. (2018) provide an explanation for observed instabilities consequent of the true data distribution being concentrated on a lower dimensional manifold using discriminator gradients orthogonal to the tangent space of the data manifold. Further, the authors introduce regularization via gradient penalties that leads to convergence guarantees under less restrictive assumptions than were previously known. In this paper, we further extend these results to show that convergence to differential Stackelberg equilibria is guaranteed under a wide array of hyperparameter configurations (i.e., learning rate ratio and regularization).

**Bilinear Games.** We would be remiss to not include a discussion of gradient methods applied to an important class of zero-sum games: bilinear games. Bilinear games fall within a measure zero set of $C^2$ games that do not possess the generic properties that at critical points $x^*$, $\det(D_2^2 f(x^*)) \neq 0, \det(\mathtt{S}_1(J(x^*))) \neq 0$ (see Fiez et al. 2020 for the genericity statements regarding local minmax equilibria in zero-sum games). This means they need special treatment. For zero-sum bilinear unconstrained games, however, the behavior of gradient descent-ascent is already known. In particular, the zero point is always a center type equilibrium of the continuous time dynamics so that the continuous time dynamics are recurrent. A forward Euler discretization of the continuous time dynamics gives gradient descent-ascent and such dynamics will always diverge. The alternating gradient descent update introduced by Bailey et al. (2020) is one solution to this problem such that the behavior of the continuous time dynamics is preserved under the discretization. This method simply uses a different discretization scheme that respects the continuous time behavior. We also note that due to the fact that zero-sum bilinear games do not admit equilibria with generic properties, simply introducing regularization can remedy the problem. In particular, if the maximizing player's update is modified to include the derivative of $-\frac{\mu}{2}\|y\|_2^2$, then the local minmax (which is not strict) becomes a strict local minmax for the regularized game. In this case, our results do apply.

**Nonconvex-Concave Optimization.** A final related line of work is on nonconvex-concave optimization (Lin et al., 2020a;b; Lu et al., 2020; Nouiehed et al., 2019; Ostrovskii et al., 2020; Rafique et al., 2018). The focus in this set of works (among many others on the topic) is on characterizing the iteration complexity to stationary points, rather than stability and asymptotic convergence as in the non-convex, non-concave zero-sum game setting. The primary relevance of work on this problem is that a number of the algorithms rely on timescale separation and variations of gradient descent-ascent. Moreover, the methods for obtaining fast convergence rates may be relevant to future work attempting to characterize fast rates in the non-convex, non-concave setting after there is a more fundamental understanding of the stability and asymptotic convergence.

## A.2 Historical Perspective: Dynamical Systems and Control

The study of gradient descent-ascent dynamics with timescale separation between the minimizing and maximizing players is closely related to that of singularly perturbed dynamical systems (Kokotovic et al., 1986). Such systems arise in classical control and dynamical systems in the context of physical systems that either have multiple states which evolve on different timescales due to some underlying immutable physical process or property, or a single dynamical system which evolves on a sub-manifold of the larger state-space. For example, robot manipulators or end effectors often have have slower mechanical dynamics than electrical dynamics. On the other hand, in electrical circuits or mechanical systems, certain resistor-capacitor circuits or spring-mass systems have a state which evolves subject to a constraint equation (Lagerstrom & Casten, 1972; Sastry & Desoer, 1981). Due to their prevalence, singularly perturbed systems have been studied extensively with one of the outcomes being a number of works on determining the range of perturbation parameters for which the overall system is stable (Kokotovic et al., 1986; Saydy, 1996; Saydy et al., 1990). We exploit these results and analysis techniques to develop novel results for learning in games. One of contributions of this work is the introduction of the algebraic analysis techniques to the machine learning and game theory communities. These tools open up new avenues for algorithm synthesis; we comment on potential directions in the concluding discussion section.

This being said, there are a couple key difference between the present setting and that of the classical literature including the following:

1. **The perturbation parameter is no longer an immutable characteristic of the physical system, but rather a hyperparameter subject to design.** Indeed, in singular perturbation theory, the typical dynamical system studied takes the form

$$\dot{x} = g_1(x, y) \quad \epsilon \dot{y} = g_2(x, y) \tag{6}$$

where $\epsilon$ is a small parameter that abstracts some physical characteristics of the state variables. On the other hand, in learning in games, the continuous time limiting dynamical system of gradient descent-ascent for a zero-sum game defined by $f \in C^2(X \times Y, \mathbb{R})$ takes the form

$$\dot{x} = -D_1 f(x, y) \quad \dot{y} = \tau D_2 f(x, y) \tag{7}$$

where the $x$–player seeks to minimize $f$ with respect to $x$ and the $y$–player seeks to maximize $f$ with respect to $y$, and $\tau$ is the ratio of learning rates (without loss of generality) of the maximizing to the minimizing player. These learning rates—and hence the value of $\tau$—are hyperparameters subject to design in most machine learning and optimization applications. Another feature of (7) as compared to (6), is that the dynamics $D_i f$ are partial derivatives of a function $f$, which leads to the second key difference.

2. **There is structure in the dynamical system that arises from gradient-play which reflects the underlying game theoretic interactions between players.** This structure can be exploited in obtaining convergence guarantees in machine learning and optimization applications of game theory. For instance, minmax optimization is analogous to a zero sum game for which the local linearization of gradient descent-ascent dynamics has the structure

$$J = \begin{bmatrix} A & B \\ -\tau B^\top & -\tau C \end{bmatrix}$$

where $A = A^\top$ and $C = C^\top$ and $\tau$ is the learning rate ratio or timescale separation parameter. Such block matrices have very interesting properties. In particular, second order optimality conditions for a minmax equilibrium correspond to positive definiteness of the first Schur complement $\mathsf{S}_1(J) = A - BC^{-1}B^\top > 0$, and of $-C > 0$ (Fiez et al., 2020). This turns out to be keenly important for understanding convergence of gradient descent-ascent. Furthermore, due to the structure of $J$, tools from the theory of block operators (see, e.g., works by Lancaster & Tismenetsky (1985); Magnus (1988); Tretter (2008)) such as the quadratic numerical range can be exploited (and combined with singular perturbation theory) to understand the effects of hyperparameters such as $\tau$ (the learning rate ratio) and regularization (which is common in applications such as generative adversarial networks) on convergence.

## B  MATHEMATICAL AND GAME THEORETIC PRELIMINARIES

In this appendix section, we review mathematical and game theoretic preliminaries needed for the technical details of the proofs. We also include some short technical lemmas from algebra that are used in the proofs.

### B.1  GAME THEORY

We now formally present the local equilibrium definitions (see Başar & Olsder 1998) that we characterize in the main body by gradient-based sufficient conditions as is typical in the literature.

**Definition B.1** (Local Nash Equilibrium). *The joint strategy $x \in X$ is a local Nash equilibrium on $\prod_{i \in \mathcal{I}} U_i \subset X$, where $U_i \subseteq X_i$, if $f(x_1, x_2) \leq f(x'_1, x_2)$, for all $x'_1 \in U_1 \subset X_1$ and $f(x_1, x_2) \geq f(x_1, x'_2)$ for all $x'_2 \in U_2 \subset X_2$. Furthermore, if the inequalities are strict, we say $x$ is a strict local Nash equilibrium.*

**Definition B.2** (Local Stackelberg Equilibrium). *Consider $U_i \subset X_i$ for $i = 1, 2$ where, without loss of generality, player 1 is the leader (minimizing player) and player 2 is the follower (maximizing player). The strategy $x_1^* \in U_1$ is a local Stackelberg solution for the leader if, $\forall x_1 \in U_1$,*

$$\sup_{x_2 \in r_{U_2}(x_1^*)} f(x_1^*, x_2) \leq \sup_{x_2 \in r_{U_2}(x_1)} f(x_1, x_2),$$

where $r_{U_2}(x_1) = \{y \in U_2 | f(x_1, y) \geq f(x_1, x_2), \forall x_2 \in U_2\}$ is the reaction curve. Moreover, for any $x_2^* \in r_{U_2}(x_1^*)$, the joint strategy profile $(x_1^*, x_2^*) \in U_1 \times U_2$ is a local Stackelberg equilibrium on $U_1 \times U_2$.

While characterizing existence of equilibria is outside the scope of this work, we remark that Nash equilibria exist for convex costs on compact and convex strategy spaces and Stackelberg equilibria exist on compact strategy spaces (Başar & Olsder, 1998, Thm. 4.3, Thm. 4.8, & Sec. 4.9). Existence of local equilibria is guaranteed if the neighborhoods and cost functions restricted to those neighborhoods satisfy the assumptions of the cited results. The differential characterization of local Nash equilibria in continuous games was first reported in (Ratliff et al., 2013). Genericity and structural stability we studied in general-sum settings in (Ratliff et al., 2014) and in zero-sum settings in (Mazumdar & Ratliff, 2019). Fiez et al. (2020); Jin et al. (2020) present sufficient conditions for a local Stackelberg equilibria and Fiez et al. (2020) studied the genericity and structural stability.

## B.2 DYNAMICAL SYSTEMS THEORY

Recall the following equivalent characterizations of stability for an equilibrium of $\dot{x} = -g(x)$ in terms of the Jacobian matrix $J(x) = Dg(x)$.

**Theorem B.1** (Theorem 4.15, Khalil 2002). *Consider a critical point $x^*$ of $g(x)$. The following are equivalent: (a) $x^*$ is a locally exponentially stable equilibrium of $\dot{x} = -g(x)$; (b) $\operatorname{spec}(-J(x^*)) \subset \mathbb{C}_-^\circ$; (c) there exists a symmetric positive-definite matrix $P = P^\top > 0$ such that $P J(x^*) + J(x^*)^\top P > 0$.*

**Leveraging Linearization to Infer Qualitative Properties.** The Hartman-Grobman theorem asserts that it is possible to continuously deform all trajectories of a nonlinear system onto trajectories of the linearization at a fixed point of the nonlinear system. Informally, the theorem states that if the linearization of the nonlinear dynamical system $\dot{x} = F(x)$ around a fixed point $\bar{x}$—i.e., $F(\bar{x}) = 0$—has no zero or purely imaginary eigenvalues, then there exists a neighborhood $U$ of $\bar{x}$ and a homeomorphism $h : U \to \mathbb{R}^n$—i.e., $h, h^{-1} \in C(U, \mathbb{R}^n)$—taking trajectories of $\dot{x} = F(x)$ and mapping them onto those of $\dot{z} = DF(\bar{x})z$. In particular, $h(\bar{x}) = 0$.

Given a dynamical system $\dot{x} = F(x)$, the state or solution of the system at time $t$ starting from $x$ at time $t_0$ is called the flow and is denoted $\phi^t(x)$.

**Theorem B.2** (Hartman-Grobman: Theorem 7.3, Sastry 1999; Theorem 9.9, Teschl 2000). *Consider the $n$-dimensional dynamical system $\dot{x} = F(x)$ with equilibrium point $\bar{x}$. If $DF(\bar{x})$ has no zero or purely imaginary eigenvalues, there is a homeomorphism $h$ defined on a neighborhood $U$ of $\bar{x}$ taking orbits of the flow $\phi^t$ to those of the linear flow $e^{tDF(\bar{x})}$ of $\dot{x} = F(x)$—that is, the flows are topologically conjugate. The homeomorphism preserves the sense of the orbits and is chosen to preserve parameterization by time.*

The above theorem says that the qualitative properties of the nonlinear system $\dot{x} = F(x)$ in the vicinity (which is determined by the neighborhood $U$) of an isolated equilibrium $\bar{x}$ are determined by its linearization if the linearization has no eigenvalues on the imaginary axes in the complex plane. We also remark that Hartman-Grobman can also be applied to discrete time maps (Sastry, 1999, Thm. 2.18) with the same qualitative outcome.

**Limiting dynamical systems and connections to singular perturbation theory.** The continuous time dynamical system takes the form $\dot{x} = -\Lambda_\tau g(x)$ due to the timescale separation $\tau$. Such a system is known as a singularly perturbed system or a multi-timescale system in the dynamical systems theory literature (Kokotovic et al., 1986), particularly where $\tau^{-1}$ is small. Singularly perturbed systems are classically expressed as

$$\begin{aligned} \dot{x} &= -D_1 f_1(x, z) \\ \epsilon \dot{z} &= -D_2 f_2(x, z) \end{aligned} \tag{8}$$

where $\epsilon = \tau^{-1}$ is most often a physically meaningful quantity inherent to some dynamical system that describes the evolution of some physical phenomena; e.g., in circuits it may be a constant related to device material properties, and in communication networks, it is often the speed at which data flows through a physical medium such as cable. This brings up to one key point of separation in

applying dynamical systems theory to the study of algorithms versus physical system dynamics: $\epsilon$ is no longer necessarily is a physical quantity but is most often a hyper-parameter subject to design.

**Internally Chain Transitivity.** In proving results for stochastic gradient descent-ascent, we leverage what is known as the *ordinary differential equation* method in which the flow of the limiting continuous time system starting at sample points from the stochastic updates of the players actions is compared to *asymptotic psuedo-trajectories*—i.e., linear interpolations between sample points. To understand stability in the stochastic case, we need the notion of internally chain transitive sets. For more detail, the reader is referred to (Alongi & Nelson, 2007, Chap. 2–3).

A closed set $U \subset \mathbb{R}^m$ is an invariant set for a differential equation $\dot{x} = F(x)$ if any trajectory $x(t)$ with $x(0) \in U$ satisfies $x(t) \in U$ for all $t \in \mathbb{R}$. Let $\phi^t$ be a flow on a metric space $(X, d)$. Given $\varepsilon > 0$, $T > 0$ and $x, y \in X$, an $(\varepsilon, T)$-chain from $x$ to $y$ with respect to $\phi^t$ and $d$ is a pair of finite sequences $x = x_0, x_1, \ldots, x_{k-1}, x_k = y$ in $X$ and $t_0, \ldots, t_{k-1}$ in $[T, \infty)$, denoted together by $(x_0, x_1, \ldots, x_{k-1}, x_k; t_0, \ldots, t_{k-1})$, such that $d(\phi^{t_i}(x_i), x_{i+1}) < \varepsilon$ for $i = 0, 1, 2, \ldots, k - 1$. A set $U \subseteq X$ is *(internally) chain transitive* with respect to $\phi^t$ if $U$ is a non-empty closed invariant set with respect to $\phi^t$ such that for each $x, y \in U$, $\epsilon > 0$ and $T > 0$ there exists an $(\varepsilon, T)$-chain from $x$ to $y$. A compact invariant set $U$ is invariantly connected if it cannot be decomposed into two disjoint closed nonempty invariant sets. It is easy to see that every internally chain transitive set is invariantly connected.

### B.3 TOOLS FOR CONVERGENCE ANALYSIS

The following proposition is a well-known result in numerical analysis and can be found in a number of books and papers on the subject. Essentially, it provides an asymptotic convergence guarantee for a discrete time update process or dynamical system.

**Proposition B.1** (Ostrowski's Theorem Argyros 1999; Theorem 10.1.2, Ortega & Rheinboldt 1970). *Let $x^*$ be a fixed point for the discrete dynamical system $x_{k+1} = F(x_k)$. If the spectral radius of the Jacobian satisfies $\rho(DF(x^*)) < 1$, then $F$ is a contraction at $x^*$ and hence, $x^*$ is asymptotically stable.*

We analyze the *iteration complexity* or *local asymptotic rate of convergence* of learning rules of the form $x_{k+1} = h(x_k)$ in the neighborhood of an equilibrium. Given two real valued functions $F(k)$ and $G(k)$, we write $F(k) = O(G(k))$ if there exists a positive constant $c > 0$ such that $|F(k)| \leq c|G(k)|$. For example, consider iterates generated by $x_{k+1} = h(x_k)$ with initial condition $x_0$ and critical point $x^*$. Then, if $F(k) = \|x_{k+1} - x^*\| \leq M^k \|x_0 - x^*\|$, we write $F(k) = O(M^k)$ where $c = \|x_0 - x^*\|$.

### B.4 NUMERICAL AND QUADRATIC NUMERICAL RANGE.

The numerical range and quadratic numerical range of a block operator matrix are particularly useful for proving results about the spectrum of a block operator matrix as they are supersets of the spectrum (Tretter, 2008). Given a matrix $A \in \mathbb{R}^{n \times n}$, the numerical range is defined by

$$\mathcal{W}(A) = \{z \in \mathbb{C}^n : \langle Az, z \rangle, \|z\| = 1\},$$

and is a convex subset of $\mathbb{C}$. Define spaces $W_i = \{z \in \mathbb{C}^{n_i} : \|z\| = 1\}$ for each $i \in \{1, 2\}$. Consider a block operator

$$A = \begin{bmatrix} A_{11} & A_{12} \\ A_{21} & A_{22} \end{bmatrix},$$

where $A_{ii} \in \mathbb{R}^{n_i \times n_i}$ and $A_{ij} \in \mathbb{R}^{n_i \times n_j}$ for each $i, j \in \{1, 2\}$. Given $v \in W_1$ and $w \in W_2$, let $A^{v,w} \in \mathbb{C}^{2 \times 2}$ be defined by

$$A^{v,w} = \begin{bmatrix} \langle A_{11}v, v \rangle & \langle A_{12}w, v \rangle \\ \langle A_{21}v, w \rangle & \langle A_{22}w, w \rangle \end{bmatrix}.$$

The quadratic numerical range of $A$ is defined by

$$\mathcal{W}^2(A) = \bigcup_{v \in W_1, w \in W_2} \text{spec}(A^{v,w})$$

where $\mathrm{spec}(\cdot)$ denotes the spectrum of its argument.

The quadratic numerical range can be described as the set of solutions of the characteristic polynomial

$$\lambda^2 - \lambda(\langle A_{11}v, v\rangle + \langle A_{22}w, w\rangle) + \langle A_{11}v, v\rangle\langle A_{22}w, w\rangle - \langle A_{12}v, w\rangle\langle A_{21}w, v\rangle = 0 \qquad (9)$$

for $v \in W_1$ and $w \in W_2$. We use the notation $\langle Av, w\rangle = \bar{v}^\top Aw$ to denote the inner product. Note that $\mathcal{W}^2(A)$ is a (potentially non-convex) subset of $\mathcal{W}(A)$ and contains $\mathrm{spec}(A)$.

## B.5 TECHNICAL LEMMAS

In this appendix, we present a handful of technical lemmas and review some additional mathematical preliminaries excluded from the main body but which are important in proving the results in the paper.

The following technical lemma is used in proving an upper bound on the spectral radius of the linearization of the discrete time update $\tau$-GDA a requirement for obtaining the convergence rate results.

**Lemma B.1.** *The function* $c(z) = (1-z)^{1/2} + \frac{z}{4} - (1 - \frac{z}{2})^{1/2}$ *satisfies* $c(x) \leq 0$ *for all* $z \in [0, 1]$.

*Proof.* Since $c(0) = 0$ and $c(1) = \frac{1}{4} - \frac{1}{\sqrt{2}} \leq 0$, we simply need to show that $c'(z) \leq 0$ on $(0, 1)$ to get that $c(z)$ is a decreasing function on $[0, 1]$, and hence negative on $[0, 1]$. Indeed, $c'(z) = \frac{1}{4} + \frac{1}{2\sqrt{4-2z}} - \frac{1}{2\sqrt{1-z}} \leq 0$ since $(1-z)^{-1/2} - (4-2z)^{-1/2} \geq 1/2$ for all $z \in (0, 1)$. $\qquad\square$

The following technical lemma, due to Mustafa & Davidson (1994), is used in constructing the finite learning rate ratio.

**Lemma B.2** (Lemma 15, Mustafa & Davidson 1994). *Let* $V, Z \in \mathbb{R}^{p\times p}$, $W \in \mathbb{R}^{p\times q}$ *and* $Y \in \mathbb{R}^{q\times q}$. *If* $V$ *and* $Y - XV^{-1}W$ *are non-singular, then*

$$\det\begin{pmatrix} V + Z & W \\ X & Y \end{pmatrix} = \det(V)\det(Y - XV^{-1}W)\det(I + V^{-1}(I + W(Y - XV^{-1}W)^{-1}XV^{-1})Z)$$

For completeness (and because there is a typo in the original manuscript), we provide the proof here.

*Proof.* Suppose that $V$ and $Y - XV^{-1}W$ are non-singular so that the partial Schur decomposition

$$\begin{bmatrix} V & W \\ X & Y \end{bmatrix} = \begin{bmatrix} V & 0 \\ X & Y - XV^{-1}W \end{bmatrix} \begin{bmatrix} I & V^{-1}W \\ 0 & I \end{bmatrix}$$

holds, and

$$\det\left(\begin{bmatrix} V & W \\ X & Y \end{bmatrix}\right) = \det(V)\det(Y - XV^{-1}W). \qquad (10)$$

Further,

$$\begin{bmatrix} V & W \\ X & Y \end{bmatrix}^{-1} = \begin{bmatrix} I & -V^{-1}W \\ 0 & I \end{bmatrix} \begin{bmatrix} V^{-1} & 0 \\ -(Y - XV^{-1}W)^{-1}XV^{-1} & (Y - XV^{-1}W)^{-1} \end{bmatrix}.$$

Applying the determinant operator, we have that

$$\det\left(\begin{bmatrix} V + Z & W \\ X & Y \end{bmatrix}\right) = \det\left(\begin{bmatrix} V & W \\ X & Y \end{bmatrix}\right)\det\left(\begin{bmatrix} I & 0 \\ 0 & I \end{bmatrix} + \begin{bmatrix} V & W \\ X & Y \end{bmatrix}^{-1}\begin{bmatrix} Z & 0 \\ 0 & 0 \end{bmatrix}\right) \qquad (11)$$

so that

$$\det\left(\begin{bmatrix} I & 0 \\ 0 & I \end{bmatrix} + \begin{bmatrix} V & W \\ X & Y \end{bmatrix}^{-1}\begin{bmatrix} Z & 0 \\ 0 & 0 \end{bmatrix}\right) = \det(V^{-1}(I + W(Y - XV^{-1}W)^{-1}XV^{-1})Z + I). \qquad (12)$$

Combining (10) with (12) in (11) gives exactly the claimed result. $\qquad\square$

The following lemma is Theorem 2 Lancaster & Tismenetsky (1985, Chap. 13.1). We use this lemma several times in the proofs of Theorem 1 and 2 so we include it here for ease of reference. For a given matrix $A$, $\upsilon_+(A)$, $\upsilon_-(A)$, and $\zeta(A)$ are the number of eigenvalues of the argument that have positive, negative and zero real parts, respectively.

**Lemma B.3.** *Consider a matrix $A \in \mathbb{R}^{n \times n}$.*

(a) *If $P$ is a symmetric matrix such that $AP + PA^\top = Q$ where $Q = Q^\top > 0$, then $P$ is nonsingular and $P$ and $A$ have the same inertia, meaning that*

$$\upsilon_+(A) = \upsilon_+(P), \ \upsilon_-(A) = \upsilon_-(P), \ \zeta(A) = \zeta(P). \tag{13}$$

(b) *On the other hand, if $\zeta(A) = 0$, then there exists a matrix $P = P^\top$ and a matrix $Q = Q^\top > 0$ such that $AP + PA^\top = Q$ and $P$ and $A$ have the same inertia ((13) holds).*

## C    PROOF OF THEOREM 1: STABILITY OF $\tau-$GDA

To prove Theorem 1 and Corollary C.1, we introduce some techniques that are arguably new to the machine learning and artificial intelligence communities. The first is the notion of a guard map. A guard map can be used to provide a certificate of a particular behavior for a dynamical system as a parameter(s) varies. A critical point of a dynamical systems is known to be stable if the spectrum of the Jacobian at the critical point lies in the open left-half complex plane, denoted $\mathbb{C}^\circ_-$. Hence, we construct a guard map as a function of $\tau$ and show that it guards $\mathbb{C}^\circ_-$. Specifically we show that the existence of a $\tau^* \in (0, \infty)$ such that $\nu(\tau^*) = 0$ and $\nu(\tau) \neq 0$ for all $\tau \in (\tau^*, \infty)$ is equivalent to $\mathsf{S}_1(J(x^*)) > 0$ and $-D_2^2 f(x^*) > 0$ where

$$\mathsf{S}_1(J(x^*)) = \mathsf{S}_1(J_\tau(x^*)) = D_1^2 f(x^*) - D_{12} f(x^*)(D_2^2 f(x^*))^{-1} D_{21} f(x^*).$$

Towards this end, we need to introduced some notation as well as formal definitions for important concepts such as the guard map.

### C.1    NOTATION AND PRELIMINARIES

Given a matrix $A \in \mathbb{R}^{n_1 \times n_2}$, let $\text{vec}(A) \in \mathbb{R}^{n_1 n_2}$ be the vectorization of $A$. We use the convention that rows are transposed and stacked in order. That is,

$$\text{vec} : \begin{bmatrix} - a_1 - \\ \vdots \\ - a_{n_1} - \end{bmatrix} \mapsto \begin{bmatrix} a_1^\top \\ \vdots \\ a_{n_1}^\top \end{bmatrix}$$

Let $\otimes$ and $\oplus$ denote the Kronecker product and Kronecker sum respectively. Recall that $A \oplus B = A \otimes I + I \otimes B$. A less common operator, we define $\boxplus$ as an operator that generates an $\frac{1}{2}n(n+1) \times \frac{1}{2}n(n+1)$ matrix from a matrix $A \in \mathbb{R}^{n \times n}$ such that

$$A \boxplus A = H_n^+ (A \oplus A) H_n$$

where $H_n^+ = (H_n^\top H_n)^{-1} H_n^\top$ is the (left) pseudo-inverse of $H_n$, a full column rank duplication matrix. A duplication matrix $H_n \in \mathbb{R}^{n^2 \times n(n+1)/2}$ is a clever linear algebra tool for mapping a $\frac{n}{2}(n+1)$ vector to a $n^2$ vector generated by applying $\text{vec}(\cdot)$ to a symmetric matrix and it is designed to respect the vectorization map $\text{vec}(\cdot)$. In particular, if $\text{vech}(X)$ is the half-vectorization map of any symmetric matrix $X \in \mathbb{R}^{n \times n}$, then $\text{vec}(X) = H_n \text{vech}(X)$ and $\text{vech}(X) = H_n^+ \text{vec}(X)$.

Given a square matrix $A$, let $\lambda_{\max}^+(A)$ be the largest positive real eigenvalue of $A$ and if $A$ does not have a positive real eigenvalue then it is zero.

**Guardian maps.**    The use of guardian maps for studying stability of parameterized families of dynamical systems was arguably introduced by Saydy et al. (1990). Guardian or guard maps act as a certificate for a performance criteria such as stability.

Formally, let $\mathcal{X}$ be the set of all $n \times n$ real matrices or the set of all polynomials of degree $n$ with real coefficients. Consider $\mathcal{S}$ an open subset of $\mathcal{X}$ with closure $\bar{\mathcal{S}}$ and boundary $\partial \mathcal{S}$.

**Definition C.1.** *The map $\nu : \mathcal{X} \to \mathbb{C}$ is said to be a guardian map for $\mathcal{S}$ if for all $x \in \bar{\mathcal{S}}$, $\nu(x) = 0 \iff x \in \partial\mathcal{S}$.*

Consider an open subset $\Omega$ of the complex plane that is symmetric with respect to the real axis. Then, elements of $\mathcal{S}(\Omega) = \{A \in \mathbb{R}^{n \times n} : \mathrm{spec}(A) \subset \Omega\}$ are said to be stable relative to $\Omega$.

The following result gives a necessary and sufficient condition for stability of parameterized families of matrices relative to some open subset of the complex plane.

**Proposition C.1** (Proposition 1 (Saydy et al., 1990); Theorem 2 (Abed et al., 1990)). *Consider $U$ to be a pathwise connected subset of $\mathbb{R}$ and $A(\tau) \in \mathbb{R}^{n \times n}$ a matrix which depends continuously on $\tau$. Let $\mathcal{S}(\Omega)$ be guarded by the map $\nu$. The family $\{A(\tau) : \tau \in U\}$ is stable relative to $\Omega$ if and only if $(i)$ it is nominally stable—meaning $A(\tau_1) \in \mathcal{S}(\Omega)$ for some $\tau_1 \in U$—and $(ii)$ $\nu(A(\tau)) \neq 0$ for all $\tau \in U$.*

In proving Theorem 1, we define a guard map for the space of $n \times n$ Hurwitz stable matrices which is denoted by $\mathcal{S}(\mathbb{C}_-^\circ)$.

**Lemma C.1.** *The map $\nu : A \mapsto \det(A \boxplus A)$ guards the set of non-singular $n \times n$ Hurwitz stable matrices $\mathcal{S}(\mathbb{C}_-^\circ)$.*

*Proof.* This follows from the following observation: for $A \in \mathbb{R}^{n \times n}$,

$$\mathrm{vech}(AX + XA^\top) = H_n^+ \mathrm{vec}(AX + XA^\top) = H_n^+(A \oplus A)\mathrm{vec}(X) = H_n^+(A \oplus A)H_n \mathrm{vech}(X)$$

from which it can be shown that the eigenvalues of $A \boxplus A$ are $\lambda_i + \lambda_j$ for $1 \leq j \leq i \leq n$ where $\lambda_i$ for $i = 1, \ldots, n$ are the eigenvalues of $A$.

Indeed, let $S$ be a non-singular matrix such that $S^{-1}AS = M$ where $M$ is upper triangular with $\lambda_1, \ldots, \lambda_n$ on its diagonal. Observe that for any $n \times n$ matrix $P$, $H_n H_n^+(P \otimes P)H_n = (P \otimes P)H_n$ and $H_n^+(P \otimes P)H_n H_n^+ = H_n^+(P \otimes P)$. Hence, using properties of the Kronecker product (namely, that $(A_1 \otimes A_2)(B_1 \otimes B_2) = (A_1 B_1 \otimes A_2 B_2)$), we have that

$$H_n^+(S^{-1} \otimes S^{-1})H_n H_n^+(I \otimes A + A \otimes I)H_n H_n^+(S \otimes S)H_n = H_n^+(I \otimes M + M \otimes I)H_n$$

so that the spectrum of $H_n^+(I \otimes A + A \otimes I)H_n$ and $H_n^+(I \otimes M + M \otimes I)H_n$ coincide. Now, since $M$ is upper triangular, $H_n^+(I \otimes M + M \otimes I)H_n$ is upper triangular with diagonal elements $\lambda_i + \lambda_j$ $(1 \leq j \leq i \leq n)$ which can be verified by direct computation and using the definition of $H_n$. This implies that $\lambda_i + \lambda_j$ $(1 \leq j \leq i \leq n)$ are exactly the eigenvalues of $H_n^+(I \otimes A + A \otimes I)H_n$. $\qquad\square$

We note that there are several other guard maps for the space of Hurwtiz stable matrices including $\nu : A \mapsto \det(A \oplus A)$. To give some intuition for this map, it is fairly straightforward to see that the Kronecker sum $A \oplus A = A \otimes I + I \otimes A$ has spectrum $\{\lambda_j + \lambda_i\}$ where $\lambda_i, \lambda_j \in \mathrm{spec}(A)$. The operator $A \boxplus A$ is simply a more computationally efficient expression of $A \oplus A$, and as such the eigenvalues of $A \boxplus A$ are those of $A \oplus A$ removing redundancies. We use $A \boxplus A$ specifically because of its computational advantages in computing $\tau^*$.

## C.2 Proof of Theorem 1

We first prove that if $x^*$ is a differential Stackelberg equilibrium (that is, $\mathsf{S}_1(J_\tau(x^*)) > 0$ and $-D_2^2 f(x^*) > 0$), then there exists a finite $\tau^* \in (0, \infty)$ such that for all $\tau \in (\tau^*, \infty)$, $x^*$ is locally exponentially stable for $\dot{x} = -\Lambda_\tau g(x)$ (that is, $\mathrm{spec}(-J_\tau(x^*)) \subset \mathbb{C}_-^\circ$). Towards this end, we construct a guard map for the space of $n \times n$ Hurwtiz stable matrices and explicitly construct the $\tau^*$ using it.

Then we prove the other direction. That is, if there exists a finite $\tau^* \in (0, \infty)$ such that for all $\tau \in (\tau^*, \infty)$, $x^*$ is exponentially stable for $\dot{x} = -\Lambda_\tau g(x)$, then $x^*$ is a differential Stackelberg equilibrium. We prove this by contradiction.

### C.2.1 Proof that if $x^*$ is a differential Stackelberg then finite $\tau^*$ exists

For a critical point $x^*$, let

$$-J_\tau(x^*) = \begin{bmatrix} -D_1^2 f(x^*) & -D_{12} f(x^*) \\ \tau D_{12}^\top f(x^*) & \tau D_2^2 f(x^*) \end{bmatrix} = \begin{bmatrix} A_{11} & A_{12} \\ -\tau A_{12}^\top & \tau A_{22} \end{bmatrix}$$

and define

$$S_1 = \mathtt{S}_1(-J_\tau(x^*)) = A_{11} - A_{12}A_{22}^{-1}A_{12}^\top.$$

Note that this is equivalent to the first Schur complement of $-J(x^*)$ (i.e., when $\tau = 1$) since the $\tau$ and $\tau^{-1}$ cancel, and by assumption the first Schur complement of $-J(x^*)$ is positive definite. Suppose that $x^*$ is a differential Stackelberg equilibrium so that $-S_1 > 0$ and $-A_{22} > 0$.

**Polynomial guard map with family of matrices parameterized by $\tau$.**  By Lemma C.1, $\nu : A \mapsto \det(A \boxplus A)$ is a guard map for $\mathcal{S}(\mathbb{C}_-^\circ)$. Indeed, using the fact that the determinant is the product of the eigenvalues of a matrix and the fact that $\mathrm{spec}(A \boxplus A) = \{\lambda_i + \lambda_j, 1 \le i \le j \le n, \lambda_i, \lambda_j \in \mathrm{spec}(A)\}$, we have that

$$\det(A \boxplus A) = \prod_{1 \le j \le i \le n}(\lambda_i + \lambda_j) = \prod_{1 \le i \le n} 2\mathrm{Re}(\lambda_i)(4\mathrm{Re}^2(\lambda_i) + 4\mathrm{Im}^2(\lambda_i)) \prod_{\substack{1 < i < j \le n: \\ \lambda_i \neq \bar{\lambda}_j}}(\lambda_i + \lambda_j).$$

Hence, consider $\bar{\mathcal{S}}(\mathbb{C}_-^\circ)$, $\det(A \boxplus A) = 0$ if and only if $A \boxplus A$ is singular if and only if $A$ has a purely imaginary eigenvalue—that is, if and only if $A \in \partial\mathcal{S}(\mathbb{C}_-^\circ)$.[5] Now, consider the parameterized family of matrices $-J_\tau(x^*)$, parameterized by $\tau$. By an abuse of notation, let $\nu(\tau) = \det(-J_\tau(x^*) \boxplus -J_\tau(x^*))$. If we consider the subset of this family of matrices that lies in $\mathcal{S}(\mathbb{C}_-^\circ)$ (this subset could a priori be empty thought we show it is not), then for any $\tau$ such that $-J_\tau(x^*)$ is in this subset, we have that $\nu(\tau) = 0$ if and only if $-J_\tau(x^*) \boxplus (-J_\tau(x^*))$ is singular if and only if $-J_\tau(x^*) \in \partial\mathcal{S}(\mathbb{C}_-^\circ)$. Hence, $\nu(\tau) = \det(-J_\tau(x^*) \boxplus -J_\tau(x^*))$ guards $\mathcal{S}(\mathbb{C}_-^\circ)$.

In particular, if we envision $-J_\tau(x^*)$ as the input to $\nu : A \mapsto \det(A \boxplus A)$ and simply vary $\tau$ (holding all the entries of $-J_\tau(x^*)$ otherwise fixed), then $\nu : \tau \mapsto \det(-J_\tau(x^*) \boxplus (-J_\tau(x^*)))$ can be thought of simply as a function of $\tau$ which guards the set of Hurwitz stable matrices via the reasoning describe above. Indeed, by slightly overloading the notation for $\nu$,

$$\nu(\tau) := \nu_0 + \nu_1\tau + \cdots + \nu_{p-1}\tau^{p-1} + \nu_p\tau^p = \nu(-J_\tau(x^*))$$

Hence, for intuition, observe that as $\tau$ decreases (towards zero) stability is first lost when at least one eigenvalue of $-J_\tau(x^*)$ reaches the imaginary axis, at which point $\nu(\tau) = 0$.

There are two cases to consider:

**Case 1:** $\nu(\tau)$ *is an identically zero polynomial.* In this case, $-J_\tau(x^*)$ is in the interior of the complement of the set of Hurwitz stable matrices for all values of $\tau > 0$—that is, $-J_\tau(x^*) \in \mathrm{int}(\mathcal{S}^c(\mathbb{C}_-^\circ))$ for all $\tau \in \mathbb{R}_+ = (0, \infty)$.

**Case 2:** $\nu(\tau)$ *is not an identically zero polynomial.* In this case, $\nu(\tau)$ has finitely many zeros. If $\nu(\tau)$ has no positive real roots, then as $\tau$ varies in $\mathbb{R}_+$, $-J_\tau(x^*)$ does not cross $\partial\mathcal{S}(\mathbb{C}_-^\circ$— i.e., the boundary of the space of $n \times n$ Hurwitz stable matrices. Hence, $\{-J_\tau(x^*) : \tau \in \mathbb{R}_+\} \subset \mathcal{S}^c(\mathbb{C}_-^\circ)$ or $\{-J_\tau(x^*) : \tau \in \mathbb{R}_+\} \subset \mathrm{int}(\mathcal{S}^c(\mathbb{C}_-^\circ))$. It suffices to check $-J_\tau(x^*) \in \mathcal{S}^c(\mathbb{C}_-^\circ)$ or $-J_\tau(x^*) \in \mathrm{int}(\mathcal{S}^c(\mathbb{C}_-^\circ))$ for an arbitrary $\tau \in \mathbb{R}_+$.

On the other hand, if $\nu(\tau)$ has $\ell \ge 1$ real positive zeros, say $0 < \tau_1 < \cdots < \tau_\ell = \tau^*$, then by Proposition C.1, $-J_\tau(x^*) \in \mathcal{S}(\mathbb{C}_-^\circ)$ for all $\tau > \tau^*$ if and only if $-J_\tau(x^*) \in \mathcal{S}(\mathbb{C}_-^\circ)$ for arbitrarily chosen $\tau > \tau^*$. We choose the largest positive root $\tau_\ell$ because we are guaranteed that $\nu(\tau)$ stops changing sign for $\tau > \tau^*$. Further, the largest neighborhood in $\mathbb{R}_+$ for which $-J_\tau(x^*) \in \mathcal{S}(\mathbb{C}_-^\circ)$ is $(\tau_\ell, \infty)$.

Recall that we have assumed that $x^*$ is a differential Stackelberg equilibrium (that is, $S_1 > 0$ and $-A_{22} > 0$). We will show next (by way of explicit construction of $\tau^*$) that we are always in case 2.

**Construction of $\tau^*$.**  We note that there are more elegant, simpler constructions, but to our knowledge this construction gives the tightest bound on the range of $\tau$ for which $-J_\tau(x^*)$ is guarnateed to be Hurwitz stable. Recall that

$$-J_\tau(x^*) = \begin{bmatrix} -D_1^2 f(x^*) & -D_{12}f(x^*) \\ \tau D_{12}^\top f(x^*) & \tau D_2^2 f(x^*) \end{bmatrix} = \begin{bmatrix} A_{11} & A_{12} \\ -\tau A_{12}^\top & \tau A_{22} \end{bmatrix}$$

---

[5]Indeed, this holds since the only scenarios in which $\det(A \boxplus A) = 0$ are such that the eigenvalues of $A$ do not lie in $\bar{\mathcal{S}}(\mathbb{C}_-^\circ)$.

and

$$S_1 = A_{11} - A_{12} A_{22}^{-1} A_{12}^\top.$$

Let $I_m$ denote the $m \times m$ identity matrix for some $m$.

**Claim C.1.** *The finite learning rate ratio is* $\tau^* = \lambda_{\max}^+(Q)$ *where*

$$Q = 2 \left[ (A_{12} \otimes A_{22}^{-1}) H_{n_2} \quad (I_{n_1} \otimes A_{22}^{-1} A_{12}^\top) H_{n_1} \right] \begin{bmatrix} \bar{A}_{22}^{-1} H_{n_2}^+ (A_{12}^\top \otimes I_{n_2}) \\ -\bar{S}_1^{-1} H_{n_1}^+ (S_1 \otimes A_{12} A_{22}^{-1}) \end{bmatrix} - (A_{11} \otimes A_{22}^{-1}) \tag{14}$$

*with* $\bar{A}_{22} = A_{22} \boxplus A_{22}$ *and* $\bar{S}_1 = S_1 \boxplus S_1$.

*Proof.* Recall that $\nu(\tau) = \det(-J_\tau(x^*) \boxplus (-J_\tau(x^*)))$ is a guard map for $\mathcal{S}(\mathbb{C}_-^\circ)$.

We apply basic properties of the Kronecker product and sum as well as Schur's determinant formula to obtain a reduced form of the guard map. To this end, we have that

$$-J_\tau(x^*) \boxplus (-J_\tau(x^*)) = \begin{bmatrix} A_{11} \boxplus A_{11} & 2 H_{n_1}^+ (I_{n_1} \otimes A_{12}) & 0 \\ \tau(I_{n_1} \otimes (-A_{12}^\top)) H_{n_1} & A_{11} \oplus \tau A_{22} & (A_{12} \otimes I_{n_2}) H_{n_2} \\ 0 & 2\tau H_{n_2}^+ (-A_{12}^\top \otimes I_{n_2}) & \tau(A_{22} \boxplus A_{22}) \end{bmatrix}$$

Now, we apply Schur's determinant formula to get that

$$\nu(\tau) = \tau^{n_2(n_2+1)/2} \det(A_{22} \boxplus A_{22}) \det \left( \begin{bmatrix} A_{11} \boxplus A_{11} & 2 H_{n_1}^+ (I_{n_1} \otimes A_{12}) \\ \tau(I_{n_1} \otimes (-A_{12}^\top)) H_{n_1} & A_{11} \oplus \tau A_{22} + M_1 \end{bmatrix} \right) \tag{15}$$

where

$$M_1 = -2 H_{n_2}^+ (-A_{12}^\top \otimes I_{n_2})(A_{22} \boxplus A_{22})^{-1}(A_{12} \otimes I_{n_2}) H_{n_2}$$

From here, we apply Lemma B.2 to further reduce the guard map. First, note that

$$A_{11} \oplus \tau A_{22} = A_{11} \otimes I_{n_2} + I_{n_1} \otimes \tau A_{22}.$$

Let $V = I_{n_1} \otimes \tau A_{22}$, $Z = A_{11} \otimes I_{n_2} + M_1$, $Y = A_{11} \boxplus A_{11}$, $W = -\tau(I_{n_1} \otimes A_{12}^\top) H_{n_1}$, and $X = 2 H_{n_1}^+ (I_{n_1} \otimes A_{12})$. Using the two properties of the Kronecker product $(B_1 \otimes B_2)(B_3 \otimes B_4) = (B_1 B_3 \otimes B_2 B_4)$ and $(B_1 \otimes B_2)^{-1} = (B_1^{-1} \otimes B_2^{-1})$, we have that

$$Y - X V^{-1} W = A_{11} \boxplus A_{11} + 2 H_{n_1}^+ (I_{n_1} \otimes A_{12})(I_{n_1} \otimes A_{22})^{-1}(I_{n_1} \otimes A_{12}^\top) H_{n_1} \tag{16}$$

$$= A_{11} \boxplus A_{11} + 2 H_{n_1}^+ (I_{n_1} \otimes A_{12} A_{22}^{-1} A_{12}^\top) H_{n_1} \tag{17}$$

$$= A_{11} \boxplus A_{11} + H_{n_1}^+ ((I_{n_1} \otimes A_{12} A_{22}^{-1} A_{12}^\top) + (A_{12} A_{22}^{-1} A_{12}^\top \otimes I_{n_1})) H_{n_1} \tag{18}$$

$$= S_1 \boxplus S_1 \tag{19}$$

where (18) holds since $H_{n_1}^+ (I_{n_1} \otimes A_{12} A_{22}^{-1} A_{12}^\top) H_{n_1} = H_{n_1}^+ (A_{12} A_{22}^{-1} A_{12}^\top \otimes I_{n_1}) H_{n_1}$. Now, define $V^{-1} + V^{-1} W (Y - X V^{-1} W)^{-1} X V^{-1} = \tau^{-1} M_2$ where

$$M_2 = I_{n_1} \otimes A_{22}^{-1} - 2(I_{n_1} \otimes A_{22}^{-1} A_{12}^\top) H_{n_1} (S_1 \boxplus S_1)^{-1} H_{n_1}^+ (I_{n_1} \otimes A_{12} A_{22}^{-1})$$

so that applying Lemma B.2 we have

$$\nu(\tau) = \tau^{n_2(n_2+1)/2} \det(A_{22} \boxplus A_{22}) \det(S_1 \boxplus S_1) \det(I_{n_1} \otimes A_{22}) \det(\tau I_{n_1 n_2} + M_2(A_{11} \otimes I_{n_2} + M_1)) \tag{20}$$

The assumptions that $S_1 > 0$ and $-A_{22} > 0$ together imply that $\det(S_1 \boxplus S_1) \neq 0$ and $\det(I_{n_1} \otimes A_{22}) \neq 0$. Hence, $\nu(\tau) = 0$ if and only if $\det(\tau I_{n_1 n_2} + M_2(A_{11} \otimes I_{n_2} + M_1)) = 0$ since $0 < \tau < \infty$. The determinant expression is exactly an eigenvalue problem.

Since by assumption the Schur complement of $J(x^*)$ and the individual Hessian $-D_2^2 f(x^*)$ are positive definite (that is, $x^*$ is a differential Stackelberg equilibrium), Thus, the largest positive real root of $\nu(\tau) = 0$ is

$$\tau^* = \lambda_{\max}^+(-M_2(A_{11} \otimes I_{n_2} + M_1))$$

where $\lambda_{\max}^+(\cdot)$ is the largest positive real eigenvalue of its argument if one exists and otherwise its zero. Using properties of the Kronecker product and duplication matrices, it can easily be seen that $Q$, as defined in (14), is equivalent to $-M_2(A_{11} \otimes I_{n_2} + M_1)$. $\square$

The result of this claim concludes the proof that if $x^*$ is a differential Stackelberg, then there exists a finite $\tau^* \in [0, \infty)$ such that for all $\tau \in (\tau^*, \infty)$, $\mathrm{spec}(-J_\tau(x^*)) \subset \mathbb{C}_-^\circ$.

### C.2.2 PROOF THAT EXISTENCE OF FINITE $\tau^*$ IMPLIES THAT $x^*$ IS A DIFFERENTIAL STACKELBERG

To begin, consider a critical point $x^*$ such that $g(x^*) = 0$ and $\det(D_2^2 f(x^*)) \neq 0$. Then, $\det(-J_\tau(x^*)) = \tau^{n_2} \det(D_2^2 f(x^*)) \det(-\mathtt{S}_1(J(x^*)))$ so that $\det(-J_\tau(x^*)) = 0$ if and only if $\det(-\mathtt{S}_1(J(x^*))) = 0$ which implies $-J_\tau(x^*)$ is unstable for all $\tau \in (0, \infty)$ when $\det(-\mathtt{S}_1(J(x^*))) = 0$. As a result, we are left to consider when $\det(\mathtt{S}_1(J(x^*))) \neq 0$ for the remainder of the proof.

We proceed by arguing a contradiction. Let $-C \equiv -D_2^2 f(x^*)$ and $S_1 \equiv \mathtt{S}_1(J(x^*)) = D_1^2 f(x^*) - D_{12} f(x^*)(D_2^2 f(x^*))^{-1} D_{12}^\top f(x^*)$ have no zero eigenvalues—that is, $\det(S_1) \neq 0$ and $\det(C) \neq 0$.

Suppose that there exists a $\tau^* \in (0, \infty)$ such that for all $\tau \in (\tau^*, \infty)$, $\text{spec}(-J_\tau(x^*)) \subset \mathbb{C}_-^\circ$, yet $x^*$ is not a differential Stackelberg equilibrium. That is, either $-S_1$ or $C$ have at least one positive eigenvalue. Without loss of generality, let $-S_1$ have at least one positive eigenvalue.

Since $\det(S_1) \neq 0$ and $\det(C) \neq 0$, by Lemma B.3.b, there exists non-singular Hermitian matrices $P_1, P_2$ and positive definite Hermitian matrices $Q_1, Q_2$ such that $-S_1 P_1 - P_1 S_1 = Q_1$ and $C P_2 + P_2 C = Q_2$. Further, $-S_1$ and $P_1$ have the same inertia, meaning

$$\upsilon_+(-S_1) = \upsilon_+(P_1), \ \upsilon_-(-S_1) = \upsilon_-(P_1), \ \zeta(-S_1) = \zeta(P_1)$$

where for a given matrix $A$, $\upsilon_+(A)$, $\upsilon_-(A)$, and $\zeta(A)$ are the number of eigenvalues of the argument that have positive, negative and zero real parts, respectively. Similarly, $C$ and $P_2$ have the same inertia:

$$\upsilon_+(C) = \upsilon_+(P_2), \ \upsilon_-(C) = \upsilon_-(P_2), \ \zeta(C) = \zeta(P_2).$$

Since $-S_1$ has at least one strictly positive eigenvalue, $\upsilon_+(P_1) = \upsilon_+(-S_1) \geq 1$.

Define

$$P = \begin{bmatrix} I & L_0^\top \\ 0 & I \end{bmatrix} \begin{bmatrix} P_1 & 0 \\ 0 & P_2 \end{bmatrix} \begin{bmatrix} I & 0 \\ L_0 & I \end{bmatrix} \tag{21}$$

where $L_0 = (D_2^2 f(x^*))^{-1} D_{12}^\top f(x^*) = C D_{12}^\top f(x^*)$. Since $P$ is congruent to $\text{blockdiag}(P_1, P_2)$, by Sylvester's law of inertia (Horn & Johnson, 1985, Thm. 4.5.8), $P$ and $\text{blockdiag}(P_1, P_2)$ have the same inertia, meaning that $\upsilon_+(P) = \upsilon_+(\text{blockdiag}(P_1, P_2))$, $\upsilon_-(P) = \upsilon_-(\text{blockdiag}(P_1, P_2))$, and $\zeta(P) = \zeta(\text{blockdiag}(P_1, P_2))$. Consider the matrix equation $-P J_\tau(x^*) - J_\tau^\top(x^*) P = Q_\tau$ for $-J_\tau(x^*)$ where

$$Q_\tau = \begin{bmatrix} I & L_0^\top \\ 0 & I \end{bmatrix} B_\tau \begin{bmatrix} I & 0 \\ L_0 & I \end{bmatrix}$$

with

$$B_\tau = \begin{bmatrix} Q_1 & P_1 D_{12} f(x^*) - S_1 L_0^\top P_2 \\ (P_1 D_{12} f(x^*) - S_1 L_0^\top P_2)^\top & P_2 L_0 D_{12} f(x^*) + (P_2 L_0 D_{12} f(x^*))^\top + \tau Q_2 \end{bmatrix}$$

which can be verified by straightforward calculations.

Observe that $Q_\tau > 0$ is equivalent to $B_\tau > 0$ and both matrices are symmetric so that $B_\tau > 0$ if and only if $Q_1 > 0$ and $\mathtt{S}_2(B_\tau) > 0$ where

$$\mathtt{S}_2(B_\tau) = P_2 L_0 D_{12} f(x^*) + (P_2 L_0 D_{12} f(x^*))^\top + \tau Q_2$$
$$- (P_1 D_{12} f(x^*) - S_1 L_0^\top P_2)^\top Q_1^{-1} (P_1 D_{12} f(x^*) - S_1 L_0^\top P_2).$$

Now, $\mathtt{S}_2(B_\tau)$ is also a real symmetric matrix, and hence, it is positive definite if and only if all its eigenvalues are positive. To determine the range of $\tau$ such that $\mathtt{S}_2(B_\tau)$ is positive definite, we can formulate an eigenvalue problem to determine the value of $\tau$ such that the matrix $\mathtt{S}_2(B_\tau)$ becomes singular. This is analogous to the guard map approach used in the proof in the previous subsection for the other direction of the proof, and in this case, we are varying $\tau$ from zero to infinity and finding the point such that for all larger $\tau$, $\mathtt{S}_2(B_\tau)$ is positive definite. Intuitively, such an argument works since $\tau$ scales the positive definite matrix $Q_2$. Towards this end, consider the eigenvalue problem in $\tau$ given by

$$0 = \det \Big( \tau I - Q_2^{-1} \big( (P_1 D_{12} f(x^*) - S_1 L_0^\top P_2)^\top Q_1^{-1} (P_1 D_{12} f(x^*) - S_1 L_0^\top P_2)$$
$$- P_2 L_0 D_{12} f(x^*) - (P_2 L_1 D_{12} f(x^*))^\top \big) \Big).$$

Let $\tau_0$ be the maximum positive eigenvalue, and zero otherwise. Then, since eigenvalues vary continuously, for all $\tau \in (\tau_0, \infty)$, $Q_\tau > 0$ so that by Lemma B.3.a we conclude that $P$ and $-J_\tau(x^*)$ have the same inertia, but this contradicts the stability of $-J_\tau(x^*)$ for all $\tau \in (\tau^*, \infty)$ since $\upsilon_+(P) \geq 1$.

**Remark on the tightness of $\tau^*$.** The construction of $\tau^*$ is *tight* in the following sense. While it is possible to construct multiple guard maps for a domain, all guard maps have the same positive real roots by definition (Saydy, 1996, Remark 2). Hence, independent of the guard map choice, we will get the same value of $\tau^*$. Moreover, $\tau^*$ tells us exactly when the eigenvalues move into the open left-half complex plane $\mathbb{C}^\circ_-$ and remain there. Hence, this gives us the precise, tight lower bound on the value of $\tau^*$.

## C.3 $\tau$-GDA PROVABLE CONVERGES TO DIFFERENTIAL STACKELEBRG EQUILIBRIA

As a corollary to Theorem 1, we first show that the discrete time $\tau$-GDA update is locally asymptotically stable for a range of learning rates $\gamma_1$.

**Corollary C.1** (Asymptotic convergence of $\tau$-GDA). *Suppose the assumptions of Theorem 1 hold so that $x^*$ is a critical points of $g$ and $\mathtt{S}_1(J(x^*))$ and $D_2^2 f_2(x^*)$ are non-singular. There exists a $\tau^* \in (0, \infty)$ such that $\tau$-GDA with $\gamma_1 \in (0, \gamma(\tau))$ where $\gamma(\tau) = \arg\min_{\lambda \in \mathrm{spec}(J_\tau(x^*))} 2\mathrm{Re}(\lambda)/|\lambda|^2$ converges locally asymptotically for all $\tau \in (\tau^*, \infty)$ if and only if $x^*$ is a differential Stackelberg equilibrium.*

The proceeding corollary to Theorem 1 follows immediately from Lemma F.1 given in Appendix F.

*Proof.* Suppose that $x^*$ is a differential Stackelberg equilibrium so that by Theorem 1, there exists a $\tau^* \in (0, \infty)$ such that $\mathrm{spec}(-J_\tau(x^*)) \subset \mathbb{C}^\circ_-$ for all $\tau \in (\tau^*, \infty)$. Now that we have a guarantee that $-J_\tau(x^*)$ is Hurwitz stable for any $\tau \in (\tau^*, \infty)$, we apply Hartman-Grobman to get that the nonlinear system $\dot{x} = -\Lambda_\tau g(x)$ is stable in a neighborhood of $x^*$. Fix any $\tau \in (\tau^*, \infty)$ and let $\gamma = \arg\min_{\lambda \in \mathrm{spec}(J_\tau(x^*))} 2\mathrm{Re}(\lambda)/|\lambda|^2$. Then, applying Lemma F.1, for any $\gamma_1 \in (0, \gamma)$, $\tau$-GDA converges locally asymptotically to $x^*$.

On the other hand, suppose that there exists a $\tau^* \in (0, \infty)$ such that $\mathrm{spec}(-J_\tau(x^*)) \subset \mathbb{C}^\circ_-$ for all $\tau \in (\tau^*, \infty)$. Then by Theorem 1, if $x^*$ is a differential Stackelberg equilibrium. Furthermore, since $\mathrm{spec}(-J_\tau(x^*)) \subset \mathbb{C}^\circ_-$ for all $\tau \in (\tau^*, \infty)$, if we let $\gamma = \arg\min_{\lambda \in \mathrm{spec}(J_\tau(x^*))} 2\mathrm{Re}(\lambda)/|\lambda|^2$, then by Lemma F.1 $\tau$-GDA converges locally asymptotically to $x^*$ for any choice of $\gamma_1 \in (0, \gamma)$. $\qquad\square$

## D PROOF OF THEOREM 2: INSTABILITY OF $\tau$−GDA

To begin, consider a critical point $x^*$ defined by $g(x^*) = 0$ which is not a differential Stackelberg equilibria such that $\det(D_2^2 f(x^*)) \neq 0$. Then, $\det(-J_\tau(x^*)) = \tau^{n_2} \det(D_2^2 f(x^*)) \det(-\mathtt{S}_1(J(x^*)))$ so that $\det(-J_\tau(x^*)) = 0$ if and only if $\det(-\mathtt{S}_1(J(x^*))) = 0$ which implies $-J_\tau(x^*)$ is unstable for all $\tau \in (\tau_0, \infty)$ where $\tau_0 = 0$ when $\det(-\mathtt{S}_1(J(x^*))) = 0$. We now consider when $\det(\mathtt{S}_1(J(x^*))) \neq 0$ for the remainder of the proof.

Let $x^*$ be a stable critical point of $1$-GDA (without loss of generality) which is not a differential Stackelberg equilibrium. Without loss of generality, suppose that $\mathtt{S}_1(-J(x^*))$ has at least one strictly positive eigenvalue. Note that both $\mathtt{S}_1(-J(x^*))$ and $-D_2^2 f(x^*)$ are symmetric matrices and hence, have purely real eigenvalues.

Since both $\mathtt{S}_1(-J(x^*))$ and $D_2^2 f(x^*)$ have no zero valued eigenvalues, by Lemma B.3.b, there exists non-singular Hermitian matrices $P_1, P_2$ and positive definite Hermitian matrices $Q_1, Q_2$ such that $\mathtt{S}_1(-J(x^*))P_1 + P_1\mathtt{S}_1(-J(x^*)) = Q_1$ and $D_2^2 f(x^*)P_2 + P_2 D_2^2 f(x^*) = Q_2$. Further, $\mathtt{S}_1(-J(x^*))$ and $P_1$ have the same inertia, meaning

$$\upsilon_+(\mathtt{S}_1(-J(x^*))) = \upsilon_+(P_1), \ \upsilon_-(\mathtt{S}_1(-J(x^*))) = \upsilon_-(P_1), \ \zeta(\mathtt{S}_1(-J(x^*))) = \zeta(P_1)$$

where for a given matrix $A$, $\upsilon_+(A)$, $\upsilon_-(A)$, and $\zeta(A)$ are the number of eigenvalues of the argument that have positive, negative and zero real parts, respectively. Similarly, $D_2^2 f(x^*)$ and $P_2$ have the same inertia:

$$\upsilon_+(D_2^2 f(x^*)) = \upsilon_+(P_2), \ \upsilon_-(D_2^2 f(x^*)) = \upsilon_-(P_2), \ \zeta(D_2^2 f(x^*)) = \zeta(P_2).$$

Recall that we assumed $\mathtt{S}_1(-J(x^*))$ has at least one eigenvalue with strictly positive real part. Hence, $\upsilon_+(P_1) = \upsilon_+(\mathtt{S}_1(-J(x^*))) \geq 1$.

Define

$$P = \begin{bmatrix} I & L_0^\top \\ 0 & I \end{bmatrix} \begin{bmatrix} P_1 & 0 \\ 0 & P_2 \end{bmatrix} \begin{bmatrix} I & 0 \\ L_0 & I \end{bmatrix}$$

where $L_0 = (D_2^2 f(x^*))^{-1} D_{12}^\top f(x^*)$. Since $P$ is congruent to $\mathrm{blockdiag}(P_1, P_2)$, by Sylvester's law of inertia (Horn & Johnson, 1985, Thm. 4.5.8), $P$ and $\mathrm{blockdiag}(P_1, P_2)$ have the same inertia, meaning that $\upsilon_+(P) = \upsilon_+(\mathrm{blockdiag}(P_1, P_2))$, $\upsilon_-(P) = \upsilon_-(\mathrm{blockdiag}(P_1, P_2))$, and $\zeta(P) = \zeta(\mathrm{blockdiag}(P_1, P_2))$. Consider now the Lyapunov equation $-PJ_\tau(x^*) - J_\tau^\top(x^*)P = Q_\tau$ for $-J_\tau(x^*)$ where

$$Q_\tau = \begin{bmatrix} I & L_0^\top \\ 0 & I \end{bmatrix} B_\tau \begin{bmatrix} I & 0 \\ L_0 & I \end{bmatrix}$$

with

$$B_\tau = \begin{bmatrix} Q_1 & P_1 D_{12} f(x^*) + \mathtt{S}_1(-J(x^*))L_0^\top P_2 \\ (P_1 D_{12} f(x^*) + \mathtt{S}_1(-J(x^*))L_0^\top P_2)^\top & P_2 L_0 D_{12} f(x^*) + (P_2 L_0 D_{12} f(x^*))^\top + \tau Q_2 \end{bmatrix}$$

which can be verified by straightforward calculations.

Since $\upsilon_+(P_1) \geq 1$, we have that $\upsilon_+(P) \geq 1$. Now, we find the value of $\tau_0$ such that for all $\tau > \tau_0$, $Q_\tau > 0$ so that, in turn, we can apply Lemma B.3.a, to conclude that $\mathrm{spec}(-J_\tau(x^*)) \not\subset \mathbb{C}_-^\circ$. Indeed, observe that $Q_\tau > 0$ is equivalent to $B_\tau > 0$ and both matrices are symmetric so that $B_\tau > 0$ if and only if $Q_1 > 0$ and $\mathtt{S}_2(B_\tau) > 0$ where

$$\mathtt{S}_2(B_\tau) = P_2 L_1 D_{12} f(x^*) + (P_2 L_1 D_{12} f(x^*))^\top + \tau Q_2$$
$$- (P_1 D_{12} f(x^*) + \mathtt{S}_1(-J(x^*))L_0^\top P_2)^\top Q_1^{-1}(P_1 D_{12} f(x^*) + \mathtt{S}_1(-J(x^*))L_0^\top P_2).$$

Now, $\mathtt{S}_2(B_\tau)$ is also a real symmetric matrix, and hence, it is positive definite if and only if all its eigenvalues are positive. To determine the range of $\tau$ for which $Q_\tau > 0$, we simply need to solve the eigenvalue problem

$$0 = \det(\tau I - Q_2^{-1}((P_1 D_{12} f(x^*) + \mathtt{S}_1(-J(x^*))L_0^\top P_2)^\top Q_1^{-1}(P_1 D_{12} f(x^*) + \mathtt{S}_1(-J(x^*))L_0^\top P_2)$$
$$- P_2 L_0 D_{12} f(x^*) - (P_2 L_0 D_{12} f(x^*))^\top)).$$

and extract the maximum eigenvalue, namely,

$$\tau_0 = \lambda_{\max}(Q_2^{-1}((P_1 D_{12} f(x^*) + \mathtt{S}_1(-J(x^*))L_0^\top P_2)^\top Q_1^{-1}(P_1 D_{12} f(x^*)$$
$$+ \mathtt{S}_1(-J(x^*))L_0^\top P_2) - P_2 L_0 D_{12} f(x^*) - (P_2 L_0 D_{12} f(x^*))^\top)).$$

Hence, as noted previously, by Lemma B.3.a, we conclude that for all $\tau \in (\tau_0, \infty)$, $\mathrm{spec}(-J_\tau(x^*)) \not\subset \mathbb{C}_-^\circ$. This concludes the proof.

**Additional context/intuition for the proof approach.** To provide some context for the proof approach, we remark that it follows the same idea as the proof of Theorem 1 in Appendix C.2.2. Indeed, to determine the range of $\tau$ such that $\mathtt{S}_2(B_\tau)$ is positive definite, we can formulate an eigenvalue problem to determine the value of $\tau$ such that the matrix $\mathtt{S}_2(B_\tau)$ becomes singular. We vary $\tau$ from zero to infinity in order to find the point such that for all larger $\tau$, $\mathtt{S}_2(B_\tau)$ is positive definite. Intuitively, such an argument works since $\tau$ scales the positive definite matrix $Q_2$.

## E    PROOF OF THEOREM 3: STABILITY OF $\tau$–GDA IN REGULARIZED GANS

As in Mescheder et al. (2018), we only apply the regularization to the discriminator. In the following proof, we use $\nabla_x(\cdot)$ to denote the partial gradient with respect to $x$ of the argument $(\cdot)$ when the argument is the discriminator $\mathrm{D}(\cdot; \omega)$ in order prevent any confusion between the notation $D(\cdot)$ which we use elsewhere for derivatives.

To prove the first part of this result, we following similar arguments to Theorem 4.1 of (Mescheder et al., 2018). To prove the second part, we leverage the concept of the quadratic numerical range.

For both components of the proof, we will use the following form of the Jacobian of the regularized game. Indeed, first observe that the structural form of $J_{(\tau,\mu)}(x^*)$ is

$$J_{(\tau,\mu)}(x^*) = \begin{bmatrix} 0 & B \\ -\tau B^\top & \tau(C + \mu R) \end{bmatrix} \tag{22}$$

where $B = D_{12}f(x^*)$, $C = -D_2^2 f(x^*)$ and $R = D_2^2 R_j(x^*)$ where $R_j$ is either gradient penalty indexed by $j = 1, 2$.[6] This follows from Assumption 1-a., which implies that $\mathrm{D}(x; \omega^*) = 0$ in some neighborhood of $\mathrm{supp}(p_\mathcal{D})$ and hence, $\nabla_x \mathrm{D}(x; \omega^*) = 0$ and $\nabla_x^2 \mathrm{D}(x; \omega^*) = 0$ for $x \in \mathrm{supp}(p_\mathcal{D})$. In turn, we have that $D_1^2 f(x^*) = 0$.

**Proof that $x^* = (\theta^*, \omega^*)$ is a differential Stackelberg equilibrium.** For any fixed $\mu \in (0, \infty)$, then we first observe that $x^*$ is also a critical point of the unregularized dynamics. Indeed, by Assumption 1-a., $\mathrm{D}(x; \omega^*) = 0$ in some neighborhood of $\mathrm{supp}(p_\mathcal{D})$ and hence, $\nabla_x \mathrm{D}(x; \omega^*) = 0$ and $\nabla_x^2 \mathrm{D}(x; \omega^*) = 0$ for $x \in \mathrm{supp}(p_\mathcal{D})$. Further, $D_2 R_j(\theta, \omega) = \mu \mathbb{E}_{p_i(x)}[D_2(\nabla_x \mathrm{D}(x; \omega))\nabla_x \mathrm{D}(x, \omega)]$ for $j = 1, 2$ where $p_1(x) = p_\mathcal{D}(x)$ and $p_2(x) = p_\theta(x)$. Thus, using the above observation that $\nabla_x \mathrm{D}(x; \omega^*) = 0$, we have that $D_2 R_i(\theta^*, \omega^*) = 0$ for $i = 1, 2$ meaning that the derivative of the regularizer with respect to $\omega$ is zero at $x^* = (\theta^*, \omega^*)$ which in turn implies that $D_1 f(x^*) = 0$ and $-D_2 f(x^*) = 0$. Hence, $x^*$ is a critical point of the unregularized dynamics as claimed. Further, $C + \mu R > 0$ which follows from Lemma D.5 in (Mescheder et al., 2018). From Lemma D.6 in (Mescheder et al., 2018), due to Assumption 1-c., if $v \neq 0$ and $v \notin T_{\theta^*} \mathcal{M}_\mathrm{G}$, then $Bv \neq 0$ which implies that $B$ can only be rank deficient on $T_{\theta^*} \mathcal{M}_\mathrm{G}$. Using this fact along with the structure of the Jacobian as in (22), we have that the Schur complement of $J_{(\tau,\mu)}(x^*)$ is equal to $B^\top (C + \mu R)^{-1} B > 0$ since $C + \mu R > 0$. Hence, $x^* = (\theta^*, \omega^*)$ is a differential Stackelberg equilibrium.

**Proof of stability.** Examining (22), it is straightforward to see that the quadratic numerical range $\mathcal{W}^2(J_{(\tau,\mu)})$ has eigenvalues of the form

$$\lambda_{\tau,\mu} = \tfrac{1}{2}(\tau(c + \mu r)) \pm \tfrac{1}{2}\sqrt{(-\tau(c + \mu r))^2 - 4\tau|b|^2}$$

where $b = \langle D_{12}f(x^*)v, w \rangle$, $c = \langle -D_2^2 f(x^*)w, w \rangle$ and $r = \langle D_2^2 R_i(x^*)w, w \rangle$ for vectors $v \in W_1 \cap (T_{\theta^*}\mathcal{M}_\mathrm{G})^\perp$ and $w \in W_2 \cap (T_{\omega^*}\mathcal{M}_\mathrm{D})^\perp$ where $U^\perp$ denotes the orthogonal complement of $U$. We claim that for any value of $\mu \in (0, \infty)$ and any $\tau \in (0, \infty)$, $\mathrm{Re}(\lambda_{\tau,\mu}) > 0$. Indeed, we argue this by considering the two possible cases: (1) $(\tau(c + \mu r))^2 \leq 4|b|^2\tau$ or (2) $(\tau(c + \mu r))^2 > 4\tau|b|^2$.

- **Case 1**: Suppose that $(\tau(c + \mu r))^2 \leq 4|b|^2\tau$. Then, $\mathrm{Re}(\lambda_{\tau,\mu}) = \tfrac{1}{2}(\tau(c + \mu r)) > 0$ trivially since $c + \mu r > 0$.

- **Case 2**: Suppose that $(\tau(c + \mu r))^2 > 4\tau|b|^2$. In this case, we want to ensure that

$$\mathrm{Re}(\lambda_\tau) > \tfrac{1}{2}(\tau(c + \mu r)) - \tfrac{1}{2}\sqrt{(-\tau(c + \mu r))^2 - 4\tau|b|^2} > 0.$$

  which holds since

$$(\tau(c + \mu r))^2 > (-\tau(c + \mu r))^2 - 4\tau|b|^2 \iff 0 > -4\tau|b|^2$$

This concludes the proof.

## E.1 NECESSARY CONDITIONS ON GAN ARCHITECTURE

The following proposition provides necessary conditions on the sizes of the network architectures for the discriminator and generator network for stability.

Theorem A.7 of Mescheder et al. (2018) shows that matrices of the form

$$-J = \begin{bmatrix} 0 & -B \\ B^\top & -C \end{bmatrix} \tag{23}$$

are stable if $B$ is full rank and $C > 0$. The following proposition provides necessary conditions on the sizes of the network architectures for the discriminator and generator network for stability.

---

[6]Mescheder et al. (2018) imply that their results hold for a convex combination of the two gradient penalties, which would in turn imply our results will hold in this case. However, we have not included the details here.

**Proposition E.1.** *Consider training a generative adversarial network via a zero-sum game with generator network $G_\theta$, discriminator network $D_\omega$, and loss $f(\theta, \omega)$ with regularization $R_j(\theta, \omega)$ (for some $j \in \{1, 2\}$) such that Assumption 1 is satisfied for an equilibrium $x^* = (\theta^*, \omega^*)$. Independent of the learning rate ratio and the regularization parameter $\mu$, for $x^*$ to be stable it is necessary that the dimension of the discriminator network parameter vector is at least half as large as the corresponding generator network parameter vector: $n_2 \geq n_1/2$ where $\theta \in \mathbb{R}^{n_1}$ and $\omega \in \mathbb{R}^{n_2}$.*

The intuition for the why this proposition should hold follows immediately from observing the structure of the Jacobian: for any matrix of the form (23), at least one eigenvalue will be purely imaginary if $n_2 < n_1/2$ where $B \in \mathbb{R}^{n_1 \times n_2}$ and $C \in \mathbb{R}^{n_2 \times n_2}$. This proposition follows immediately from observing the structure of the Jacobian: for any matrix of the form

$$-J = \begin{bmatrix} 0 & -B \\ B^\top & -C \end{bmatrix}$$

at least one eigenvalue will be purely imaginary if $n_2 < n_1/2$ where $B \in \mathbb{R}^{n_1 \times n_2}$ and $C \in \mathbb{R}^{n_2 \times n_2}$. Indeed, by Lyapunov's stability theorem for linear systems (Hespanha, 2018, Theorem 8.2), a matrix $A$ is Hurwitz stable if and only if for every symmetric positive definite $Q = Q^\top > 0$, there exists a unique symmetric positive definite $P = P^\top > 0$, such that $A^\top P + PA = -Q$. Hence, $-J$ is Hurwitz stable if and only if there exists a $P = P^\top > 0$ such that

$$0 < Q = \begin{bmatrix} 0 & -B \\ B^\top & C \end{bmatrix} \begin{bmatrix} P_1 & P_2 \\ P_2^\top & P_3 \end{bmatrix} + \begin{bmatrix} P_1 & P_2 \\ P_2^\top & P_3 \end{bmatrix} \begin{bmatrix} 0 & B \\ -B^\top & C \end{bmatrix}$$

$$= \begin{bmatrix} -BP_2^\top - P_2B^\top & -BP_3 + P_1B + P_2C \\ B^\top P_1 + CP_2^\top - P_3B^\top & B^\top P_2 + CP_3 + P_2^\top B + P_3C \end{bmatrix}$$

Since this is a symmetric positive definite matrix, the block diagonal components must also be symmetric positive definite so that $-BP_2 - P_2B^\top > 0$.[7] Recall that $B \in \mathbb{R}^{n_1 \times n_2}$ and $P_2 \in \mathbb{R}^{n_2 \times n_1}$. Hence, a necessary condition for this matrix to be positive definite is that $n_2 \geq n_1/2$ for $-BP_2 - P_2B^\top$ to have full rank; of course this is not sufficient, but it is necessary. It is easy to see this argument is independent of whether a learning rate ratio $\tau \neq 0$ or regularization is incorporated.

## F  PROOF OF HELPER LEMMAS AND THEOREM 4 FOR $\tau-$GDA CONVERGENCE

In this appendix section, we prove the following two lemmas. The proofs build on one another so we state them jointly here. We then invoke them to prove Theorem 4.

**Lemma F.1.** *Consider a zero-sum game $(f_1, f_2) = (f, -f)$ defined by $f \in C^r(X, \mathbb{R})$ for some $r \geq 2$. Suppose that $x^*$ is a differential Stackelberg equilibrium and that given $\tau > 0$, $\mathrm{spec}(-J_\tau(x^*)) \subset \mathbb{C}_-^\circ$. Let $\gamma = \min_{\lambda \in \mathrm{spec}(J_\tau(x^*))} 2\mathrm{Re}(\lambda)/|\lambda|^2$. For any $\gamma_1 \in (0, \gamma)$, $\tau$-GDA converges locally asymptotically.*

**Lemma F.2.** *Consider a zero-sum game $(f_1, f_2) = (f, -f)$ defined by $f \in C^r(X, \mathbb{R})$ for some $r \geq 2$. Suppose that $x^*$ is a differential Stackelberg equilibrium and that given $\tau$, $\mathrm{spec}(-J_\tau(x^*)) \subset \mathbb{C}_-^\circ$. Let $\gamma = \min_{\lambda \in \mathrm{spec}(J_\tau(x^*))} 2\mathrm{Re}(\lambda)/|\lambda|^2$, and $\lambda_\mathfrak{m} = \arg\min_{\lambda \in \mathrm{spec}(J_\tau(x^*))} 2\mathrm{Re}(\lambda)/|\lambda|^2$. For any $\alpha \in (0, \gamma)$, $\tau$-GDA with learning rate $\gamma_1 = \gamma - \alpha$ converges locally asymptotically at a rate of $O((1 - \frac{\alpha}{4\beta})^{k/2})$ where $\beta = (2\mathrm{Re}(\lambda_\mathfrak{m}) - \alpha|\lambda_\mathfrak{m}|^2)^{-1}$.*

### F.1  PROOF OF LEMMA F.1

Suppose that $x^*$ is a differential Stackelberg or Nash equilibrium and that $0 < \tau < \infty$ is such that $\mathrm{spec}(-J_\tau(x^*)) \subset \mathbb{C}_-^\circ$. For the discrete time dynamical system $x_{k+1} = x_k - \gamma_1 \Lambda_\tau g(x_k)$, it is well known that if $\gamma_1$ is chosen such that $\rho(I - \gamma_1 J_\tau(x^*)) < 1$, then $x_k$ locally (exponentially) converges to $x^*$ (Ortega & Rheinboldt, 1970). With this in mind, we formulate an optimization problem to find the upper bound $\gamma$ on the learning rate $\gamma_1$ such that for all $\gamma_1 \in (0, \gamma)$, the spectral radius of the local linearization of the discrete time map is a contraction which is precisely $\rho(I - \gamma_1 J_\tau(x^*)) < 1$. The optimization problem is given by

$$\gamma = \min_{\gamma > 0} \left\{ \gamma : \max_{\lambda \in \mathrm{spec}(J_\tau(x^*))} |1 - \gamma\lambda| \leq 1 \right\}. \tag{24}$$

---

[7] If a block matrix $Q$ with block entries $Q_{ij}$ for $i, j \in \{1, 2\}$ is positive definite symmetric, then $Q_{ii} > 0$ for $i = 1, 2$.

The intuition is as follows. The inner maximization problem is over a finite set $\mathrm{spec}(J_\tau(x^*)) = \{\lambda_1, \ldots, \lambda_n\}$ where $J_\tau(x^*) \in \mathbb{R}^{n \times n}$. As $\gamma$ increases away from zero, each $|1 - \gamma\lambda_i|$ shrinks in magnitude. The last $\lambda_i$ such that $1 - \gamma\lambda_i$ hits the boundary of the unit circle in the complex plane (that is, $|1 - \gamma\lambda_i| = 1$) gives us the optimal value of $\gamma$ and the element of $\mathrm{spec}(J_\tau(x^*))$ that achieves it. Examining the constraint, we have that for each $\lambda_i$, $\gamma(\gamma|\lambda_i|^2 - 2\mathrm{Re}(\lambda_i)) \leq 0$ for any $\gamma > 0$. As noted this constraint will be tight for one of the $\lambda$, in which case $\gamma = 2\mathrm{Re}(\lambda)/|\lambda|^2$ since $\gamma > 0$. Hence, by selecting $\gamma = \min_{\lambda \in \mathrm{spec}(J_\tau(x^*))} 2\mathrm{Re}(\lambda)/|\lambda|^2$, we have that $|1 - \gamma_1\lambda| < 1$ for all $\lambda \in \mathrm{spec}(J_\tau(x^*))$ and any $\gamma_1 \in (0, \gamma)$.

To see this is the case, let

$$\gamma = \min_{\lambda \in \mathrm{spec}(J_\tau(x^*))} 2\mathrm{Re}(\lambda)/|\lambda|^2$$

and

$$\lambda_\mathtt{m} = \arg \min_{\lambda \in \mathrm{spec}(J_\tau)} 2\mathrm{Re}(\lambda)/|\lambda|^2.$$

Using the expression for $\gamma$, we have that

$$1 - 2\gamma\mathrm{Re}(\lambda) + \gamma^2(\mathrm{Re}(\lambda)^2 + \mathrm{Im}(\lambda)^2) = 1 - 2\frac{2\mathrm{Re}(\lambda_\mathtt{m})}{|\lambda_\mathtt{m}|^2}\mathrm{Re}(\lambda) + \left(\frac{2\mathrm{Re}(\lambda_\mathtt{m})}{|\lambda_\mathtt{m}|^2}\right)^2 |\lambda|^2.$$

Now, using the fact that $\mathrm{Re}(\lambda)/|\lambda|^2 > \mathrm{Re}(\lambda_\mathtt{m})/|\lambda_\mathtt{m}|^2$, we have

$$1 - 4\frac{\mathrm{Re}(\lambda_\mathtt{m})}{|\lambda_\mathtt{m}|^2}\mathrm{Re}(\lambda) + \left(\frac{2\mathrm{Re}(\lambda_\mathtt{m})}{|\lambda_\mathtt{m}|^2}\right)^2 |\lambda|^2 \leq 1 - 2\frac{2\mathrm{Re}(\lambda_\mathtt{m})}{|\lambda_\mathtt{m}|^2}\mathrm{Re}(\lambda) + \left(\frac{2\mathrm{Re}(\lambda_\mathtt{m})}{|\lambda_\mathtt{m}|^2}\right)^2 \frac{|\lambda_\mathtt{m}|^2\mathrm{Re}(\lambda)}{\mathrm{Re}(\lambda_\mathtt{m})}$$

$$= 1 - 4\frac{\mathrm{Re}(\lambda_\mathtt{m})}{|\lambda_\mathtt{m}|^2}\mathrm{Re}(\lambda) + 4\frac{\mathrm{Re}(\lambda_\mathtt{m})}{|\lambda_\mathtt{m}|^2}\mathrm{Re}(\lambda)$$

$$= 1$$

as claimed. From this argument, it is clear that for any $\gamma_1 \in (0, \gamma)$, $|1 - \gamma_1\lambda| < 1$ for all $\lambda \in \mathrm{spec}(J_\tau(x^*))$.

Now, consider any $\alpha \in (0, \gamma)$ and let $\beta = (2\mathrm{Re}(\lambda_\mathtt{m}) - \alpha|\lambda_\mathtt{m}|^2)^{-1}$. Observe that $\gamma_1 = \gamma - \alpha$ so that $\gamma_1 \in (0, \gamma)$. Hence,

$$|1 - (\gamma - \alpha)\lambda_\mathtt{m}|^2 = \left(1 - \left(\frac{2\mathrm{Re}(\lambda_\mathtt{m})}{|\lambda_\mathtt{m}|^2} - \alpha\right)\mathrm{Re}(\lambda_\mathtt{m})\right)^2 + \left(\frac{2\mathrm{Re}(\lambda_\mathtt{m})}{|\lambda_\mathtt{m}|^2} - \alpha\right)^2 \mathrm{Im}(\lambda_\mathtt{m})^2$$

$$= 1 - 4\frac{\mathrm{Re}(\lambda_\mathtt{m})^2}{|\lambda_\mathtt{m}|^2} + 2\alpha\mathrm{Re}(\lambda_\mathtt{m}) + 4\frac{\mathrm{Re}(\lambda_\mathtt{m})^2}{|\lambda_\mathtt{m}|^2} - 4\alpha\mathrm{Re}(\lambda_\mathtt{m}) + \alpha^2|\lambda_\mathtt{m}|^2$$

$$= 1 - 2\alpha\mathrm{Re}(\lambda_\mathtt{m}) + \alpha^2|\lambda_\mathtt{m}|^2$$

$$= 1 - \frac{\alpha}{\beta}$$

so that

$$\rho(I - \gamma_1 J_\tau(x^*)) < \left(1 - \frac{\alpha}{\beta}\right)^{1/2}.$$

Hence, the $\rho(I - \gamma_1 J_\tau(x^*)) < 1$ so that an application of Proposition B.1 gives us the desired result.

### F.2 PROOF OF LEMMA F.2

To prove this lemma, we build directly on the conclusion of the proof of Lemma F.1. Indeed, since

$$\rho(I - \gamma_1 J_\tau(x^*)) < \left(1 - \frac{\alpha}{\beta}\right)^{1/2},$$

given $\varepsilon = \frac{\alpha}{4\beta} > 0$ there exists a norm $\|\cdot\|$ (cf. Lemma 5.6.10 in Horn & Johnson (1985))[8] such that

$$\|I - \gamma_1 J_\tau(x^*)\| \leq \left(1 - \frac{\alpha}{\beta}\right)^{1/2} + \frac{\alpha}{4\beta} \leq \left(1 - \frac{\alpha}{2\beta}\right)^{1/2}$$

___

[8]The norm that exists can easily be constructed as essentially a weighted induced 1-norm. Note that the norm construction is not unique. The proof in Horn & Johnson (1985) is by construction and the construction of this norm can be found there.

where the last inequality holds by Lemma B.1. Taking the Taylor expansion of $I - \gamma_1 g_\tau(x)$ around $x^*$, we have

$$I - \gamma_1 g_\tau(x) = (I - \gamma_1 g_\tau(x^*)) + (I - \gamma_1 J_\tau(x^*))(x - x^*) + R_2(x - x^*)$$

where $R_2(x - x^*)$ is the remainder term satisfying $R_2(x - x^*) = o(\|x - x^*\|)$ as $x \to x^*$.[9] This implies that there is a $\delta > 0$ such that $\|R_2(x - x^*)\| \le \frac{\alpha}{8\beta}\|x - x^*\|$ whenever $\|x - x^*\| < \delta$. Hence,

$$
\begin{aligned}
\|I - \gamma_1 g_\tau(x) - (I - \gamma_1 g_\tau(x^*))\| &\le \left( \|I - \gamma_1 J_\tau(x^*)\| + \frac{\alpha}{4\beta} \right) \|x - x^*\| \\
&\le \left( \left(1 - \frac{\alpha}{2\beta}\right)^{1/2} + \frac{\alpha}{8\beta} \right) \|x - x^*\| \\
&\le \left(1 - \frac{\alpha}{4\beta}\right)^{1/2} \|x - x^*\|
\end{aligned}
$$

where the last inequality holds again by Lemma B.1. Hence,

$$\|x_k - x^*\| \le \left(1 - \frac{\alpha}{4\beta}\right)^{k/2} \|x_0 - x_*\| \tag{25}$$

whenever $\|x_0 - x^*\| < \delta$ which verifies the claimed convergence rate.

### F.3   PROOF OF THEOREM 4

This result follows directly from Theorem 1, Theorem B.2, Lemma F.2, and the following result.

**Proposition F.1** (Jin et al. 2020). *Consider a zero-sum game $(f_1, f_2) = (f, -f)$ defined by $f \in C^r(X, \mathbb{R})$ for some $r \ge 2$. Suppose that $x^*$ is a differential Nash equilibrium. Then, $\mathrm{spec}(-J_\tau(x^*)) \subset \mathbb{C}_-^\circ$ for all $\tau \in (0, \infty)$.*

Now to prove Theorem 4, we apply Theorem 1 to construct $\tau^*$ via the guard map $\nu(\tau) = \det(-J_\tau(x^*) \boxplus -J_\tau(x^*))$ such that for all $\tau \in (\tau^*, \infty)$, $\mathrm{spec}(J_\tau(x^*)) \subset \mathbb{C}_+^\circ$. This guarantees that $\mathrm{spec}(-J_\tau(x^*)) \subset \mathbb{C}_-^\circ$ for any $\tau \in (\tau^*, \infty)$ and hence the nonlinear dynamical system

$$\dot{x} = -\Lambda_\tau g(x)$$

is locally asymptotically (in fact, exponentially) stable by the Hartman-Grobman theorem (cf. Theorem B.2). Therefore, for any $\tau \in (\tau^*, \infty)$, by Lemma F.2, $\tau$-GDA converges with a rate of $O((1 - \frac{\alpha}{4\beta})^{k/2})$. Finally, $\tau^* = 0$ by Proposition F.1 if $x^*$ is a differential Nash equilibrium. This concludes the proof.

**Intuition for Proposition F.1.**   While this result was shown in Jin et al. (2020), we give some intuition for why it holds based on the quadratic numerical range tool. Indeed simple observation of the eigenvalues of the quadratic numerical range show that $\mathrm{spec}(J_\tau(x^*)) \subset \mathbb{C}_+^\circ$ for any $\tau \in (0, \infty)$. Recall that

$$\mathcal{W}^2(J_\tau(x^*)) = \bigcup_{v \in W_1, w \in W_2} \mathrm{spec}(J_\tau^{v,w}(x^*))$$

where

$$J_\tau^{v,w}(x^*) = \begin{bmatrix} \langle D_1^2 f(x^*)v, v \rangle & \langle D_{12}f(x^*)w, v \rangle \\ \langle -\tau D_{12}^\top f(x^*)v, w \rangle & \langle -\tau D_2^2 f(x^*)w, w \rangle \end{bmatrix}$$

and $W_i = \{z \in \mathbb{C}^{n_i} : \|z\| = 1\}$ for each $i = 1, 2$. Fix $v \in W_1$ and $w \in W_2$ and consider

$$J_\tau^{v,w}(x^*) = \begin{bmatrix} a & b \\ -\tau \bar{b} & \tau d \end{bmatrix}$$

Then, the elements of $\mathcal{W}^2(J_\tau(x^*))$ are of the form

$$\lambda_\tau = \tfrac{1}{2}(a + \tau d) \pm \tfrac{1}{2}\sqrt{(a - \tau d)^2 - 4\tau|b|^2}$$

---

[9]The notation $R_2(x - x^*) = o(\|x - x^*\|)$ as $x \to x^*$ means $\lim_{x \to x^*} \|R_2(x - x^*)\| / \|x - x^*\| = 0$.

where $a = \langle D_1^2 f(x^*)v, v \rangle$, $b = \langle D_{12}f(x^*)w, v \rangle$ and $d = \langle -D_2^2 f(x^*)w, w \rangle$ for vectors $v \in W_1$ and $w \in W_2$. Since $D_1^2 f(x^*) > 0$ and $-D_2 f(x^*) > 0$, $\mathrm{Re}(\lambda(J_{v,w}(x^*))) > 0$ since $\lambda(J_{v,w}(x^*)) = \frac{1}{2}(a+d) \pm \frac{1}{2}\sqrt{(a-d)^2 - 4|b|^2}$. The introduction of $\tau \in (0, \infty)$ does not alter the sign. This is obvious when $\mathrm{Im}(\lambda_\tau) \neq 0$. On the other hand, if $\mathrm{Im}(\lambda_\tau) = 0$ so that $(a - \tau d)^2 > 4\tau|b|^2$, then

$$\mathrm{Re}(\lambda_\tau) > \tfrac{1}{2}(a + \tau d) - \tfrac{1}{2}\sqrt{(a - \tau d)^2 - 4\tau|b|^2} > 0.$$

The last inequality is easily seen to be equivalent to $-ad < |b|^2$, which holds for any pair of vectors $(v, w)$ such that $v \in W_1$ and $w \in W_2$ since $a > 0$ and $d > 0$. Hence, for any $\tau \in (0, \infty)$, $\mathrm{spec}(J_\tau(x^*)) \subset \mathbb{C}_+^\circ$ since the spectrum of an operator is contained in its quadratic numerical range and the above argument shows that $\mathcal{W}^2(J_\tau(x^*)) \subset \mathbb{C}_+^\circ$.

## G    PROOF OF COROLLARY 1: FINITE TIME CONVERGENCE OF $\tau$−GDA

Let $\| \cdot \|$ be the norm that exists (via construction a la Horn & Johnson (1985, Lem. 5.6.10)) in the proof of Lemma F.2 which is given in Appendix F. Following standard arguments, (25) in the proof of Lemma F.2 implies a finite time convergence guarantee. Indeed, let $\varepsilon > 0$ be given. Since $0 < \frac{\alpha}{4\beta} < 1$ we have that $(1 - \alpha/(4\beta))^k < \exp(-k\alpha/(4\beta))$. Hence,

$$\|x_k - x^*\| \leq \exp(-k\alpha/(4\beta))\|x_0 - x^*\|.$$

In turn, this implies that $x_k \in B_\varepsilon(x^*)$, meaning that $x_k$ is a $\varepsilon$-differential Stackelberg equilibrium for all $k \geq \lceil \frac{4\beta}{\alpha} \log(\|x_0 - x^*\|/\varepsilon) \rceil$ whenever $\|x_0 - x^*\| < \delta$.

Now, given that $f_i \in C^r(X, \mathbb{R})$ for $r \geq 2$, $I - \gamma_1 J_\tau(x)$ is locally Lipschitz with constant $L$ so that we can find an explicit expression for $\delta$ in terms of $L$. Indeed, recall that $R_2(x - x^*) = o(\|x - x^*\|)$ as $x \to x^*$ which means $\lim_{x \to x^*} \|R_2(x - x^*)\|/\|x - x^*\| = 0$ so that

$$\|R_2(x - x^*)\| \leq \int_0^1 \|I - \gamma_1 J_\tau(x^* + \eta(x - x^*)) - (I - \gamma_1 J_\tau(x^*))\|\|x - x^*\| \, d\eta \leq \frac{L}{2}\|x - x^*\|^2$$

Observing that

$$\|R_2(x - x^*)\| \leq \frac{L}{2}\|x - x^*\|^2 = \frac{L}{2}\|x - x^*\|\|x - x^*\|,$$

we have that the $\delta > 0$ such that $\|R_2(x - x^*)\| \leq \alpha/(8\beta)\|x - x^*\|$ is $\delta = \alpha/(4L\beta)$.

**Comments on computing the neighborhood $B_\delta(x^*)$.** We note that we have essentially given a proof that there exists a neighborhood on which $\tau$-GDA converges. Of course, due to the non-convexity of the problem in general, this neighborhood could be arbitrarily small. We provide an estimate of the neighborhood size using the local Lipschitz constant of the local linearization $I - \gamma_1 J_\tau(x^*)$. One way to better understand the size of this neighborhood is to use Lyapunov analysis, a tool which is well explored in the singular perturbation theory (Kokotovic et al., 1986). In particular, Lyapunov methods can be applied directly to the nonlinear system if one can construct Lyapunov functions for the fast and slow subsystems individually—also known as the boundary layer model and reduced order model. With these Lyapunov functions in hand, one can "stitch" the two together (via convex combination) and show under some reasonable assumptions that this combined function is a Lyapunov function for the overall singularly perturbed system. The benefit of this analysis is that the Lyapunov function gives one an estimate of the region of attraction (via, e.g., the level sets); however, it is not easy to construct a Lyapunov function for a nonlinear system in general. We leave expanding to such methods to future work.

## H    CONVERGENCE OF STOCHASTIC GDA WITH TIMESCALE SEPARATION

In this section, we analyze convergence when players do not have oracle access to their gradients but instead have an unbiased estimator in the presence of zero mean, finite variance noise. Specifically, we show that the agents will converge locally asymptotically almost surely to a differential Stackelberg equilibrium.

The key insight in this section is that due to Theorem 1 in the main body, we know that a critical point $x^*$ is stable for $\dot{x} = -\Lambda_\tau g(x)$ for a range of finite learning rates $\tau \in (\tau^*, \infty)$ if and only

if $x^*$ is a differential Stackelberg equilibrium. Hence, treating $\dot{x} = -\Lambda_\tau g(x)$ as the continuous time limiting differential equation in the so-called ordinary differential equation (ODE) method in stochastic approximation (Borkar, 2008), we apply classical stochastic approximation analysis to conclude that the stochastic gradient descent-ascent update with timescale separation converges.

## H.1 Asymptotic Convergence Guarantees via Stochastic Approximation

The stochastic form of the update is given by

$$x_{k+1} = x_k - \gamma_k(\Lambda_\tau g(x_k) + w_{k+1}) \tag{26}$$

where $w_{k+1}$ is a zero mean, finite variance random variable and $\{\gamma_k\}$ is the learning rate sequence.

**Assumption 2.** *The stochastic process $\{w_k\}$ is a martingale difference sequence with respect to the increasing family of $\sigma$-fields defined by*

$$\mathcal{F}_k = \sigma(x_\ell, w_\ell, \ell \le k), \ \forall k \ge 0,$$

*so that $\mathbb{E}[w_{k+1}|\ \mathcal{F}_k] = 0$ almost surely (a.s.) for all $k \ge 0$. Moreover, $w_k$ is square-integrable so that, for some constant $C > 0$,*

$$\mathbb{E}[\|w_{k+1}\|^2|\ \mathcal{F}_k] \le C(1 + \|x_k\|^2) \ a.s., \ \forall k \ge 0.$$

We note that this assumption has been relaxed in the literature (Thoppe & Borkar, 2019), however simplicity, we state the theorem with the most accessible criteria. We remark below in the paragraph on extensions to concentration bounds on the nature of the relaxed assumptions.

**Theorem H.1.** *Consider a zero-sum game $(f, -f)$ such that $f \in C^r(X, \mathbb{R})$ for some $r \ge 2$. Suppose that Assumption 2 holds and that $\{\gamma_k\}$ is square summable but not summable—i.e., $\sum_k \gamma_k^2 < \infty$, yet $\sum_k \gamma_k = \infty$. For any $\tau \in (0, \infty)$, the sequence $\{x_k\}$ generated by (26) converges to a, possibly sample path dependent, internally chain transitive invariant set of $\dot{x} = -\Lambda_\tau g(x)$. Moreover, if $x^*$ is a differential Stackelberg equilibrium, then there exists a finite $\tau^* \in [0, \infty)$ such that $\{x_k\}$ almost surely converges locally asymptotically to $x^*$ for every $\tau \in (\tau^*, \infty)$.*

*Proof.* The convergence of $\{x_k\}$ to a, possibly sample path dependent, compact connected internally chain transitive invariant set of $\dot{x} = -\Lambda_\tau g(x)$ follows from classical results in stochastic approximation theory (Borkar (2008, Chap. 2); Benaim (1996)).

Suppose that $x^*$ is a differential Stackelberg equilibrium. By Theorem 1, there exists a finite $\tau^* \in [0, \infty)$ such that for all $\tau \in (\tau^*, \infty)$, $x^*$ is a locally exponentially stable equilibrium of the continuous time dynamics $\dot{x} = -\Lambda_\tau g(x)$—that is, $\mathrm{spec}(-J_\tau(x^*)) \subset \mathbb{C}_-^\circ$ for all $\tau \in (\tau^*, \infty)$.

Fix arbitrary $\tau \in (\tau^*, \infty)$. Since $\mathrm{spec}(-J_\tau(x^*)) \subset \mathbb{C}_-^\circ$, $\det(-J_\tau(x^*)) \ne 0$ so that $x^*$ is an isolated critical point. Furthermore, exponentially stability of $x^*$ implies that there exists a (local) Lyapunov function defined on a neighborhood of $x^*$ by the converse Lyapunov theorem (Sastry (1999, Thm. 5.17), Krasovskii (1963, Thm. 4.3)). Let $U$ be the neighborhood of $x^*$ on which the local Lyapunov function is defined, such that $U$ contains no other critical points (which is possible since $x^*$ is isolated). That is, let $\Phi : U \to [0, \infty)$ be the local Lyapunov function defined on $U$ where $x^* \in U$, $\Phi$ is positive definite on $U$, and for all $x \in U$, $\frac{d}{dt}\Phi(x) \le 0$ where equality holds for $z \in U$ if and only if $\Phi(z) = 0$. By Corollary 3 (Borkar, 2008, Chap. 2), $\{x_k\}$ converges to an internally chain transitive invariant set contained in $U$ almost surely. The only internally chain transitive invariant set in $U$ is $x^*$. $\square$

The following corollary shows that if there is a finite $\tau^*$ such that $x^*$ is stable for $\dot{x} = -\Lambda_\tau g(x)$, then by Theorem 1 $x^*$ must be a differential Stackelberg equilibrium and in turn, $\{x_k\}$ almost surely converges locally asymptotically to $x^*$ by the above theorem.

**Corollary H.1.** *Consider a zero-sum game $(f, -f)$ such that $f \in C^2(X, \mathbb{R})$. Suppose that Assumption 2 holds and that $\{\gamma_k\}$ is square summable but not summable: $\sum_k \gamma_k^2 < \infty$, yet $\sum_k \gamma_k = \infty$. If there exists a finite $\tau^* \in [0, \infty)$ such that $\mathrm{spec}(-J_\tau(x^*)) \subset \mathbb{C}_-^\circ$ for all $\tau \in (\tau^*, \infty)$, then $x^*$ is a differential Stackelberg equilibrium and $\{x_k\}$ almost surely converges locally asymptotically to $x^*$.*

While (local) almost sure convergence in gradient descent-ascent (Chasnov et al., 2019) to a critical point[10] in the stochastic setting, the result requires time varying learning rates with a sufficient separation in timescale. Specifically, the players need to be using learning rate sequences $\{\gamma_{i,k}\}$ for each $i \in \{1, 2\}$ such that (without loss of generality) not only is it assumed that $\gamma_{1,k} = o(\gamma_{2,k})$, but also $\sum_k \gamma_{1,k}^2 + \gamma_{2,k}^2 < \infty$ and $\sum_k \gamma_{i,k} = \infty$ for each $i \in \{1, 2\}$. The challenge with these assumptions on the learning rate sequences is that empirically the sequences that satisfy them result in poor behavior along the learning path such as getting stuck at saddle points or making no progress. This is, in essence, due to the fact that the faster player—that is, player 2 if $\gamma_{1,k} = o(\gamma_{2,k})$— equilibrates too quickly causing progress to stall. This can result in undesirable behavior such as vanishing gradients (so that the discriminator does not provide enough information for the generator to make progress), mode collapse, or failure to converge in practical applications such as generative adversarial networks.

On the other hand, our convergence result gives a similar guarantee with less restrictive requirements on the learning rate sequence. In particular, only a single learning rate sequence is required (so that the algorithm can be viewed as a single timescale stochastic approximation update) as long as the fast player (who, without loss of generality, is player 2 in this paper) scales their estimated gradient by $\tau \in (\tau^*, \infty)$ where $\tau^*$ is as in Theorem 1.

## H.2 Extensions to concentration bounds and relaxed assumptions on stepsizes

It is possible to obtain concentration bounds and even finite time, high probability guarantees on convergence leveraging recent advances in stochastic approximation (Borkar, 2008; Kamal, 2010; Thoppe & Borkar, 2019). To our knowledge, the concentration bounds in (Thoppe & Borkar, 2019) require the weakest assumptions on learning rates—e.g., the learning rate sequence $\{\gamma_k\}$ needs only to satisfy $\sum_k \gamma_k = \infty$, $\lim_{k \to \infty} \gamma_k = 0$, and $\sum_k \gamma_k \leq 1$. Specifically, since it is assumed, for the zero sum game $(f, -f)$, that $f \in C^2(X, \mathbb{R})$ and $x^*$ is a differential Stackelberg equilibrium, Theorem 1 implies that $x^*$ is a locally asymptotically stable attractor of $\dot{x} = -\Lambda_\tau g(x)$ for arbitrary fixed $\tau \in (\tau^*, \infty)$, and hence, the concentration bounds in Theorem 1.1 and 1.2 of (Thoppe & Borkar, 2019) directly apply.

Furthermore, we note that in applications such as generative adversarial networks, while it has been observed that timescale separation heuristics such as unrolling or annealing the stepsize of the discriminator work well, in the stochastic case, summmable/square-summable assumptions on stepsizes are generally too restrictive in practice since they lead to a rapid decay in the stepsize which, in turn, can stall progress. On the other hand, stepsize sequences such as $\gamma_k = 1/(k+1)^\beta$ for $\beta \in (0, 1]$— a sequence which satisfies the assumptions posed in (Thoppe & Borkar, 2019)—tend not to have this issue of decaying too rapidly for appropriately chosen $\beta$, while also maintaining the guarantees of the theoretical results. We state a convergence guarantee under these relaxed assumptions in Proposition H.1 below.

Let $\tilde{x}(t)$ be the asymptotic pseudo-trajectories of the stochastic approximation process $\{x_k\}$. That is, $\tilde{x}(t)$ are linear interpolates between the sample points $x_k$ generated by the stochastic $\tau$-GDA process, and are defined by

$$\tilde{x}(t) = \tilde{x}(t_k) + \frac{(t - t_k)}{\gamma_k}(\tilde{x}(t_{k+1}) - \tilde{x}(t_k))$$

where $t_k = t_k + \gamma_k$ and $t_0 = 0$.

**Assumption 3.** *The stochastic process $\{w_k\}$ is a martingale difference sequence with respect to the increasing family of $\sigma$-fields defined by*

$$\mathcal{F}_k = \sigma(x_\ell, w_\ell, \ell \leq k), \ \forall k \geq 0,$$

*so that $\mathbb{E}[w_{k+1} | \mathcal{F}_k] = 0$ almost surely for all $k \geq 0$. Furthermore, there exists $c_1, c_2 \in C(\mathbb{R}^d, \mathbb{R}_{>0})$ such that*

$$\Pr\{\|w_{k+1}\| > v | \mathcal{F}_k\} \leq c_1(x_k) \exp(-c_2(x_k)v), \ n \geq 0$$

*for all $v \geq \tilde{v}$ where $\tilde{v}$ is some sufficiently large, fixed number.*

---

[10]To date it has not been shown that for a sufficient separation in timescale the only critical point attractors are local minmax.

**Proposition H.1.** *Suppose that Assumption 3 holds and that $x^*$ is a differential Stackelberg equilibrium. Let $\gamma_k = 1/(k+1)^\beta$ where $\beta \in (0,1]$. There exists a $\tau^* \in [0,\infty)$ and an $\epsilon_0 \in (0,\infty)$ such that for any fixed $\epsilon \in (0,\epsilon_0]$, there exists functions $h_1(\epsilon) = O(\log(1/\epsilon))$ and $h_2(\epsilon) = O(1/\epsilon)$ so that when $T \geq h_1(\epsilon)$ and $k_0 \geq K_\tau$ where $K_\tau$ is such that $1/\gamma_k \geq h_2(\epsilon)$ for all $k \geq K_\tau$, the stochastic iterates of $\tau$-GDA with stepsize sequence $\gamma_k$ and timescale separation $\tau \in (\tau^*, \infty)$ satisfy*

$$\Pr\{\|\tilde{x}(t) - x^*\| \leq \epsilon \, \forall t \geq t_{k_0} + T + 1 \mid \tilde{x}(t_{k_0}) \in B_\epsilon(x^*)\} = 1 - O(k_0^{1-\beta/2} \exp(-C_\tau k_0^{\beta/2}))$$

*for some constant $C_\tau > 0$.*

The proof largely follows from the proofs of Theorem 1.1 and 1.2 in (Thoppe & Borkar, 2019), combined with the existence of a finite timescale separation parameter obtained via Theorem 1. Indeed, since $x^*$ is a differential Stackelberg equilibrium, by Theorem 1 there exists a range of $\tau$—namely, $(\tau^*, \infty)$—such that for any $\tau \in (\tau^*, \infty)$, $x^*$ is a locally asymptotically stable equillibrium for $\dot{x} = -\Lambda_\tau g(x)$. Hence, fixing any $\tau \in (\tau^*, \infty)$, a converse Lyapunov theorem can be applied to construct a local Lyapunov function. Let $V : \mathbb{R}^n \to \mathbb{R}$ be this Lyapunov function so that there exists $r, r_0, \epsilon_0 > 0$ such that $r > r_0$, and

$$B_\epsilon(x^*) \subseteq V^{r_0} \subset \mathcal{N}_{\epsilon_0}(V^{r_0}) \subseteq V^r$$

for any $\epsilon \in (0, \epsilon_0]$ where, for a given $q > 0$, $V^q = \{x \in \text{dom}(V) : V(x) \leq q\}$ and $\mathcal{N}_{\epsilon_0}(V^{r_0})$ is an $\epsilon_0$–neighborhood of $V^{r_0}$—i.e., $\mathcal{N}_{\epsilon_0}(V^{r_0}) = \{x \in \mathbb{R}^n \mid \exists y \in V^{r_0}, \|x - y\| \leq \epsilon_0\}$. From here, the result follows from an application of the results in the work by Thoppe & Borkar (2019).

The utility of this result is that it provides a guarantee in the stochastic setting for a more reasonable and practically useful stepsize sequence. However, constructing the constants such as $K_\tau$, $C_\tau$ and $\epsilon_0$ is highly non-trivial as can be seen in the work of Thoppe & Borkar (2019) and similar works in the area of stochastic approximation (Borkar, 2008). One direction of future work is examining the Lyapunov approach for directly analyzing the nonlinear singularly perturbed system; it is known, however, that the stochastic singularly perturbed systems have much weaker guarantees in terms of stability (Kokotovic et al., 1986, Chap. 4).

# I   STABILITY OF $\infty$−GDA: A SINGULAR PERTURBATION APPROACH

The examples included in Section 3 provide evidence that there exists a range of finite learning rate ratios for which differential Stackelberg equilibrium are stable and a range of learning rate ratios for which non-equilibrium critical points are unstable. Yet, until this paper, no result has appeared in the literature on gradient descent-ascent with timescale separation confirming this behavior in general.

The closest existing result studies the limiting case $\tau \to \infty$. As mentioned previously Jin et al. (2020) show that as $\tau \to \infty$, the set of stable critical points with respect to the dynamics $\dot{x} = -\Lambda_\tau g(x)$ coincide with the set of differential Stackelberg equilibrium. However, an equivalent result in the context of general singularly perturbed systems has been known in the literature ( Kokotovic et al. 1986, Chap. 2). We give a proof based on this type of analysis because it reveals a new set of analysis tools to the study of game-theoretic formulations of machine learning and optimization problems. The formal statement (which is a restatement of the result from Jin et al. 2020 in our notation) is given below.

**Proposition I.1.** *Consider a zero-sum game $(f_1, f_2) = (f, -f)$ defined by $f \in C^r(X, \mathbb{R})$ for some $r \geq 2$. Suppose that $x^*$ is such that $g(x^*) = 0$ and $\det(D_2^2 f_2(x^*)) \neq 0$. Then, as $\tau \to \infty$, $\text{spec}(-J_\tau(x^*)) \subset \mathbb{C}_-^\circ$ if and only if $x^*$ is a differential Stackelberg equilibrium.*

The structure of this proof is as follows. We begin by introducing general background for analyzing general singularly perturbed systems. Following this, we consider the linearization of the singularly perturbed system that approximates the simultaneous gradient dynamics and describe how insights made about this system translate to the corresponding nonlinear system. Finally, we analyze the stability of the linear system around a critical point to arrive at the stated result. The analysis is primarily from Kokotovic et al. (1986).

**Analysis of General Singularly Perturbed Systems.**   Let us begin by considering a general singularly perturbed system for $x \in \mathbb{R}^n$, $z \in \mathbb{R}^m$, and a sufficiently small parameter $\varepsilon > 0$ given by

$$\dot{x} = f(x, z, \varepsilon, t), \quad x(t_0, \varepsilon) = x_0, x \in \mathbb{R}^n$$
$$\varepsilon\dot{z} = g(x, z, \varepsilon, t), \quad z(t_0, \varepsilon) = z_0, z \in \mathbb{R}^m \tag{27}$$

where $f$ and $g$ are assumed to be sufficiently many times continuously differential functions of the arguments $x$, $z$, $\varepsilon$, and $t$. Observe that when $\varepsilon = 0$, the dimension of the system given in (27) drops from $n + m$ to $n$ since $\dot{z}$ degenerates into the equation

$$0 = g(\bar{x}, \bar{z}, 0, t) \tag{28}$$

where the notation of $\bar{x}, \bar{z}$ indicates that the variables belong to the system with $\varepsilon = 0$. We further require the assumption that (28) has $k \geq 1$ isolated roots, which for each $i \in \{1, \ldots, k\}$ are given by

$$\bar{z} = \bar{\phi}_i(\bar{x}, t).$$

We now define an $n$-dimensional manifold $M_\varepsilon$ for any $\varepsilon > 0$ characterized by the expression

$$z(t, \varepsilon) = \phi(x(t, \varepsilon), \varepsilon), \tag{29}$$

where $\phi$ is sufficiently many times continuously differentiable function of $x$ and $\varepsilon$. For $M_\varepsilon$ to be an invariant manifold of the system given in (27), the expression in (29) must hold for all $t > t^*$ if it holds for $t = t^*$. Formally, if

$$z(t^*, \varepsilon) = \phi(x(t^*, \varepsilon), \varepsilon) \to z(t, \varepsilon) = \phi(x(t, \varepsilon), \varepsilon) \quad \forall t \geq t^*, \tag{30}$$

then $M_\varepsilon$ is an invariant manifold for (27). Differentiating the expression in (30) with respect to $t$, we obtain

$$\dot{z} = \frac{d}{dt}\phi(x(t, \varepsilon), \varepsilon) = \frac{d\phi}{\partial x}\dot{x}. \tag{31}$$

Now, multiplying the expression in (31) by $\varepsilon$ and substituting in the forms of $\dot{x}$, $\dot{z}$, and $z$ from (27) and (29), the manifold condition becomes

$$g(x, \phi(x, \varepsilon), \varepsilon, t) = \varepsilon\frac{\partial\phi}{\partial x}f(x, \phi(x, \varepsilon), \varepsilon, t), \tag{32}$$

which $\phi(x, \varepsilon)$ must satisfy for all $x$ of interest and all $\varepsilon \in [0, \varepsilon^*]$, where $\varepsilon^*$ is a positive constant.

We now define

$$\eta = z - \phi(x, \varepsilon).$$

Then, in terms of $x$ and $\eta$, the system becomes

$$\dot{x} = f(x, \phi(x, \varepsilon) + \eta, \varepsilon, t)$$
$$\varepsilon\dot{\eta} = g(x, \phi(x, \varepsilon) + \eta, \varepsilon, t) - \varepsilon\frac{\partial\phi}{\partial x}f(x, \phi(x, \varepsilon) + \eta, \varepsilon, t).$$

**Remark 1.** *One interesting observation is that the above system is exactly the continuous time limiting system for the $\tau$-Stackelberg learning update in Fiez et al. (2020) under a simple transformation of coordinates.*

Observe that the invariant manifold $M_\varepsilon$ is characterized by the fact that $\eta = 0$ implies $\dot{\eta} = 0$ for all $x$ for which the manifold condition from (32) holds. This implies that if $\eta(t_0, \varepsilon) = 0$, it is sufficient to solve the system

$$\dot{x} = f(x, \phi(x, \varepsilon), \varepsilon, t), x(t_0, \varepsilon) = x_0.$$

This system is often referred to as the exact slow model and is valid for all $x, z \in M_\varepsilon$ and $M_\varepsilon$ known as the slow manifold of (35).

**Linearization of Simultaneous Gradient Descent Singularly Perturbed System.** We now consider the singularly perturbed system for simulataneous gradient descent given by

$$\dot{x} = -D_1^2 f_1(x, z)$$
$$\varepsilon\dot{z} = -D_2 f_2(x, z). \tag{33}$$

Let us linearize the system around a point $(x^*, z^*)$. Then,[11]

$$D_1 f_1(x, z) \approx D_1 f_1(x^*, z^*) + D_1^2 f_1(x^*, z^*)(x - x^*) + D_{12} f_1(x^*, z^*)(z - z^*)$$
$$D_2 f_2(x, z) \approx D_2 f_2(x^*, z^*) + D_{21} f_2(x^*, z^*)(x - x^*) + D_2^2 f_2(x^*, z^*)(z - z^*). \tag{34}$$

Defining $u = (x - x^*)$ and $v = (z - z^*)$ and considering a point $(x^*, z^*)$ such that $D_1 f_1(x^*, z^*) = 0$ and $D_2 f_2(x^*, z^*) = 0$, then linearized singularly perturbed system is given by

$$\dot{u} = -D_1^2 f_1(x^*, z^*) u - D_{12} f_1(x^*, z^*) v$$
$$\varepsilon \dot{v} = -D_{21} f_2(x^*, z^*) u - D_2^2 f_2(x^*, z^*) v. \tag{35}$$

To simplify notation, let us define $J_\tau$ as follows

$$J_\tau = \begin{bmatrix} D_1^2 f_1(x^*, z^*) & D_{12} f_1(x^*, z^*) \\ \varepsilon^{-1} D_{21} f_2(x^*, z^*) & \varepsilon^{-1} D_2^2 f_2(x^*, z^*) \end{bmatrix} = \begin{bmatrix} A_{11} & A_{12} \\ \varepsilon^{-1} A_{21} & \varepsilon^{-1} A_{22} \end{bmatrix}$$

along with

$$\dot{w} = \begin{bmatrix} \dot{u} \\ \dot{v} \end{bmatrix} \quad \text{and} \quad w = \begin{bmatrix} u \\ v \end{bmatrix}.$$

Then, an equivalent form of (35) is given by

$$\dot{w} = -J_\tau w. \tag{36}$$

In what follows, we make insights about the behavior of the nonlinear system given in (33) around a critical point $(x^*, z^*)$ by analyzing the linear system given in (36). Recall that if $(x^*, z^*)$ is asymptotically stable with respect to the linear system in (36), then it is also asymptotically stable with respect to the nonlinear system from (33). Moreover, to determine asymptotic stability, it is sufficient to prove that $\mathrm{spec}(J_\tau((x^*, z^*)) \subset \mathbb{C}_+^\circ$. In what follows, we specialize the general analysis of singularly perturbed systems to the singularly perturbed linear system given in (36).

**Stability of Critical Points of Simutaneous Gradient Descent.** The manifold condition from (32) for the system in (36) is given by

$$A_{21} u + A_{22} \phi(u, \varepsilon) = \varepsilon \frac{\partial \phi}{\partial u} (A_{11} u + A_{12} \phi(u, \varepsilon)). \tag{37}$$

We claim that (37) can be satisfied by a function $\phi$ that is linear in $u$. Indeed, defining

$$v = \phi(u, \varepsilon) = -L(\varepsilon) u$$

and then substituting back into (32), we get the simplified manifold condition of

$$A_{21} - A_{22} L(\varepsilon) = -\varepsilon L(\varepsilon) A_{11} + \varepsilon L(\varepsilon) A_{12} L(\varepsilon). \tag{38}$$

Before we prove that an $L(\varepsilon)$ always exists to satisfy (38), consider the change of variables

$$\eta = v + L(\varepsilon) u.$$

The change of variables transforms the system from (36) into the equivalent representation

$$\begin{bmatrix} \dot{u} \\ \dot{\eta} \end{bmatrix} = \begin{bmatrix} A_{11} - A_{12} L(\varepsilon) & A_{12} \\ R(L, \varepsilon) & A_{22} + \varepsilon L(\varepsilon) A_{12} \end{bmatrix} \begin{bmatrix} u \\ \eta \end{bmatrix} \tag{39}$$

where

$$R(L, \varepsilon) = A_{21} - A_{22} L(\varepsilon) + \varepsilon L(\varepsilon) A_{11} - \varepsilon L(\varepsilon) A_{12} L(\varepsilon). \tag{40}$$

Consider that $R(L, \varepsilon) = 0$. Then, the system from (39) has the upper block-triangular form

$$\begin{bmatrix} \dot{x} \\ \dot{\eta} \end{bmatrix} = \begin{bmatrix} A_{11} - A_{12} L(\varepsilon) & A_{12} \\ 0 & A_{22} + \varepsilon L(\varepsilon) A_{12} \end{bmatrix} \begin{bmatrix} x \\ \eta \end{bmatrix}, \tag{41}$$

which has the effect of generating a replacement fast subsystem given by

$$\varepsilon \dot{\eta} = (A_{22} + \varepsilon L A_{12}) \eta.$$

We now proceed to show that an $L(\varepsilon)$ such that $R(L, \varepsilon) = 0$ always exists.

---

[11]Here, the $\approx$ means, e.g., $D_1 f_1(x, z) = D_1 f_1(x^*, z^*) + D_1^2 f_1(x^*, z^*)(x - x^*) + D_{12} f_1(x^*, z^*)(z - z^*) + O(\|x - x^*\|^2 + \|z - z^*\|^2)$, and similarly for $D_2 f_2(x, z)$.

**Lemma I.1.** *If $A_{22}$ is such that $\det(A_{22}) \neq 0$, there is an $\varepsilon^*$ such that for all $\varepsilon \in [0, \varepsilon^*]$, there exists a solution $L(\varepsilon)$ to the matrix quadratic equation*

$$R(L, \varepsilon) = A_{21} - A_{22}L(\varepsilon) + \varepsilon L(\varepsilon)A_{11} - \varepsilon L(\varepsilon)A_{12}L = 0 \tag{42}$$

*which is approximated according to*

$$L(\varepsilon) = A_{22}^{-1}A_{21} + \varepsilon A_{22}^{-2}A_{21}A_0 + O(\varepsilon^2), \tag{43}$$

*where*

$$A_0 = A_{11} - A_{12}A_{22}^{-1}A_{21}. \tag{44}$$

*Proof.* To begin, observe that for $\varepsilon = 0$, the unique solution to (42) is given by $L(0) = A_{22}^{-1}A_{21}$. Now, differentiating $R(L, \varepsilon)$ from (42) with respect to $\varepsilon$, we find

$$A_{22} + \varepsilon L(\varepsilon)A_{12}\frac{dL}{d\varepsilon} - \varepsilon\frac{dL}{d\varepsilon}(A_{11} - A_{12}L(\varepsilon)) = L(\varepsilon)A_{11} - L(\varepsilon)A_{12}L(\varepsilon).$$

The unique solution of this equation at $\varepsilon$ is

$$\left.\frac{dL}{d\varepsilon}\right|_{\varepsilon=0} = A_{22}^{-1}L(0)(A_{11} - A_{12}L(0)) = A_{22}^{-2}A_{21}A_0.$$

Accordingly, (43) represents the first two terms of the MacLaurin series for $L(\varepsilon)$. $\qquad\square$

We remark that $L(\varepsilon)$ as defined in (43) is unique in the sense that even though $R(L, \epsilon)$ as given in (42) may have several real solutions, only one is approximated by (43).

The characteristic equation of (41) is equivalent to that for the system from (36) owing to the similarity transform between the systems. The block-triangular form of (36) admits a characteristic equation given by

$$\psi(s, \varepsilon) = \frac{1}{\varepsilon^m}\psi_s(s, \varepsilon)\psi_f(p, \varepsilon) = 0, \tag{45}$$

where

$$\psi_s(s, \varepsilon) = \det(sI - (A_{11} - A_{12}L(\varepsilon))) \tag{46}$$

is the characteristic polynomial of the slow subsystem, and

$$\psi_f(p, \varepsilon) = \det(pI - (A_{22} + \varepsilon A_{12}L(\varepsilon))) \tag{47}$$

is the characteristic polynomial of the fast subsystem in the timescale $p = s\varepsilon$. Consequently, $n$ of the eigenvalues of (36) denoted by $\{\lambda_1, \ldots, \lambda_n\}$ are the roots of the slow characteristic equation $\psi_s(s, \varepsilon) = 0$ and the rest of the eigenvalues $\{\lambda_{n+1}, \ldots, \lambda_{n+m}\}$ are denoted by $\lambda_i = \nu_j/\varepsilon$ for $i = n + j$ and $j \in \{1, \ldots, m\}$ where $\{\nu_1, \ldots, \nu_m\}$ are the roots of the fast characteristic equation $\psi_f(p, \varepsilon) = 0$.

The roots of $\psi_s(s, \varepsilon)$ at $\varepsilon = 0$, given by the solution to

$$\psi_s(s, 0) = \det(sI - (A_{11} - A_{12}L(0))) = 0, \tag{48}$$

are the eigenvalues of the matrix $A_0$ defined in (44) since $L(0) = A_{22}^{-1}A_{21}$ as shown in Lemma I.1. The roots of the fast characteristic equation at $\varepsilon = 0$, given by the solution to

$$\psi_f(p, 0) = \det(pI - A_{22}) = 0 \tag{49}$$

are the eigenvalues of the matrix $A_{22}$. The roots of the systems correspond to the conditions for a differential Stackelberg equilibrium, which thus gives the result.

We now proceed by characterizing how closely the eigenvalues of the system at $\varepsilon = 0$ approximate the eigenvalues of the system from (36) as $\varepsilon \to 0$.

If $\det(A_{22}) \neq 0$, then as $\varepsilon \to 0$, $n$ eigenvalues of the system given in (36) tend toward the eigenvalues of the matrix $A_0$ while the remaining $m$ eigenvalues of the system from (36) tend to infinity with the rate $1/\varepsilon$ along asymptotes defined by the eigenvalues of $A_{22}$ given as $\mathrm{spec}(A_{22})/\varepsilon$ as a result of the continuity of coefficients of the polynomials from (46) and (47) with respect to $\varepsilon$.

Now, consider the special (but generic) case in which the eigenvalues of $A_0$ are distinct and the eigenvalues of $A_{22}$ are distinct, but $A_0$ and $A_{22}$ may have common eigenvalues. Then, taking the total derivative of (45) with respect to $\varepsilon$ we have that

$$\frac{\partial \psi_s}{\partial s} \frac{ds}{d\varepsilon} + \frac{\partial \psi_s}{\partial \varepsilon} = 0$$

Now, observe that $\partial \psi_s / \partial s \neq 0$ since the eigenvalues of $A_0 = A_{11} - A_{12} A_{22}^{-1} A_{21}$ are distinct.[12] For each $i = 1, \ldots, n$, this gives us a well-defined derivative $ds/d\varepsilon$ (by the implicit mapping theorem) and hence, with $s(0) = \lambda_i(A_0)$, the $O(\varepsilon)$ approximation of $s(\varepsilon)$ follows directly. That is,

$$\lambda_i = \lambda_i(A_0) + O(\varepsilon), \; i = 1, \ldots, n_1$$

Similarly, taking the total derivative of $\psi_f(p, \varepsilon) = 0$ and again applying the implicit function theorem, we have

$$\lambda_{i+n_1} = \varepsilon^{-1}(\lambda_j(A_{22} + O(\varepsilon)), \; i = 1, \ldots, n_2$$

where we have used the fact that $p = s\varepsilon$.

## J  FURTHER DETAILS ON RELATED WORK

In this section, we provide further details on the discussion from Section A regarding the results presented by Jin et al. (2020) on the local stability of gradient descent-ascent with a finite timescale separation. The purpose of this discussion is to make clear that Proposition 27 from the work of Jin et al. (2020) does not disagree with the results we provide in Theorem 1 and Theorem 2 and is instead complementary. In what follows, we recall Proposition 27 of Jin et al. (2020) in separate pieces in the terminology of this paper and delineate its meaning from our results on the stability of gradient descent-ascent with a finite timescale separation.

To begin, we consider the component of Proposition 27 from Jin et al. (2020) which says that given any *fixed* and finite timescale separation $\tau > 0$, a zero-sum game can be constructed with a differential Stackelberg equilibrium that is not stable with respect to the continuous time limiting system of $\tau$-GDA given by the dynamics $\dot{x} = -\Lambda_\tau g(x)$.

**Proposition J.1** (Rephrasing of Jin et al. 2020, Proposition 27(a)). *For any fixed $\tau > 0$, there exists a zero-sum game $\mathcal{G} = (f, -f)$ such that $\mathrm{spec}(J_\tau(x^*)) \not\subset \mathbb{C}_+^\circ$ for a differential Stackelberg equilibrium $x^*$.*

We now explain the proof. Let us consider any $\epsilon > 0$ and the game

$$f(x, y) = -x^2 + 2\sqrt{\epsilon}xy - (\epsilon/2)y^2. \tag{50}$$

At the unique critical point $(x^*, y^*) = (0, 0)$, the Jacobian of the dynamics is given by

$$J_\tau(x^*, y^*) = \begin{bmatrix} -2 & 2\sqrt{\epsilon} \\ -2\tau\sqrt{\epsilon} & \tau\epsilon \end{bmatrix}.$$

Moreover, observe that $(x^*, y^*)$ is a differential Stackelberg equilibrium and not a differential Nash equilibrium since $D_1^2 f(x^*, y^*) = -2 \not\succ 0$, $-D_2^2 f(x^*, y^*) = \epsilon > 0$ and $\mathtt{S}_1(J(x^*, y^*)) = 2 > 0$. Finally, the spectrum of the Jacobian is

$$\mathrm{spec}(J_\tau(x^*, y^*)) = \left\{ \frac{-2 + \tau\epsilon \pm \sqrt{\tau^2\epsilon^2 - 12\tau\epsilon + 4}}{2} \right\}.$$

Let us now fix $\tau$ as any arbitrary positive value. Then, consider the game construction from (50) with $\epsilon = 1/\tau$. For the fixed choice of $\tau$ and subsequent game construction, we get that

$$\mathrm{spec}(J_\tau(x^*, y^*)) = \{ (-1 \pm i\sqrt{7})/2 \} \not\subset \mathbb{C}_+^\circ.$$

This in turn means the differential Stackelberg equilibrium is not stable with respect to the dynamics $\dot{x} = -\Lambda_\tau g(x)$ for the given choice of $\tau$. Since the choice of $\tau$ was arbitrary, this is a valid procedure

---

[12]Recall that having distinct eigenvalues is a generic condition for a matrix an $n_1 \times n_1$ matrix, though not explicitly required for the asymptotic results; its only a condition for the big-O approximation $\lambda_i = \lambda_i(A_0) + O(\varepsilon)$ for $i = 1, \ldots, n_1$ and $\lambda_i = \varepsilon^{-1}(\lambda_j(A_{22}) + O(\varepsilon))$ where $i = n_1 + j$ for $j = 1, \ldots, n_2$.

to generate a game with a differential Stackelberg equilibrium that is not stable with respect to $\dot{x} = -\Lambda_\tau g(x)$ given a choice of $\tau$ beforehand.

This result contrasts with that of Theorem 1 in the following fundamental way. In the proof of Proposition J.1, $\tau$ is fixed and then the game is constructed, whereas in Theorem 1 the game is fixed and then the conditions on $\tau$ given. To illustrate this point, consider the game construction from (50) with $\epsilon$ fixed to be an arbitrary positive value. It can be verified that $\text{spec}(J_\tau(x^*, y^*)) \subset \mathbb{C}_+^\circ$ for all $\tau > 2/\epsilon$. This means that given the differential Stackelberg equilibria in this game construction, there is indeed a finite $\tau^*$ such that the equilibrium is stable with respect to $\dot{x} = -\Lambda_\tau g(x)$ for all $\tau \in (\tau^*, \infty)$. Put concisely, Proposition J.1 is showing that there is exists a continuum of games for which a differential Stackelberg equilibrium is unstable with an improper choice of finite learning rate ratio $\tau$. On the other hand, Theorem 1 is proving that given a game with a differential Stackelberg equilibrium, there exists a range of suitable finite learning rate ratios such that the differential Stackelberg equilibrium is guaranteed to be stable.

We now move on to examining the portion of Proposition 27 from Jin et al. (2020) which says that given any *fixed* and finite timescale separation $\tau > 0$, a zero-sum game can be constructed with a critical point that is not a differential Stackelberg equilibrium which is stable with respect to the continuous time limiting system of $\tau$-GDA given by $\dot{x} = -\Lambda_\tau g(x)$.

**Proposition J.2** (Rephrasing of Jin et al. 2020, Proposition 27(b)). *For any fixed $\tau$, there exists a zero-sum game $\mathcal{G} = (f, -f)$ such that $\text{spec}(J_\tau(x^*)) \subset \mathbb{C}_+^\circ$ for a critical point $x^*$ satisfying $g(x^*) = 0$ that is not a differential Stackelberg equilibrium.*

In a similar manner as following Proposition J.1, we now explain the proof of Proposition J.2 and then contrast the result with Theorem 2. Again, consider any $\epsilon > 0$, along with the game construction

$$f(x, y) = x_1^2 + 2\sqrt{\epsilon}x_1 y_1 + (\epsilon/2)y_1^2 - x_2^2/2 + 2\sqrt{\epsilon}x_2 y_2 - \epsilon y_2^2. \tag{51}$$

At the unique critical point $(x^*, y^*) = (0, 0)$, the Jacobian of the dynamics is given by

$$J_\tau(x^*, y^*) = \begin{bmatrix} 2 & 0 & 2\sqrt{\epsilon} & 0 \\ 0 & -1 & 0 & 2\sqrt{\epsilon} \\ -2\tau\sqrt{\epsilon} & 0 & -\tau\epsilon & 0 \\ 0 & -2\tau\sqrt{\epsilon} & 0 & 2\tau\epsilon \end{bmatrix}$$

Observe that $(x^*, y^*)$ is neither a differential Nash equilibrium nor a differential Stackelberg equilibrium since $D_1^2 f(x^*, y^*) = \text{diag}(2, -1)$ and $-D_2^2 f(x^*, y^*) = \text{diag}(\epsilon, 2\epsilon)$ are both indefinite. The spectrum of the Jacobian is

$$\text{spec}(J_\tau(x^*, y^*)) = \left\{ \frac{2 - \tau\epsilon \pm \sqrt{\tau^2\epsilon^2 - 12\tau\epsilon + 4}}{2}, \frac{-1 + 2\tau\epsilon \pm \sqrt{4\tau^2\epsilon^2 - 12\tau\epsilon + 1}}{2} \right\}.$$

Now, fix $\tau$ as any arbitrary positive value, then consider the game construction from (51) with $\epsilon = 1/\tau$. For the fixed choice of $\tau$ and resulting game construction given the choice of $\epsilon$, we have that

$$\text{spec}(J_\tau(x^*, y^*)) = \{1 \pm i\sqrt{7}, 1 \pm i\sqrt{7}\} \subset \mathbb{C}_+^\circ.$$

This indicates that the non-equilibrium critical point is stable with respect to the dynamics $\dot{z} = -\Lambda_\tau g(z)$ where $z = (x, y)$ for the given choice of $\tau$. Similar to the proof of Proposition J.1, since the choice of $\tau$ was arbitrary, the procedure to generate a game with a non-equilibrium critical point that is stable with respect to $\dot{z} = -\Lambda_\tau g(z)$ is valid given a choice of $\tau$ beforehand.

The key distinction between Proposition J.2 and Theorem 2 is analogous to that between Proposition J.1 and Theorem 1. Indeed, the proof and result of Proposition J.2 rely on $\tau$ being fixed followed by the game being constructed. On the other hand, in Theorem 2 the game is fixed and then the conditions on $\tau$ given. To make this clear, consider the game construction from (51) with $\epsilon$ fixed to be an arbitrary positive value. It turns out that $\text{spec}(J_\tau(x^*, y^*)) \not\subset \mathbb{C}_+^\circ$ for all $\tau > 2/\epsilon$ since

$$\text{Re}\left( \frac{2 - \tau\epsilon \pm \sqrt{\tau^2\epsilon^2 - 12\tau\epsilon + 4}}{2} \right) < 0.$$

As a result, given the unique critical point of the game there is a finite $\tau_0$ such that the non-equilibrium critical point is not stable with respect to $\dot{x} = -\Lambda_\tau g(x)$ for all $\tau \in (\tau_0, \infty)$. In

summary, Proposition J.2 is showing that there is exists a continuum of games for which a non-equilibrium critical point is stable given an unsuitable choice of finite learning rate ratio $\tau$. In contrast, Theorem 2 is showing that given a game with a non-equilibrium critical point, there exists a range of finite learning rate ratios such that it is not stable.

To recap, the discussion in this section is meant to explicitly contrast Proposition 27 from the work of Jin et al. (2020) with Theorem 1 and Theorem 2 since they may potentially appear contradictory to each other without close inspection. The result of Jin et al. (2020) shows that (i) given a fixed finite learning ratio, there exists a game for with a differential Stackelberg equilibria that is not stable and (ii) given a fixed finite learning ratio, there exists a game with a non-equilibrium critical point that is stable. From a different perspective, we show that (i) given a fixed game and differential Stackelberg equilibrium, there exists a range of finite learning rate ratios for which the equilibrium is stable (Theorem 1) and (ii) given a fixed game and a non-equilibrium critical point, there exists a range of finite learning rate ratios for which the critical point is not stable (Theorem 2).

## K    EXPERIMENTS SUPPLEMENT

In this section we present several experiments not included in the body of the paper along with supplemental simulation results and details for the experiments presented in Section 5. We numerically investigate Example 1 in Section K.1 and a game similar to that from Example 2 in Section K.2. After that, we investigate a polynomial game with multiple equilibria in Section K.3. We study a torus game in Section K.4 and examine the connection between timescale separation and the region of attraction. Then, in Section K.5, we return to the Dirac-GAN game and consider the non-saturating objective function. In Section K.6, we explore a generative adversarial network formulation using the Wasserstein cost function with a linear generator and quadratic discriminator for the problem of learning a covariance matrix. We finish in Section K.7 by presenting further results and details on our experiments training generative adversarial networks on image datasets along with additional generative adversarial network experiments parameterized by neural networks with a mixture of Gaussians. Code for the experiments is included in the supplemental material.

### K.1    QUADRATIC GAME: TIMESCALE SEPARATION AND STACKELBERG STABILITY

We now revisit the game from Example 1 that demonstrated there exists differential Stackelberg equilibrium that are unstable for choices of the timescale separation $\tau$. To be clear, we repeat the game construction and some characteristics of the game. Let us consider the quadratic zero-sum game defined by the cost

$$f(x_1, x_2) = \frac{1}{2} \begin{bmatrix} x_1 \\ x_2 \end{bmatrix}^\top \begin{bmatrix} -v & 0 & -v & 0 \\ 0 & \frac{1}{2}v & 0 & \frac{1}{2}v \\ -v & 0 & -\frac{1}{2}v & 0 \\ 0 & \frac{1}{2}v & 0 & -v \end{bmatrix} \begin{bmatrix} x_1 \\ x_2 \end{bmatrix} \tag{52}$$

where $x_1, x_2 \in \mathbb{R}^2$ and $v > 0$. The unique critical point of the game given by $x^* = (x_1^*, x_2^*) = (0, 0)$ is a differential Stackelberg equilibrium. The spectrum of the Jacobian evaluated at the equilibrium is given by

$$\text{spec}(J_\tau(x^*)) = \left\{ \frac{v(2\tau + 1 \pm \sqrt{4\tau^2 - 8\tau + 1})}{4}, \frac{v(\tau - 2 \pm \sqrt{\tau^2 - 12\tau + 4})}{4} \right\}.$$

As mentioned in Example 1, it turns out that $\text{spec}(J_\tau(x^*) \subset \mathbb{C}_+^\circ$ only when $\tau \in (2, \infty)$. We remark that we computed $\tau^*$ using the theoretical construction from Theorem 1 and found that it recovered the precise value of $\tau^* = 2$ such that the equilibrium is stable for all $\tau \in (\tau^*, \infty)$ with respect to the dynamics $\dot{x} = -\Lambda_\tau g(x)$. In the experiments that follow, we consistently observe that the construction of $\tau^*$ from the theory is tight.

For this experiment, we select $v = 4$ and simulate $\tau$-GDA from the initial condition $(x_1^0, x_2^0) = (5, 4, 3, 2)$ with $\gamma_1 = 0.0005$ and $\tau \in \{2, 2.5, 3, 5, 10\}$. In Figures 4a and 4b, we show the trajectories of the players coordinate pairs $(x_{11}, x_{21})$ and $(x_{21}, x_{22})$, respectively. We observe that $\tau$-GDA cycles around the equilibrium with $\tau = 2$ since it is marginally stable with respect to the dynamics. For $\tau \in (2, \infty)$, the equilibrium is stable and $\tau$-GDA ends up converging to it at a rate

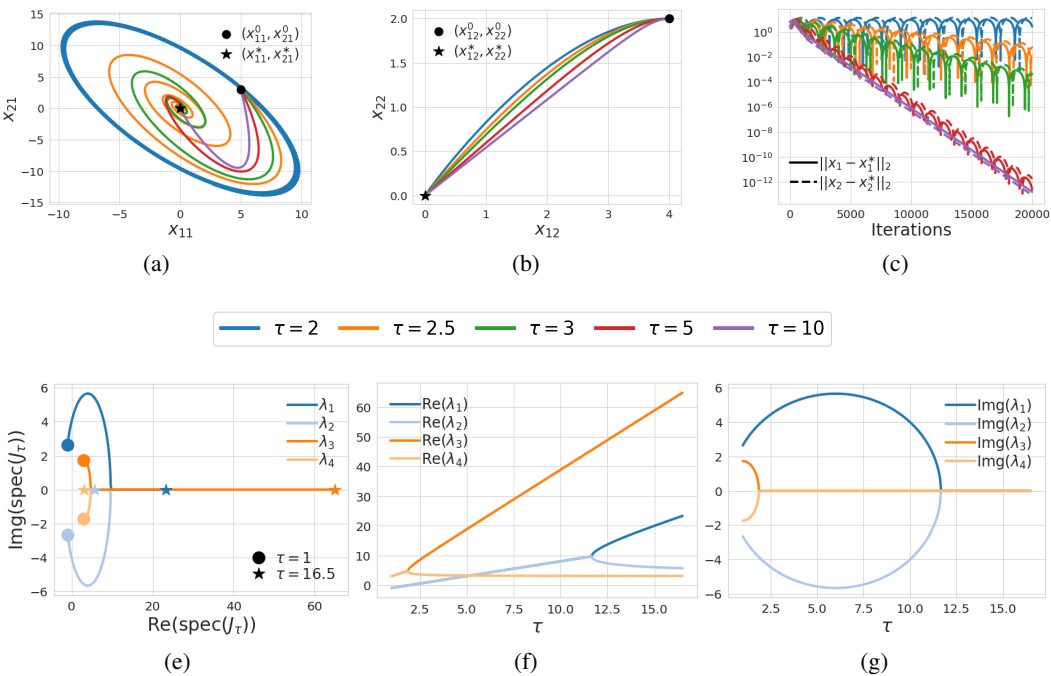

Figure 4: Experimental results for the quadratic game defined in (52) of Section K.1 and presented in Example 1. Figures 4a and 4b show trajectories of the players coordinate pairs $(x_{11}, x_{21})$ and $(x_{21}, x_{22})$ for a range of learning rate ratios, respectively. Figures 4c shows the distance from the equilibrium along the learning paths. Figures 4e, 4f, and 4g show the trajectories of the eigenvalues, the real parts of the eigenvalues, and the imaginary parts of the eigenvalues for the $J_\tau(x^*)$ as a function of the $\tau$, respectively.

that depends on the choice of $\tau$. We demonstrate how the convergence rate depends on the choice of $\tau$ in Figure 4c by showing the distance from the equilibrium along the learning path for each of the trajectories. The primary observation is that the cyclic behavior of $\tau$-GDA dissipates as $\tau$ grows and as a result the dynamics then rapidly converge to the equilibrium.

The behavior of the learning dynamics as a function of the timescale separation $\tau$ can be further explained by evaluating the eigenvalues of the game Jacobian at the equilibrium. We show the eigenvalues of the Jacobian at the equilibrium in several forms in Figures 4e, 4f, and 4g. Analyzing the spectrum, we are able to verify that for all $\tau \in (2, \infty)$ the equilibrium is indeed stable. Moreover, we see that the imaginary parts of the conjugate pairs of eigenvalues decay after $\tau = 1$ and $\tau = 6$, and then the eigenvalues of the conjugate pairs eventually become purely real at $\tau = 1.87$ and $\tau = 11.66$, respectively. After the eigenvalues of a conjugate pair become purely real, they split so that one of the eigenvalues asymptotically converges to an eigenvalue of $S_1(J(x^*))$ by moving back along the real line, while the other eigenvalue tends toward an eigenvalue of $-\tau D_2^2 f(x^*)$. This occurrence is exactly what was described in Section 3 as an immediate implication of Proposition I.1 when the eigenvalues of $S_1(J(x^*))$ and $\tau D_2^2 f(x^*)$ are distinct. The convergence rate is in fact limited by the eigenvalues splitting since as $\tau$ grows, the spectrum of the Jacobian is limited by the eigenvalues of the Schur complement which remain constant. A related open question centers on finding the worst case convergence rate as a function of the spectral properties of $S_1(J(x^*))$ and $D_2^2 f(x^*)$. Finally, the evolution of the eigenvalues as a function of the timescale separation $\tau$ demonstrates that the rotational dynamics in $\tau$-GDA vanish as the ratio between the magnitude of the real and imaginary parts of the eigenvalues grows.

### K.2 POLYNOMIAL GAME: TIMESCALE SEPARATION AND NON-EQUILIBRIUM STABILITY

We now return to a game similar to that from Example 2 with a non-equilibrium critical point which is stable without timescale separation and becomes unstable for a range of finite learning ratios with

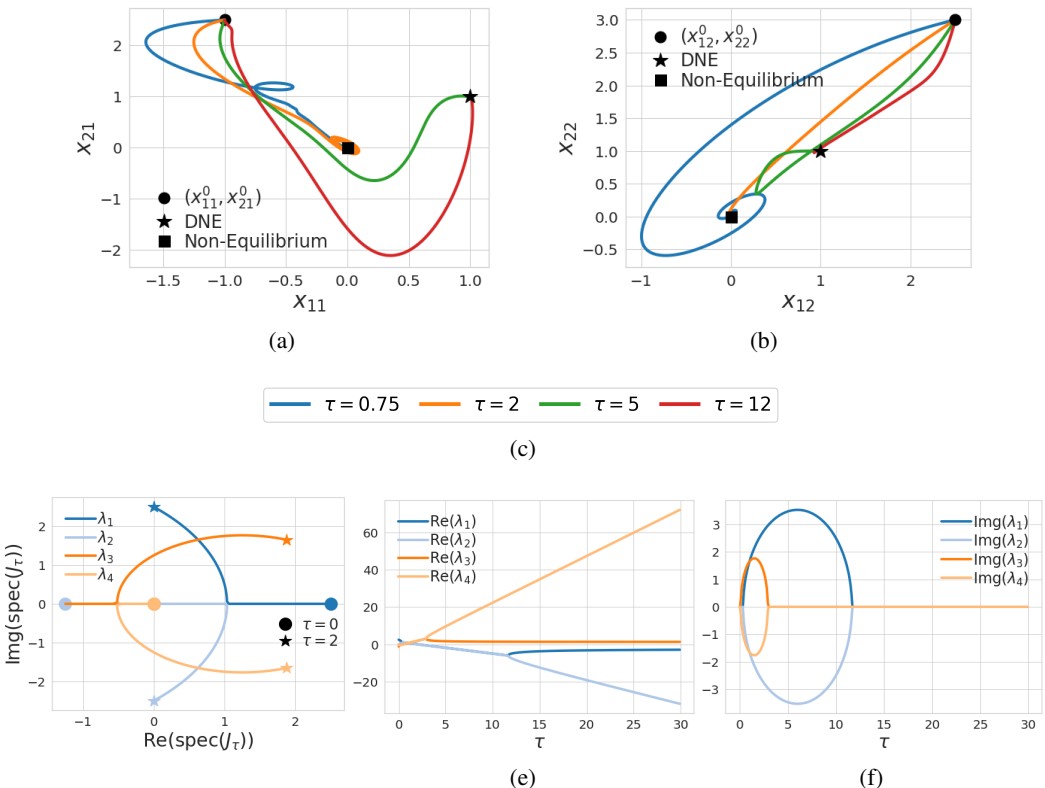

Figure 5: Experimental results for the polynomial game defined in (53) of Section K.2 and presented in Example 2. Figures 5a and 5b show trajectories of the players coordinate pairs $(x_{11}, x_{21})$ and $(x_{21}, x_{22})$ for a range of learning rate ratios, respectively. Figures 5d, 5e, and 5f show the trajectories of the eigenvalues, the real parts of the eigenvalues, and the imaginary parts of the eigenvalues for $J_\tau(x^*)$ as a function of the $\tau$, respectively where $x^*$ is the non-equilibrium critical point.

multiple equilibria in the vicinity. Consider a zero-sum game defined by the cost

$$
\begin{aligned}
f(x_1, x_2) = \frac{5}{4}\left(x_{11}^2 + 2x_{11}x_{21} + \frac{1}{2}x_{21}^2 - \frac{1}{2}x_{12}^2 + 2x_{12}x_{22} - x_{22}^2\right)(x_{11} - 1)^2 \\
+ x_{11}^2\left(\sum_{i=1}^2 (x_{1i} - 1)^2 - (x_{2i} - 1)^2\right).
\end{aligned}
\tag{53}
$$

This game has critical points at $(0, 0, 0, 0)$, $(1, 1, 1, 1)$, and $(-4.73, 0.28, -92.47, 0.53)$. Among the critical points, only $(1, 1, 1, 1)$ and $(-4.73, 0.28, -92.47, 0.53)$ are game-theoretically meaningful equilibrium. In fact, they are each differential Nash equilibrium and are locally stable for any choice of $\tau \in (0, \infty)$ as a result of Proposition F.1. On the other hand, the critical point $x^* = (0, 0, 0, 0)$ is neither a differential Nash equilibrium nor a differential Stackelberg equilibrium. However, $x^*$ is stable for $\tau \in (0, 2)$ and it is marginally stable for $\tau = 2$. In general, convergence to the non-equilibrium critical point $x^*$ in the presence of multiple game-theoretically meaningful equilibrium would be viewed as undesirable. In fact, this is precisely the type of critical point that sophisticated schemes for converging to only differential Nash equilibria or only differential Stackelberg equilibria seek to avoid (Adolphs et al., 2019; Fiez et al., 2020; Mazumdar et al., 2019; Wang et al., 2020). We show in this example that the simple inclusion of timescale separation in gradient descent-ascent is sufficient to avoid $x^*$ and instead converge to a differential Nash equilibrium.

Indeed, for all $\tau \in (2, \infty)$ the non-equilibrium critical point $x^*$ is unstable with respect to $\dot{x} = -\Lambda_\tau g(x)$. We simulate $\tau$-GDA from the initial condition $(x_1^0, x_2^0) = (-1.5, 2.5, 2.5, 3)$ with $\gamma_1 = 0.0005$ and $\tau \in \{0.75, 2, 5, 12\}$, where we use the superscript to denote the time index so as not to be confused with the multiple indexes for player choice variables. In Figures 5a and 5b, we show the trajectories of the players coordinate pairs $(x_{11}, x_{21})$ and $(x_{21}, x_{22})$, respectively. We observe that $\tau$-GDA converges to the non-equilibrium critical point $x^*$ with $\tau = 0.75$ as expected and the dynamics

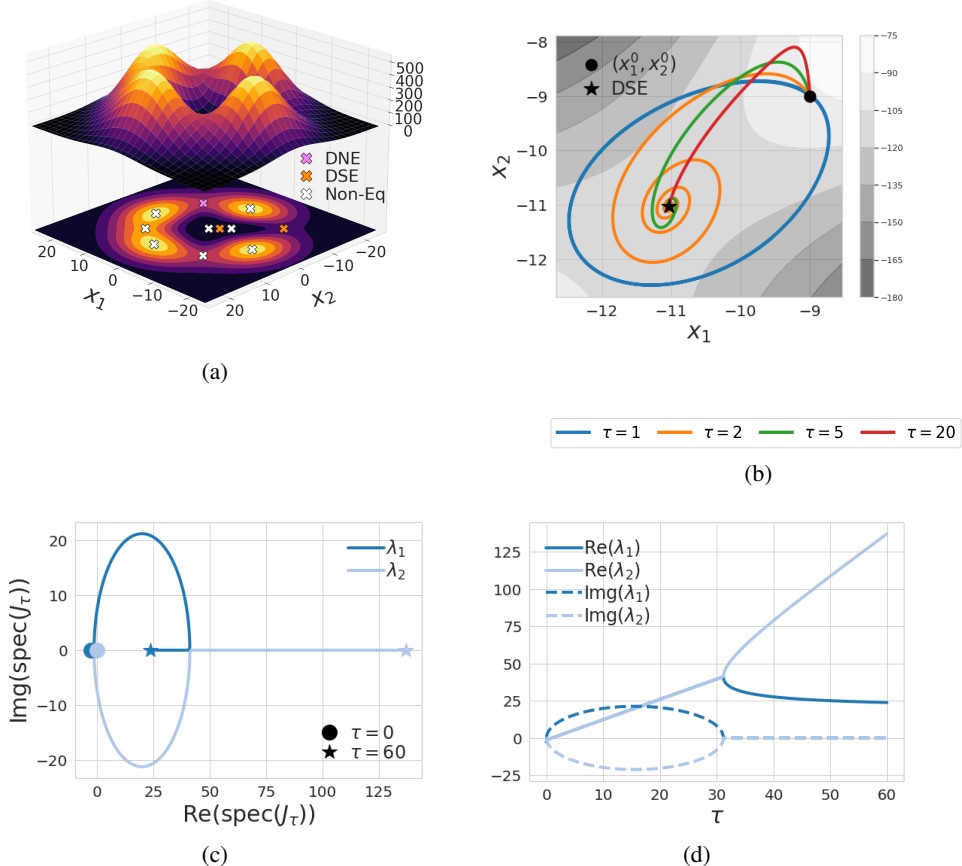

Figure 6: Experimental results for the polynomial game defined in (54) of Section K.3. Figures 6a provides a 3d view of the cost function $-f(x_1, x_2)$ along with the cost contours and critical point locations. Figure 6b shows trajectories of $\tau$-GDA for a range of learning rate ratios given an initialization around the differential Stackelberg equilibrium $(x_1^*, x_2^*) = (-11.03, -11.03)$. Figures 6c and 6d show the evolution of the eigenvalues from $J_\tau(x^*)$ as a function of $\tau$ where $x^*$ is the differential Stackelberg equilibrium $(x_1^*, x_2^*) = (-11.03, -11.03)$.

move near it and then cycle around it with $\tau = 2$ since the critical point becomes marginally stable. However, for $\tau = 5$ and $\tau = 12$, $\tau$-GDA avoids the non-equilibrium critical point since it becomes unstable and instead the dynamics converge to the nearby differential Nash equilibrium. We show the eigenvalues of the Jacobian at the non-equilibrium critical point $x^* = (0, 0, 0, 0)$ in several forms in Figures 5d–5f. Again, we observe that the eigenvalues quickly become purely real as $\tau$ grows and then they split, and asymptotically converge toward the eigenvalues of $S_1(J(x^*))$ and $-\tau D_2^2 f(x^*)$. Together, this example demonstrates that often there is a reasonable finite learning rate ratio such that non-meaningful critical points become unstable for $\tau$-GDA.

### K.3 POLYNOMIAL GAME: VECTOR FIELD WARPING AND REGION OF ATTRACTION

Consider a zero-sum game defined by the cost

$$f(x_1, x_2) = -e^{-\left(0.01x_1^2 + 0.01x_2^2\right)} \left((0.3x_1 + x_2^2)^2 + (0.3x_2 + x_1^2)^2\right). \tag{54}$$

The cost structure of this game is visualized in Figure 6a, where we present a three dimensional view of $-f(x_1, x_2)$ along with the cost contours and the locations of critical points. This game has eleven critical points including one differential Nash equilibrium and two differential Stackelberg equilibria that are not a differential Nash equilibrium. The critical points that are neither a

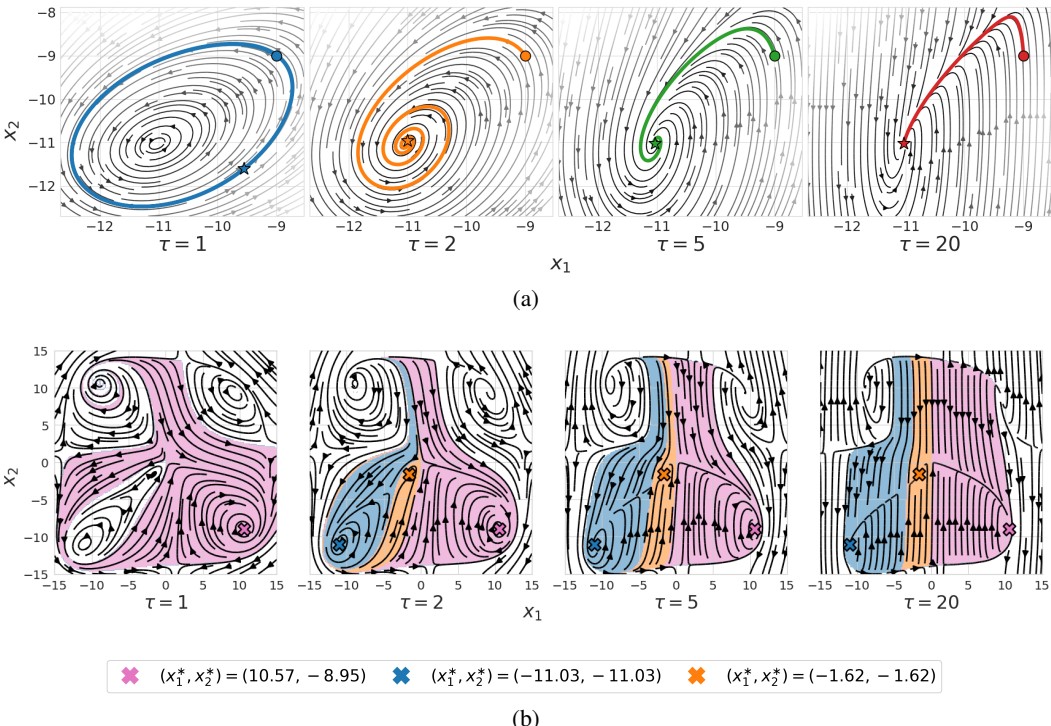

Figure 7: Experimental results for the polynomial game defined in (54) of Section K.3. In Figure 7a, we overlay the trajectories from Figure 6b produced by $\tau$-GDA onto the vector field generated by the choice of timescale separation selection $\tau$. The shading of the vector field is dictated by its magnitude so that lighter shading corresponds to a higher magnitude and darker shading corresponds to a lower magnitude. Figure 7b demonstrates the effect of timescale separation on the region of attractions around critical points by coloring points in the strategy space according to the equilibrium $\tau$-GDA converges. We remark that areas without coloring indicate where $\tau$-GDA did not converge in the time horizon.

differential Nash equilibrium nor a differential Stackelberg equilibrium are unstable for any choice of timescale separation $\tau$. The differential Nash equilibrium is at $(x_1, x_2) = (10.57, -8.95)$ and it is stable for all $\tau \in (0, \infty)$ by Proposition F.1. The differential Stackelberg equilibria are at $(x_1, x_2) = (-1.625, -1.625)$ and $(x_1^*, x_2^*) = (-11.03, -11.03)$; each is stable for all $\tau \in (1, \infty)$. We computed $\tau^*$ for the pair of differential Stackelberg equilibrium using the theoretical construction from Theorem 1 and observed that it properly recovered $\tau^* = 1$ for each equilibrium as the timescale separation such that the continuous time system is stable for all $\tau \in (\tau^*, \infty)$. Finally, we note that while the set of equilibrium follow a linear translation, this game is generic and the equilibria are in fact isolated.

In Figure 6b, we show the trajectories of $\tau$-GDA with $\gamma_1 = 0.0001$ and $\tau \in \{1, 2, 5, 20\}$ given the initialization $(x_1^0, x_2^0) = (-9, -9)$ near the differential Stackelberg equilibrium at $(x_1^*, x_2^*) = (-11.03, -11.03)$. Moreover, in Figure 7a, we overlay the trajectories on the vector field generated by the respective timescale separation parameters. As expected, the choice of $\tau = 1$ results in a trajectory that cycles around the equilibrium in a closed curve since it is marginally stable and $J_\tau(x^*)$ has purely imaginary eigenvalues. Notably, as $\tau$ grows, the cyclic behavior dissipates as the timescale separation reshapes the vector field until the trajectory moves near directly to the zero derivative line of the maximizing player and then follows a path along that line toward the equilibrium and converges rapidly. The eigenvalues of $J_\tau(x^*)$ as a function of $\tau$ are presented in Figures 6c and 6d. As was the case for the previous experiments, we observe that after the eigenvalues become purely real as $\tau$ grows, they then split and asymptotically converge toward the eigenvalues of $S_1(J(x^*))$ and $-\tau D_2^2 f(x^*)$. It is worth noting that much of the rotational behavior in the dynamics and vector field disappears as a result of timescale separation well before the eigenvalues become

purely real; this seems to occur after the timescale separation is such that the magnitude of the real part of the eigenvalues is greater than that of the imaginary part.

Finally, in Figure 7b, we demonstrate how the choice of timescale separation $\tau$ not only warps the vector field but also shapes the regions of attraction around critical points. The vector field is again shown for each $\tau \in \{1, 2, 5, 20\}$, but now zoomed out to include each of the equilibria. The colors overlayed on the vector field indicate the equilibria that the dynamics converge to given an initialization at that position. Positions in the strategy space without color did not converge to an equilibrium in the fixed horizon of 75000 iterations with $\gamma_1 = 0.001$. This is explained by the fact that the dynamics are not guaranteed to be globally convergent and may get stuck in limit cycles or may simply move slowly for a long time in flat regions of the optimization landscape. We produced this experiment by running $\tau$-GDA for a dense set of initial conditions chosen uniformly over the space of interest. It is clear from the experiment that the choice of timescale separation determines not only the stability of equilibria, but also has a fundamental impact on the equilibria the dynamics converge to from a given initial condition as a result of the warping of the vector field. As a concrete example, given an initialization of $(x_1, x_2) = (-10, -2)$, the dynamics with $\tau = 1$ converge to the differential Nash equilibria at $(x_1, x_2) = (10.57, -8.95)$. However, for any $\tau > 1$, the dynamics instead converge to the differential Stackelberg equilibrium at $(x_1, x_2) = (-11.03, -11.03)$ that is significantly closer to the initial condition. This example motivates future work on methods for obtaining accurate estimates of the regions of attraction around critical points and techniques to design $\tau$ in order to explicitly shape the region of attraction around an equilibrium of interest. We refer to the end of Section G for further discussion on potentially relevant analysis methods in this direction.

### K.4 LOCATION GAME ON THE TORUS

We use the example in this section to further study the role of timescale separation on the regions of attraction around critical points. Consider the zero-sum game defined by the cost

$$f(x_1, x_2) = -0.15 \cos(x_1) + \cos(x_1 - x_2) + 0.15 \cos(x_2). \tag{55}$$

This game can be interpreted as a location game on the torus. Specifically, the first player seeks to be far from the second player but near zero, while the second player seeks to be near the first player. This is a non-convex game on a non-convex strategy space. The critical points are given by the set[13]:

$$\{x : \ g(x) = 0\} = \{(0, 0), (\pi, \pi), (\pi, 0), (0, \pi), (-1.646, -1.496), (1.646, 1.496)\}.$$

The critical points $(0, 0)$ and $(\pi, \pi)$ are the only differential Stackelberg equilibrium and neither is a differential Nash equilibrium. The differential Stackelberg equilibrium at $(0, 0)$ is stable for all $\tau \in (\tau^*, \infty)$ where $\tau^* = 0.74$ and the differential Stackelberg equilibrium $(\pi, \pi)$ is stable for all $\tau \in (\tau^*, \infty)$ where $\tau = 1.35$. The rest of the critical points are unstable for any choice of $\tau$. We remark that we computed $\tau^*$ for each differential Stackelberg equilibrium using the construction from Theorem 1 in Section 3 and it again gave the exact value of $\tau^*$ such that the system is stable for all $\tau > \tau^*$.

In Figure 8a, we show the trajectories of $\tau$-GDA with $\gamma_1 = 0.001$ and $\tau \in \{1, 2, 5, 10\}$ given the initializations $(x_1^0, x_2^0) = (2, -1)$ and $(x_1^0, x_2^0) = (1.9, -2.1)$ overlayed on the vector field generated by the respective timescale separation parameters. We observe that as the timescale separation $\tau$ grows, the rotational dynamics in the vector field dissipate and the directions of movement become sharp. As we mentioned in previous examples, $\tau$-GDA moves directly to the zero line of $-D_2 f(x_1, x_2)$ and then along that line to an equilibrium given sufficient timescale separation. The warping of the vector field that occurs as a result of timescale separation impacts the equilibrium that the dynamics converge to from a fixed initial condition and the neighborhood on which $\tau$-GDA converges to an equilibrium. In other words, the *region of attraction* around critical points depends heavily on the timescale separation $\tau$.

To illustrate this fact, in Figure 8b we show the regions of attraction for each choice of timescale separation. The vector fields are again shown for each $\tau \in \{1, 2, 5, 10\}$, but now with colors overlayed indicating the equilibria that the dynamics converge to given an initialization at that position. This

---

[13]Note that because the joint strategy space is a torus, $(\pm\pi, \pm\pi) = (\mp\pi, \pm\pi)$, $(\pi, 0) = (-\pi, 0)$, and $(0, -\pi) = (0, \pi)$.

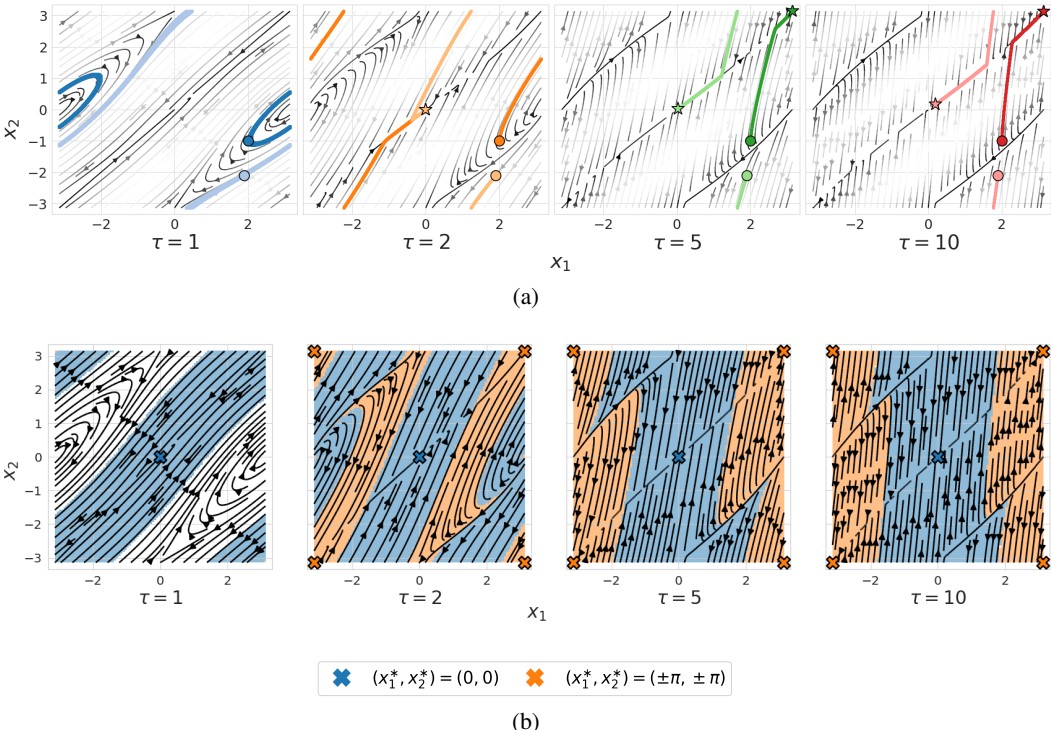

(a)

(b)

Figure 8: Experimental results for the torus game defined in (55) of Appendix K.4. In Figure 8a, we overlay multiple trajectories produced by $\tau$-GDA onto the vector field generated by the choice of timescale separation selection $\tau$. The shading of the vector field is dictated by its magnitude so that lighter shading corresponds to a higher magnitude and darker shading corresponds to a lower magnitude. Figure 8b demonstrates the effect of timescale separation on the regions of attraction around critical points by coloring points in the strategy space according to the equilibrium $\tau$-GDA converges. We remark that areas without coloring indicate where $\tau$-GDA did not converge in the time horizon.

experiment was generated by running $\tau$-GDA with a dense set of initial conditions chosen uniformly over the strategy space. Positions in the strategy space without color did not converge to an equilibrium in the fixed horizon of 20000 iterations with $\gamma_1 = 0.04$. This happens when $\tau$-GDA is not initialized in the local neighborhood of attraction around a stable equilibrium. For the choice of $\tau = 1$, $(0,0)$ is the only stable equilibrium. However, as demonstrated in Figure 8a, $\tau$-GDA fails to converge to the equilibrium from the initial conditions $(x_1^0, x_2^0) = (2, -1)$ and $(x_1^0, x_2^0) = (1.9, -2.1)$. This behavior is further demonstrated over the strategy space in Figure 8b and highlights the local nature of the guarantees since convergence is only assured given an initialization in a suitable local neighborhood around a stable critical point. On the other hand, $\tau$-GDA converges to an equilibrium from any initial condition for $\tau \in \{2, 5, 10\}$ as can be seen by Figure 8b. Notably, the equilibrium to which the learning dynamics converge depends on the timescale separation and initial condition. To give a concrete example, consider the initial conditions shown in Figure 8a of $(x_1^0, x_2^0) = (2, -1)$ and $(x_1^0, x_2^0) = (1.9, -2.1)$. For the initial condition $(x_1^0, x_2^0) = (2, -1)$, $\tau$-GDA converges to the equilibrium at $(0,0)$ for each $\tau \in \{2, 5, 10\}$. Yet, for the initial condition $(x_1^0, x_2^0) = (1.9, -2.1)$, $\tau$-GDA converges to the equilibrium at $\{(0,0), (\pi, \pi), (\pi, \pi)\}$ for the respective choices of $\tau \in \{2, 5, 10\}$. In other words, the region of attraction around the critical points changes so that from a fixed initial condition $\tau$-GDA may converge to distinct equilibrium depending on the initial condition. From Figure 8b, we see that the region of attraction around $(x_1^0, x_2^0) = (1.9, -2.1)$ grows from $\tau = 1$ to $\tau = 2$ and $\tau = 4$, but then shrinks at $\tau = 10$. This example highlights that timescale separation has a fundamental impact on the region of attraction around critical points and as $\tau$ grows it is possible for the region of attraction around an equilibrium to shrink. Collectively, this motivates explicit methods for trying to shape the region of attraction around desirable equilibria.

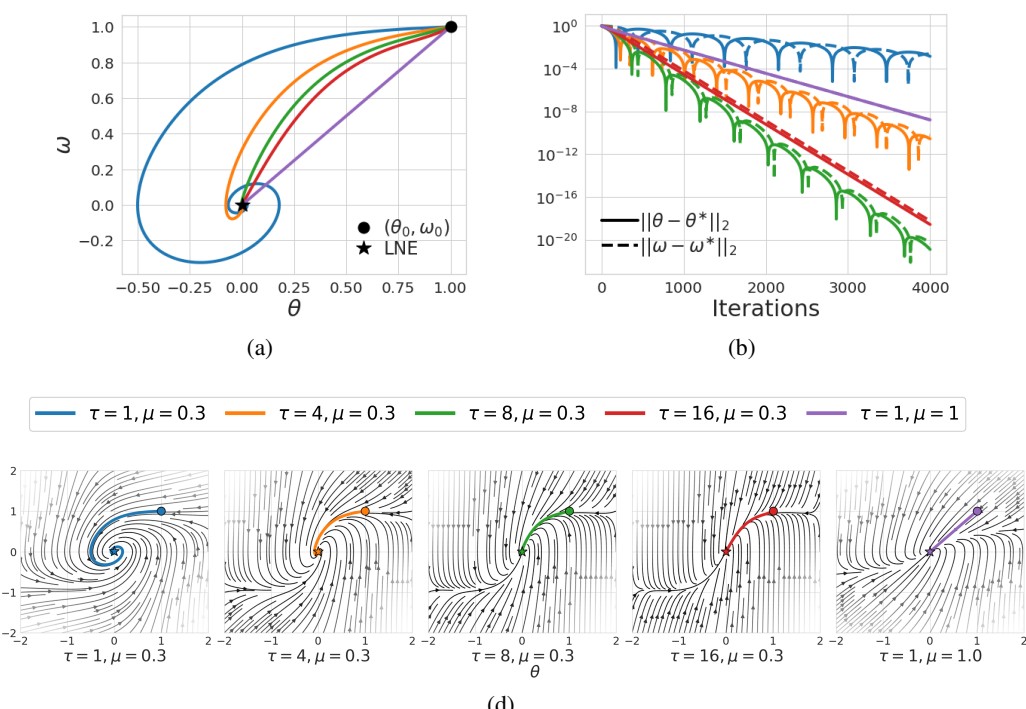

Figure 9: Experimental results for the Dirac-GAN game defined in (56) of Appendix K.5. Figure 9a shows trajectories of $\tau$-GDA for $\tau \in \{1, 4, 8, 16\}$ with regularization $\mu = 0.3$ and $\tau = 1$ with regularization $\mu = 1$. Figure 9b shows the distance from the equilibrium along the learning paths. Figure 9d shows the trajectories of $\tau$-GDA overlaid on the vector field generated by the respective timescale separation and regularization parameters. The shading of the vector field is dictated by its magnitude so that lighter shading corresponds to a higher magnitude and darker shading corresponds to a lower magnitude.

## K.5 DIRAC-GAN AND REGULARIZATION: NON-SATURATING FORMULATION

In Section 5, we presented experiments for the Dirac-GAN game studied by Mescheder et al. (2017) using the original generative adversarial network formulation of Goodfellow et al. (2014). In this section, we revisit the Dirac-GAN game using the non-saturating generative adversarial network formulation also proposed by Goodfellow et al. (2014). Recall that the zero-sum game which arises from the original objective with regularization $\mu > 0$ is defined by the cost

$$f(\theta, \omega) = \ell(\theta\omega) + \ell(0) - \frac{\mu}{2}\omega^2.$$

As discussed in Section 5, the unique critical point of the game is $(\theta^*, \omega^*) = (0, 0)$ and it corresponds to the local Nash equilibrium of the unregularized game and a differential Stackelberg equilibrium of the regularized game. Moreover, the equilibrium is stable with respect to the continous time dynamics for all $\tau > 0$ and $\mu > 0$ so that the discrete time update $\tau$-GDA converges with a suitable learning rate $\gamma_1$.

The non-saturating generative adversarial network formulation proposed by Goodfellow et al. (2014) in the context of the Dirac-GAN game corresponds to player 1 maximizing $\ell(-\theta\omega)$ instead of minimizing $\ell(\theta\omega)$. This results in the general-sum game defined by the costs

$$(f_1(\theta, \omega), f_2(\theta, \omega)) = (-\ell(-\theta\omega) + \ell(0) - \frac{\mu}{2}\omega^2, -\ell(\theta\omega) - \ell(0) + \frac{\mu}{2}\omega^2). \tag{56}$$

As shown by Mescheder et al. (2018), the unique critical point of the game remains at $(\theta^*, \omega^*) = (0, 0)$. Moreover, it can be observed that $J_\tau(\theta^*, \omega^*)$ in this formulation is identical to the game

Jacobian for the Dirac-GAN, which is given by

$$J_\tau(\theta^*, \omega^*) = \begin{bmatrix} 0 & \ell'(0) \\ -\tau\ell'(0) & \tau\mu \end{bmatrix},$$

(57)

so this game is locally equivalent to the zero-sum game that arises from the original objective proposed by Goodfellow et al. (2014). This is despite the fact that the non-saturating objective was motivated by global concerns (vanishing gradients early in the training process) rather than local considerations. In Figure 9 we present experiments with $\tau$-GDA for the regularized Dirac-GAN game with the non-saturating objective and $\ell(t) = -\ell(1 + \exp(-t))$. We observe similar behavior as the experiments with the standard objective and refer back to Section 5 for the insights we draw from the simulation. This experiment is primarily included for completeness and to motivate our use of the non-saturating objective in the generative adversarial networks experiments we perform on image datasets in Section 5.

### K.6 GENERATIVE ADVERSARIAL NETWORK: LEARNING A COVARIANCE MATRIX

We now consider a generative adversarial network formulation presented by Daskalakis et al. (2018) for learning a covariance matrix. This is a simple example with degeneracies much like the Dirac-GAN game, but it can be generalized to arbitrary dimensional strategy spaces and has served as a benchmark for comparing convergence rates in a number of recent papers on learning in games. Often, the example is used to show that gradient descent-ascent cycles and converges slowly. However, by and large, timescale separation is not considered. We show that gradient descent-ascent converges fast in this game with suitable timescale separation and further explore the interplay between timescale separation, regularization, and rate of convergence. We primarily follow the notation of Daskalakis et al. (2018) when describing the problem.

The objective of this problem is to learn a covariance matrix using the Wasserstein GAN formulation. The real data $x$ is drawn from a mean-zero multivariate normal distribution with an unknown covariance matrix $\Sigma$. The generator is restricted to be a linear function of the random input noise $z \sim \mathcal{N}(0, I)$ and is of the form $G_V(z) = Vz$. The discriminator is restricted to the set of all quadratic functions, which we represent by $D_W(x) = x^\top W x$. The parameters of the generator and the discriminator are given by $W \in \mathbb{R}^{d \times d}$ and $V \in \mathbb{R}^{d \times d}$, respectively. For the given generator and discriminator classes the Wasserstein GAN game is defined by the cost

$$f(V, W) = \mathbb{E}_{x \sim \mathcal{N}(0, \Sigma)}[x^\top W x] - \mathbb{E}_{z \sim \mathcal{N}(0, I)}[z^\top V^\top W V z].$$

As shown by Daskalakis et al. (2018), the cost function can be simplified to be expressed as

$$f(V, W) = \sum_{i=1}^{d} \sum_{j=1}^{d} W_{ij} \Big( \Sigma_{ij} - \sum_{k=1}^{d} V_{ik} V_{jk} \Big).$$

With this cost, the individual gradients for gradient descent-ascent are given by

$$g(V, W) = (-(W + W^\top)V, -(\Sigma - VV^\top)).$$

From the individual gradients, it is clear that the critical points of the game are given by $(V, W)$ such that $VV^\top = \Sigma$ and $W + W^\top = 0$. Moreover, given the form of $g(V, W)$, the game Jacobian at any critical point $(V^*, W^*)$ is of the form

$$J_\tau(V^*, W^*) = \begin{bmatrix} 0 & D_{12}f(V^*, W^*) \\ -\tau D_{12}^\top f(V^*, W^*) & 0 \end{bmatrix}.$$

Consequently, the eigenvalues of the game Jacobian are purely imaginary and the critical points are not stable. To fix this problem, Daskalakis et al. (2018) regularized both the generator and discriminator. We only regularize the discriminator in this example. The cost function of the zero-sum game with regularization $\mu > 0$ is given by

$$f(V, W) = \sum_{i=1}^{d} \sum_{j=1}^{d} W_{ij} \Big( \Sigma_{ij} - \sum_{k=1}^{d} V_{ik} V_{jk} \Big) - \frac{\mu}{2} \text{Tr}(W^\top W).$$

(58)

The individual gradients for gradient descent-ascent in this regularized game are then

$$g(V, W) = (-(W + W^\top)V, -(\Sigma - VV^\top) + \frac{\mu}{2}W).$$

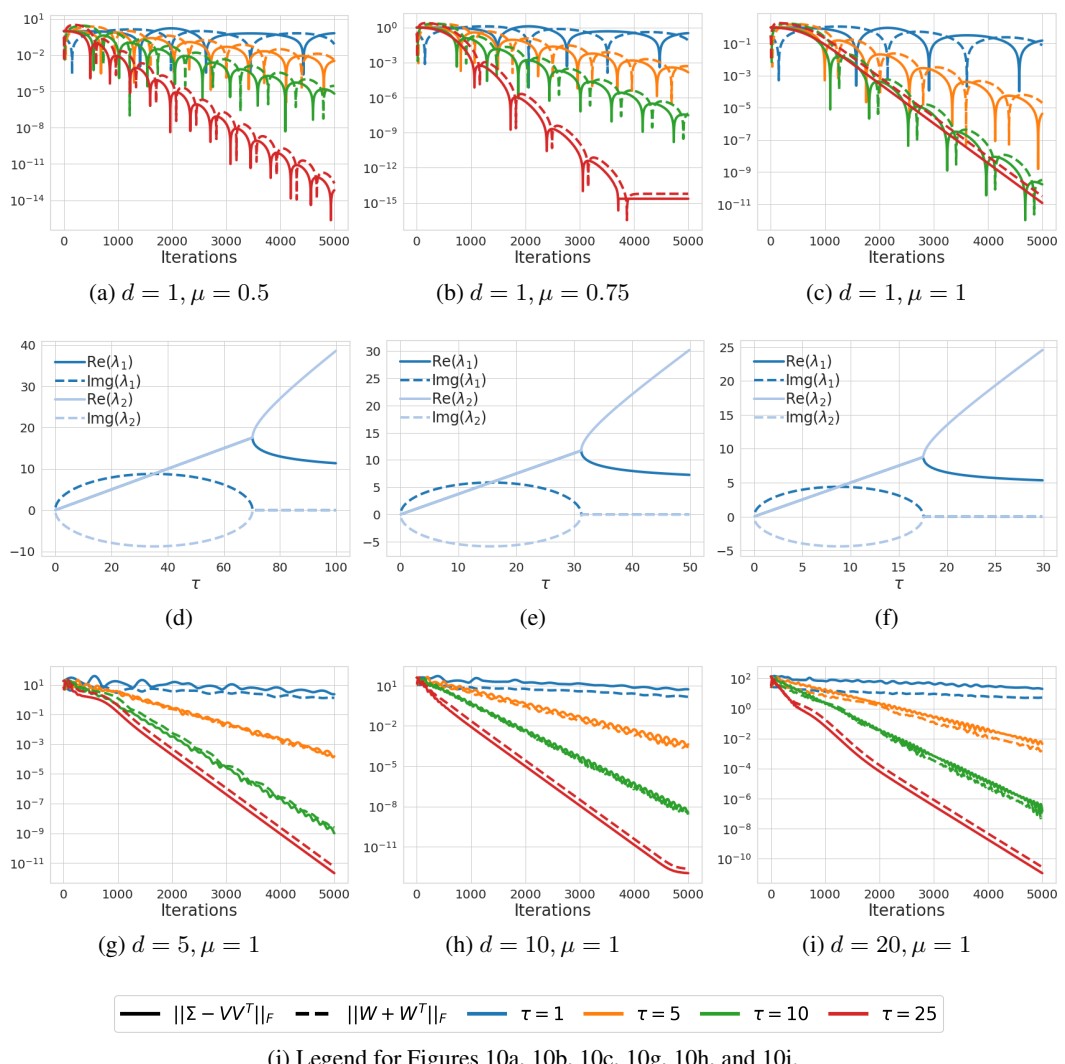

(a) $d = 1, \mu = 0.5$

(b) $d = 1, \mu = 0.75$

(c) $d = 1, \mu = 1$

(d)

(e)

(f)

(g) $d = 5, \mu = 1$

(h) $d = 10, \mu = 1$

(i) $d = 20, \mu = 1$

(j) Legend for Figures 10a, 10b, 10c, 10g, 10h, and 10i.

Figure 10: Experimental results for the generative adversarial network formulation for learning a covariance matrix defined by the cost from (58) of Section K.6. Figures 10a, 10b, and 10c show the distance from the equilibrium along the learning paths of $\tau$-GDA with $d = 1$. Figures 10d, 10e, and 10f show the trajectories of the eigenvalues of $J_\tau(x^*)$ as a function of the $\tau$, respectively. Figures 10g, 10h, and 10i show the distance from the equilibrium along the learning paths of $\tau$-GDA with $d = 5, 10, 20$.

We begin by considering the simplest form of this problem, which is that $d = 1$. The critical points with this restriction are $(V^*, W^*) = (\sigma, 0)$ and $(V^*, W^*) = (-\sigma, 0)$ and the game Jacobian evaluated at them is

$$J_\tau(V^*, W^*) = \begin{bmatrix} 0 & -2\sigma \\ 2\tau\sigma & \tau\mu \end{bmatrix}.$$

Each critical point is a local Nash equilibrium of the unregularized game and a differential Stackelberg equilibrium of the regularized game since $-D_2^2 f(V^*, W^*) = \mu > 0$ and $\mathbb{S}_1(J(V^*, W^*)) = 4\sigma^2/\mu > 0$. Furthermore, $\mathrm{spec}(J_\tau(V^*, W^*)) = \{(\tau\mu \pm \sqrt{\tau^2\mu^2 - 16\tau\sigma^2})/2\}$ so that each critical point is stable for all $\tau \in (0, \infty)$ and $\mu \in (0, \infty)$ since $\mathrm{spec}(J_\tau(\theta^*, \omega^*)) \subset \mathbb{C}_+^\circ$. Thus, given a suitably chosen learning rate $\gamma_1$, the discrete time update $\tau$-GDA locally converges to an equilibrium. For this reason, we focus on studying the rate of convergence for the problem as a function

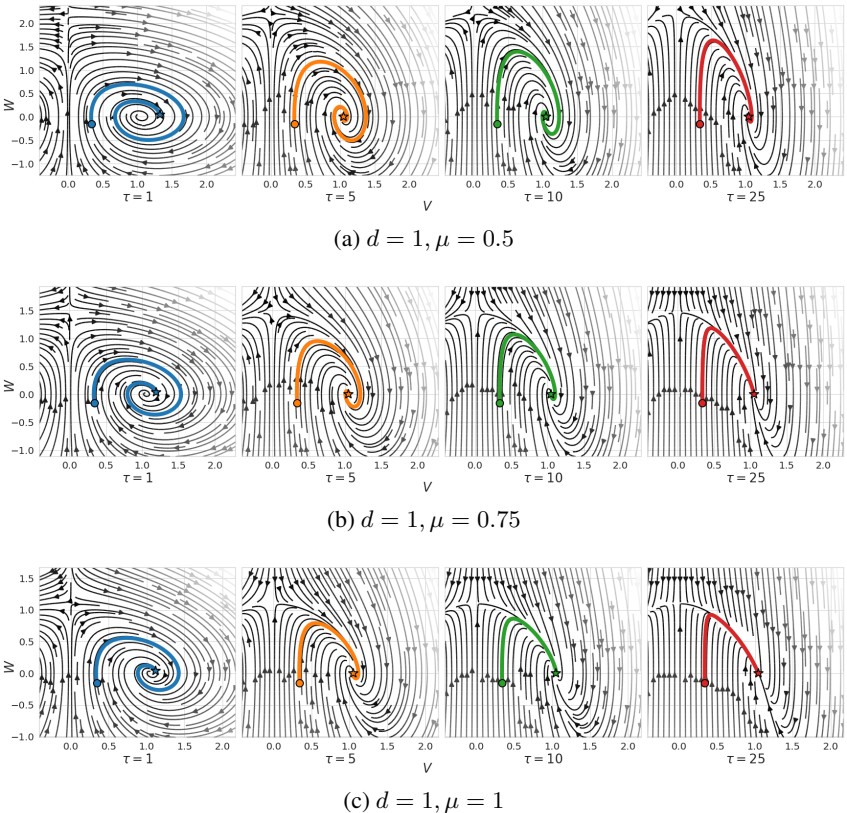

(a) $d = 1, \mu = 0.5$

(b) $d = 1, \mu = 0.75$

(c) $d = 1, \mu = 1$

Figure 11: Experimental results for learning a covariance matrix defined by the cost from (58) of Section K.6. We overlay the trajectories produced by $\tau$-GDA onto the vector field generated by the choices of $\tau$ and $\mu$. The shading of the vector field is dictated by its magnitude so that lighter shading corresponds to a higher magnitude and darker shading corresponds to a lower magnitude.

of timescale separation and regularization. Figures 10a, 10b, and 10c show the distance from an equilibrium along the learning path of $\tau$-GDA with $\tau \in \{1, 5, 10, 25\}$ given a fixed initial condition with learning rate $\gamma_1 = 0.001$ and regularization $\mu \in \{0.5, 0.75, 1\}$, respectively. Moreover, Figures 10d, 10e, and 10f show the trajectories of the eigenvalues for $J_\tau(V^*, W^*)$ as a function of $\tau$ for the regularization parameters $\mu \in \{0.5, 0.75, 1\}$. Finally, Figures 11a, 11b, and 11c show the trajectories of $\tau$-GDA overlayed on the vector field generated by the respective timescale separation and regularization parameters.

From the eigenvalue trajectories, we see that as $\mu$ grows, the eigenvalues become purely real at a smaller value of $\tau$. Moreover, as $\mu$ increases, the magnitude of the real and imaginary parts of the eigenvalues decreases. We observe the effect of this on the convergence, where the dynamics do not cycles as much for larger $\mu$. Again, we see the trade-off between timescale separation, regularization, and convergence. For example, despite the eigenvalues being purely real with $\mu = 1$ and $\tau = 25$ so that there is no rotational dynamics, the convergence is slower than for $\mu = 0.75$ where there is some non-zero imaginary piece of the eigenvalues.

Figures 10g, 10h, and 10i show the distance from a critical point along the learning path of $\tau$-GDA with $\tau \in \{1, 5, 10, 25\}$ given a fixed initial condition with learning rate $\gamma_1 = 0.001$, regularization $\mu = 1$, and the dimension of the problem $d$ among the set $\{5, 10, 20\}$, respectively. The primary purpose of showing this set of results is simply to be clear that the behavior for $d = 1$, which is easier to explain and visualize, transfers over to higher dimensional formulations of this problem. This is to be expected since the problem dimension is not necessarily fundamental to the convergence rate, but rather it depends on the conditioning of $\Sigma$ and each $\Sigma$ was chosen so that the behavior was comparable for each choice of dimension.

### K.7 Generative Adversarial Networks Parameterized by Neural Networks

In this section, we provide a much more detailed discussion of our experiments training generative adversarial networks on image datasets than was included in Section 5 and also present more the results at greater depth. We also present experiments training a generative adversarial network with $\tau$-GDA to learn a mixture of Gaussians.

**Background.** The empirical benefits of training with a timescale separation have been documented previously. For example, Heusel et al. (2017) showed on a number of image datasets that a timescale separation between the generator and discriminator improves generation performance as measured by the Frechet Inception Distance (FID). Since then a significant number of papers have presented results training generative adversarial networks with timescale separation. Moreover, it is common in the literature for the discriminator to be updated multiple times between each update of the generator (Arjovsky et al., 2017). Indeed, it has been widely demonstrated that this heuristic improves the stability and convergence of the training process and locally it has a similar effect as including a timescale separation between the generator and discriminator. The disadvantage of this approach is that the number of gradient calls per generator update increases and consequently the convergence is then slower in terms of wall clock time when a similar effect could potentially be achieved by a learning rate separation between the generator and discriminator. We remark that it appears to be reasonably common for practitioners to fix a shared learning rate for the generator and discriminator along with a pre-selected number of discriminator updates per generator update and not thoroughly investigate the impact timescale separation has on the training process.

The goal of our generative adversarial network experiments is to reinforce the importance of the timescale separation between the generator and the discriminator as a hyperparameter in the training process, demonstrate how it changes the behavior along the learning path, and show that it is compatible with a number of common training heuristics. This is to say that our goal is not necessarily to show state-of-the art performance, but rather to perform experiments that allow us to make insights relevant to the theory in this paper. We remark that our empirical work on training generative adversarial networks is distinct from and complimentary to that of Heusel et al. (2017) in several ways. The theory given by Heusel et al. (2017) only applies to stochastic stepsizes, however in the experiments they implemented constant step sizes. We train with mini-batches and decaying stepsizes in our image dataset experiments, which does satisfy the theory we provide as detailed in Section H. Moreover, by and large, the experiments by Heusel et al. (2017) compare a fixed learning rate ratio between the generator and discriminator to multiple fixed shared learning rates for the generator and discriminator. In contrast, we fix a learning rate for the generator and explore the behavior of the training process as the timescale parameter $\tau$ is swept over a given range.

### K.7.1 Generative Adversarial Networks: Mixture of Gaussians

We now provide the results from training generative adversarial networks to learn a mixture of Gaussians. The underlying data distribution consists of Gaussian distributions configured in a circle arrangement with means given by $\mu = [\sin(\omega), \cos(\omega)]$ for $\omega \in \{k\pi/4\}_{k=0}^{7}$, each with covariance $\sigma^2 I$ where $\sigma^2 = 0.05$. Each sample of real data given to the discriminator is selected uniformly at random from the set of Gaussian distributions. We train the generator using latent vectors $z \in \mathbb{R}^{16}$ sampled from a standard normal distribution in each training batch. The batch size for each player in the game is 512. The network for the generator and discriminator contain two and one hidden layers respectively, each which contain 32 neurons and ReLU activation functions. The training objective is the non-saturating objective and we run experiments without and with the $R_1$ gradient penalty proposed by Mescheder et al. (2018) using parameter $\mu = 0.1$. The generator learning rate is fixed to be $\gamma_1 = 0.005$ and the discriminator learning rate is fixed as $\gamma_2 = \tau\gamma_1$ where we experiment with $\tau \in \{4, 8, 16, 32, 64, 100\}$. For each parameter choice (timescale separation $\tau$ and regularization $\mu$), the experiment is repeated with 50 random seeds. The training does not rely on any adaptive gradient methods (Adam, RMSprop, etc.) and is the 'vanilla' stochastic $\tau$-GDA dynamics. We evaluate the performance along the learning path by computing the KL-divergence between the generated data and the real data, where we sample 4096 data points from each.

The results of this experiment are presented in Figure 12. We show the mean of the KL-divergence and the standard error of the means across the runs along the learning path without ($\mu = 0$) and with regularization ($\mu = 0.1$) in Figures 12a and 12b, respectively. Moreover, Figures 12c and 12d

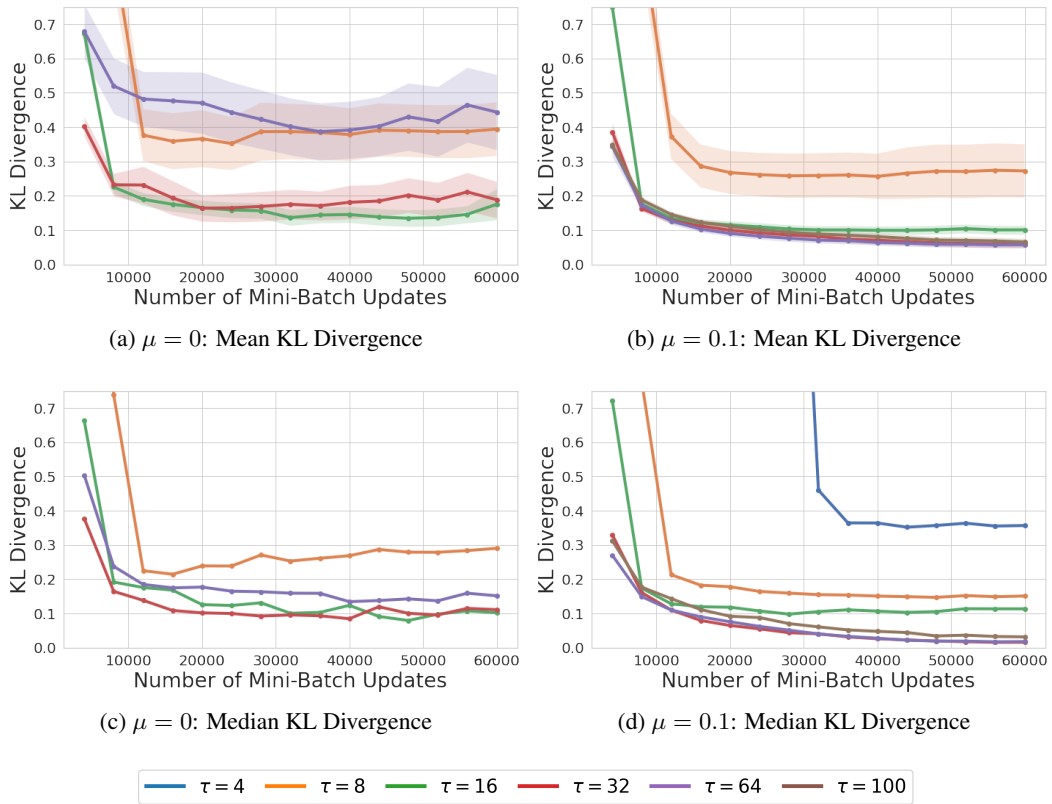

Figure 12: KL-divergence between generated and real data for a mixture of Gaussians.

show the medians of the KL-divergence across the runs without ($\mu = 0$) and with regularization ($\mu = 0.1$), respectively. From Figure 12a, we observe that the choices of $\tau = 4$ and $\tau = 100$ do not show on the plot since they perform poorly, which may be a result of equilibrium not being stable for $\tau = 4$ and numerically conditioning for $\tau = 100$. Furthermore, we see that timescale separation improves the results up to a reasonable timescale parameter and after which the performance degrades. Furthermore, Figure 12b reveals that the results are improved with regularization and we see that $\tau = 100$ ends up performing well, potentially since the regularization can alleviate some of the problems of numerical stability. In general, we draw similar conclusions from the median scores as reported in Figures 12c and 12d.

The primary purpose of this experiment is to train generative adversarial networks parameterized by neural networks using $\tau$-GDA without heuristics such as adaptive gradient methods or parameter averaging as is employed on the image dataset experiments from Section 5 and expanded on further in Appendix K.7.2. Notably, we see that consistent themes emerge that timescale separation improves convergence until hitting a limiting value and regularization can improve the rate of convergence but there is an interplay with the timescale separation.

### K.7.2 GENERATIVE ADVERSARIAL NETWORKS: IMAGE DATASETS

We now provide further details on the methods for the experiments training generative adversarial networks with image datasets presented in Section 5 along with more in-depth results.

**Methods.** We again note that we built our experiments based on the methods and implementations of Mescheder et al. (2018) and used the publicly available code from the paper available at https://github.com/LMescheder/GAN_stability. We effectively only changed the learning rates, retained multiple exponential averages at once, and modified the updates to be simultaneous in the code. In Figure 17 we provide the network architectures from our exper-

iments and in Figure 18 we include the hyperparameters that were selected. The architectures are analogous to that reported in Mescheder et al. (2018), but scaled down since we run experiments with $32 \times 32 \times 3$ images. For evaluation, we computed the Frechet Inception Distance using 10k samples from the real and generated data. For both experiments and across the set of hyperparameters we performed the evaluation using a fixed random noise vector to make for an equal comparison and a fixed set of real images which were randomly selected. The evaluation was done using the training data. We used the FID score implementation in pytorch available at `https://github.com/mseitzer/pytorch-fid`.

We train the generative adversarial networks with the non-saturating objective function and the $R_1$ gradient penalty proposed by Mescheder et al. (2018) with regularization parameters $\mu \in \{1, 10\}$. We note that the non-saturating objective results in a game that is not zero-sum, however it is commonly used in practice and under the realizable assumptions it can be locally equivalent to the zero-sum objective as discussed in Section K.5. The theory we provide does not apply to using RMSprop, but it is ubiquitous in practice for training generative adversarial networks and we are interested in exploring the interplay of timescale separation with common heuristics to understand if similar conclusions hold when using them as from the previous experiments regarding timescale separation with the 'vanilla' $\tau$-`GDA` dynamics. Moreover, we note that similarly Heusel et al. (2017) and Mescheder et al. (2018) also rely upon Adam or RMSprop in generative adversarial experiments. A final heuristic and hyperparameter that we explore in conjunction with the timescale separation $\tau$ is that of using an exponential moving average to produce the model that is evaluated. This means that at each update $k$, given that the parameters of the generator are given by $x_{1,k}$, the moving average $\bar{x}_k = x_{1,k}\beta + \bar{x}_{1,k-1}(1 - \beta)$ is kept where $\beta \in (0, 1)$. Experimental studies have shown that this heuristic can yield a significant improvement in terms of both the inception score and the FID (Gidel et al., 2019a; Yazici et al., 2019). The success of this method is thought to be a result of dampening both rotational dynamics and the noise from the randomness in the mini-batches of data.

**Experimental Results.** We run the training algorithm with the learning rate ratio $\tau$ belonging to the set $\{1, 2, 4, 8\}$ for CIFAR-10 and $\{1, 2, 4, 8, 16\}$ for CelebA along with the regularization parameter $\mu$ belonging to the set $\{1, 10\}$. For each choice of $\tau$ and $\mu$, we retain exponential moving averages of the generator parameters for $\beta \in \{0.99, 0.999, 0.9999\}$. The training process is repeated 2 times for each hyperparameter configuration in the CIFAR-10 experiments and the experiments with CelebA are simulated once. The performance is evaluated along the learning path at every 10,000 updates in terms of the FID score. We report the mean scores and the standard error of the mean over the repeated experiments for each dataset. The FID score is such that a lower score beats a higher score. The experiments are computationally intensive which limits the number of repeats of experiments that can be simulated, however, we observed that the scores were quite consistent between random seeds particularly with exponential averaging of the parameters. We run the experiments with $\mu = 1$ for 150k mini-batch updates and the experiments with $\mu = 10$ for 300k mini-batch updates.

The results for each dataset across the hyperparameter configurations are presented in numeric form in Figure 15. Figure 16 shows some generated samples selected at random for each dataset with the hyperparameter configuration that performed best in terms of the FID score at the end of the training process. We now describe the key observations from the experiments for each dataset.

**CIFAR-10.** The FID scores along the learning path for CIFAR-10 with $\mu = 10$ and $\mu = 1$ are presented in Figures 13a and 13b, respectively. The corresponding scores in numeric form are given in Figures 15a, 15c, and 15e for $\mu = 10$ at 150k iterations and $\mu = 1$ at 150k and 300k iterations, respectively. To begin, we observe that the exponential moving average significantly improves performance, and of the parameters considered, $\beta = 0.9999$ performed best. This may be a result of removing noise as mentioned previously or potentially it could be from dampening oscillatory behavior in the dynamics. Moreover, we that timescale separation also has a significant impact on the FID score of the training process. Indeed, even selecting $\tau = 2$ versus $\tau = 1$ can yield an impressive performance gain. In this experiment for each regularization parameter, $\tau = 4$ converges fastest and performs the best. We see that $\tau = 2$ outperforms $\tau = 8$ when $\mu = 10$ and the relationship is flipped when viewing the evaluation at 150k updates with $\mu = 1$ and then returns back when looking at the evaluation at 300k updates. The choice of $\tau = 1$ performs the worst for each regularization parameter by a wide margin. Finally, observe that the performance with

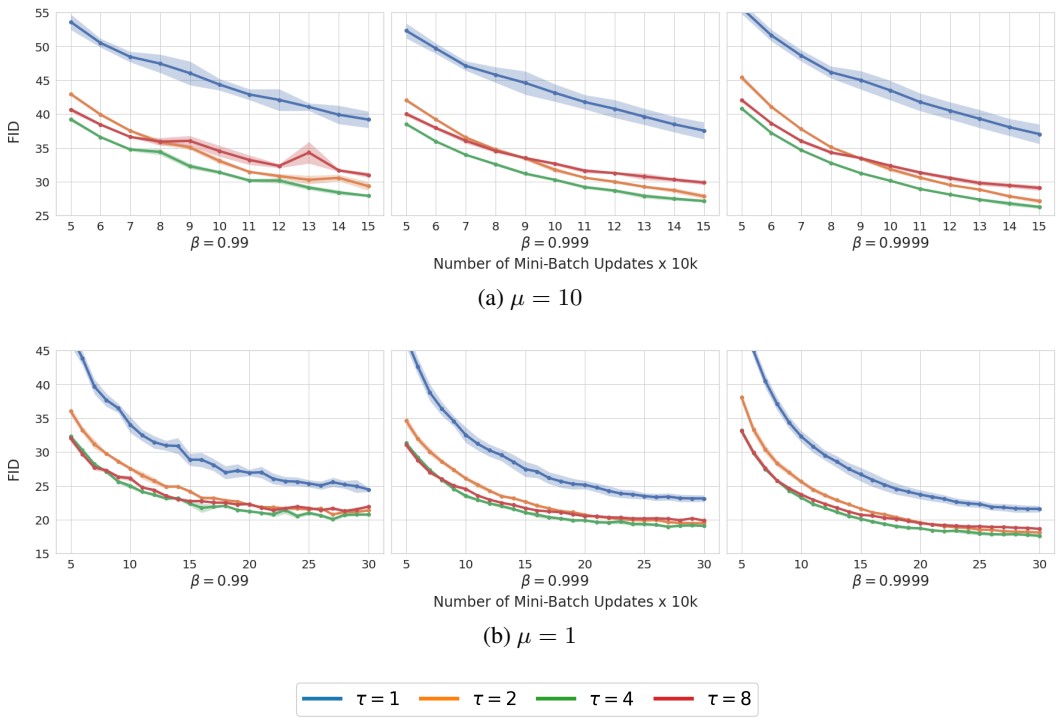

Figure 13: CIFAR-10 FID scores with regularization $\mu = 10$ in Figure 13a and $\mu = 1$ in Figure 13b.

regularization $\mu = 1$ is much better than with regularization $\mu = 10$ for each timescale separation parameter and exponential averaging parameter.

**CelebA.** The FID scores along the learning path for CIFAR-10 with $\mu = 10$ and $\mu = 1$ are presented in Figures 14a and 14b, respectively. The corresponding scores in numeric form are given in Figures 15b, 15d, and 15f for $\mu = 10$ at 150k iterations and $\mu = 1$ at 150k and 300k iterations, respectively. In this experiment we observe that while the exponential moving average helps performance, the gain is not as drastic as it was for CIFAR-10. It is not entirely clear if this is a consequence of the scores being lower or something fundamental to the optimization landscape and dynamics for the dataset. The timescale separation in combination with the regularization again has a major effect on the the FID score of the training process in this experiment. For regularization $\mu = 10$, the timescale parameters of $\tau = 4$ and $\tau = 8$ outperform $\tau = 1$, $\tau = 2$, and $\tau = 16$ by a wide margin, again highlighting that timescale separation can speed up convergence until a certain point where it can potentially slow it down owing to the effect on the conditioning of the problem locally. A similar trend can be observed with regularization $\mu = 1$, but with $\tau = 16$ performing closer to $\tau = 4$ and $\tau = 8$. For each regularization parameter and timescale parameter, we see that $\tau = 8$ performs the best. We again observe in this experiment that for all timescale separation parameters, the performance is significantly improved with regularization $\mu = 1$ as compared with $\mu = 10$. This once again highlights the importance of considering how this the hyperparameters of regularization and timescale interact and dictate the local convergence rates.

**Summary.** In summary, we took a well-performing method and implementation for training generative adversarial networks and demonstrated that timescale separation is an extremely important, and easy to implement, hyperparameter that is worth careful consideration since it can have a major impact on the convergence speed and final performance of the training process. Interestingly, the conclusions we draw are in line with the insights drawn from the simple Dirac-GAN experiment in Section 5 and from the mixture of Gaussian experiments from Appendix K.7.1. In particular, timescale separation only speeds up to convergence until hitting a limiting value and there is a key interplay between timescale separation, regularization, and convergence rate.

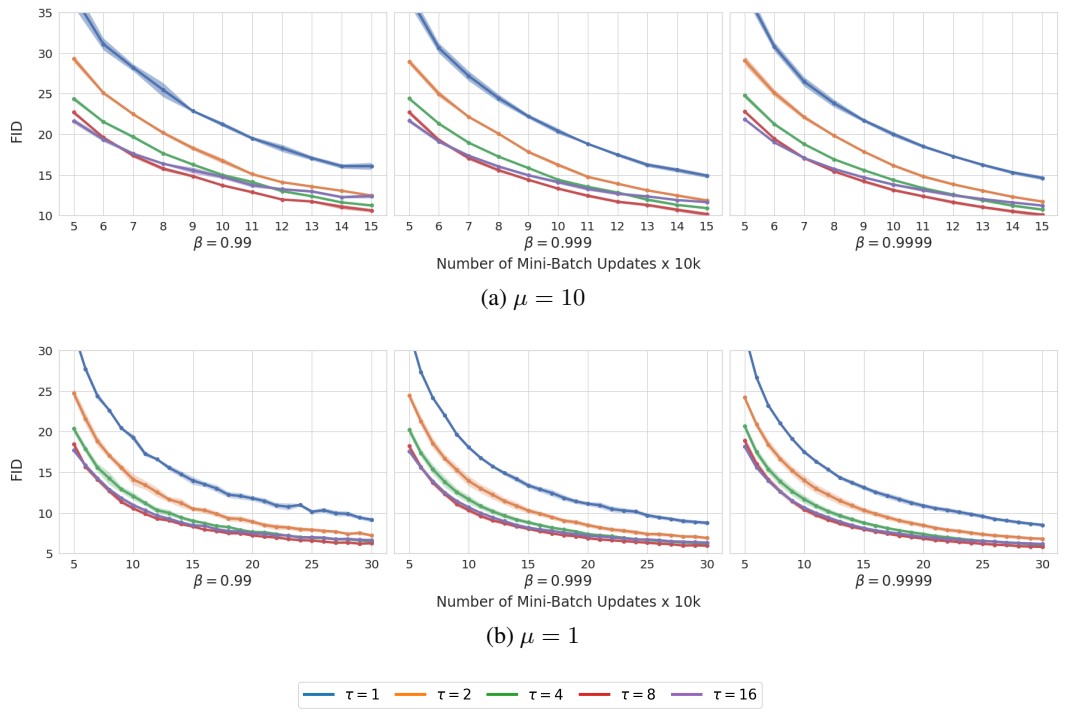

Figure 14: CelebA FID scores with regularization $\mu = 10$ in Figure 14a and $\mu = 1$ in Figure 14b.

| $\tau \backslash \beta$ | 0.99 | 0.999 | 0.9999 |
|---|---|---|---|
| 1 | $39.18 \pm 1.23$ | $37.55 \pm 1.27$ | $37.04 \pm 1.46$ |
| 2 | $29.33 \pm 0.65$ | $27.84 \pm 0.4$ | $27.14 \pm 0.34$ |
| 4 | $\mathbf{27.91 \pm 0.17}$ | $\mathbf{27.14 \pm 0.22}$ | $\mathbf{26.26 \pm 0.25}$ |
| 8 | $30.99 \pm 0.39$ | $29.86 \pm 0.34$ | $29.07 \pm 0.4$ |

(a) CIFAR-10 FID at 150k updates with $\mu = 10$

| $\tau \backslash \beta$ | 0.99 | 0.999 | 0.9999 |
|---|---|---|---|
| 1 | $16.08 \pm 0.44$ | $14.9 \pm 0.26$ | $14.63 \pm 0.26$ |
| 2 | $12.46 \pm 0.05$ | $11.85 \pm 0.11$ | $11.72 \pm 0.11$ |
| 4 | $11.24 \pm 0.13$ | $10.9 \pm 0.12$ | $10.72 \pm 0.13$ |
| 8 | $\mathbf{10.62 \pm 0.22}$ | $\mathbf{10.16 \pm 0.25}$ | $\mathbf{10.08 \pm 0.25}$ |
| 16 | $12.4 \pm 0.28$ | $11.64 \pm 0.05$ | $11.22 \pm 0.07$ |

(b) CelebA FID at 150k updates with $\mu = 10$

| $\tau \backslash \beta$ | 0.99 | 0.999 | 0.9999 |
|---|---|---|---|
| 1 | $28.87 \pm 0.92$ | $27.47 \pm 1.14$ | $26.69 \pm 1.02$ |
| 2 | $24.18 \pm 0.28$ | $22.65 \pm 0.15$ | $21.63 \pm 0.12$ |
| 4 | $\mathbf{22.38 \pm 0.36}$ | $\mathbf{21.05 \pm 0.21}$ | $\mathbf{20.12 \pm 0.13}$ |
| 8 | $22.74 \pm 0.15$ | $21.71 \pm 0.11$ | $20.72 \pm 0.08$ |

(c) CIFAR-10 FID at 150k updates with $\mu = 1$

| $\tau \backslash \beta$ | 0.99 | 0.999 | 0.9999 |
|---|---|---|---|
| 1 | $13.98 \pm 0.51$ | $13.38 \pm 0.33$ | $13.13 \pm 0.29$ |
| 2 | $10.51 \pm 0.28$ | $10.29 \pm 0.32$ | $10.33 \pm 0.36$ |
| 4 | $9.01 \pm 0.27$ | $8.83 \pm 0.25$ | $8.78 \pm 0.26$ |
| 8 | $\mathbf{8.32 \pm 0.1}$ | $\mathbf{8.04 \pm 0.14}$ | $\mathbf{7.98 \pm 0.15}$ |
| 16 | $8.47 \pm 0.04$ | $8.19 \pm 0.07$ | $8.13 \pm 0.07$ |

(d) CelebA FID at 150k updates with $\mu = 1$

| $\tau \backslash \beta$ | 0.99 | 0.999 | 0.9999 |
|---|---|---|---|
| 1 | $24.46 \pm 0.32$ | $23.14 \pm 0.5$ | $21.59 \pm 0.5$ |
| 2 | $21.37 \pm 0.11$ | $19.51 \pm 0.07$ | $18.08 \pm 0.19$ |
| 4 | $\mathbf{20.75 \pm 0.19}$ | $\mathbf{19.08 \pm 0.08}$ | $\mathbf{17.61 \pm 0.18}$ |
| 8 | $21.94 \pm 0.16$ | $19.87 \pm 0.1$ | $18.64 \pm 0.08$ |

(e) CIFAR-10 FID at 300k updates with $\mu = 1$

| $\tau \backslash \beta$ | 0.99 | 0.999 | 0.9999 |
|---|---|---|---|
| 1 | $9.16 \pm 0.29$ | $8.77 \pm 0.25$ | $8.52 \pm 0.22$ |
| 2 | $7.22 \pm 0.11$ | $6.91 \pm 0.19$ | $6.82 \pm 0.18$ |
| 4 | $6.47 \pm 0.13$ | $6.2 \pm 0.11$ | $6.07 \pm 0.10$ |
| 8 | $\mathbf{6.25 \pm 0.02}$ | $\mathbf{5.95 \pm 0.05}$ | $\mathbf{5.81 \pm 0.05}$ |
| 16 | $6.65 \pm 0.12$ | $6.35 \pm 0.11$ | $6.17 \pm 0.06$ |

(f) CelebA FID at 300k updates with $\mu = 1$

Figure 15: FID Scores on CIFAR-10 and CelebA.

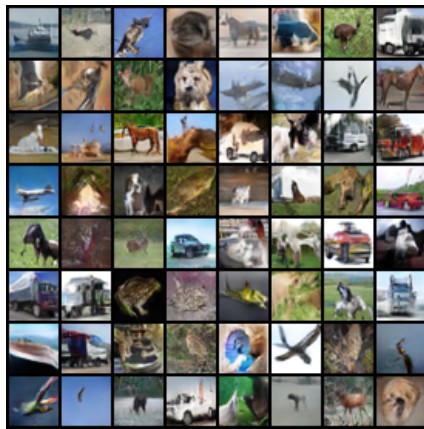

(a) CIFAR-10 generated sample images

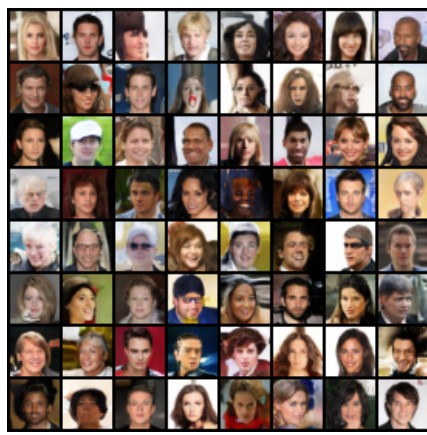

(b) CelebA generated sample images

Figure 16: Generated sample images with $\tau = 4$ and $\beta = 0.9999$

| Layer | Output Size | Filter |
|---|---|---|
| Fully Connected | $512 \cdot 4 \cdot 4$ | $256 \to 512 \cdot 4 \cdot 4$ |
| Reshape | $512 \times 4 \times 4$ | |
| Resnet-Block | $256 \times 4 \times 4$ | $512 \to 256 \to 256$ |
| NN-Upsampling | $256 \times 8 \times 8$ | |
| Resnet-Block | $128 \times 8 \times 8$ | $256 \to 128 \to 128$ |
| NN-Upsampling | $128 \times 16 \times 16$ | |
| Resnet-Block | $64 \times 16 \times 16$ | $128 \to 64 \to 64$ |
| NN-Upsampling | $64 \times 32 \times 32$ | |
| Resnet-Block | $64 \times 32 \times 32$ | $64 \to 64 \to 64$ |
| Conv2D | $3 \times 64 \times 64$ | $64 \to 3$ |

(a) Generator Network Architecture

| Layer | Output Size | Filter |
|---|---|---|
| Conv2D | $64 \times 32 \times 32$ | $3 \to 64$ |
| Resnet-Block | $64 \times 32 \times 32$ | $64 \to 64 \to 64$ |
| Avg-Pool2D | $64 \times 16 \times 16$ | |
| Resnet-Block | $128 \times 16 \times 16$ | $64 \to 64 \to 128$ |
| Avg-Pool2D | $128 \times 8 \times 8$ | |
| Resnet-Block | $256 \times 8 \times 8$ | $128 \to 128 \to 256$ |
| Avg-Pool2D | $256 \times 4 \times 4$ | |
| Resnet-Block | $512 \times 4 \times 4$ | $256 \to 256 \to 512$ |
| Fully Connected | $512 \cdot 4 \cdot 4$ | $512 \cdot 4 \cdot 4 \to 1$ |

(b) Discriminator Network Architecture

Figure 17: Network Architectures for GAN experiments on CIFAR-10 and CelebA

| Hyperparameter | Value(s) |
|---|---|
| Objective | NSGAN |
| Batch Size | 64 |
| Latent Distribution | $z \in \mathbb{R}^{256}$ |
| Generator Learning Rate | CIFAR-10: 0.0001; CelebA: 0.00005 |
| Timescale Separation $\tau$ | CIFAR-10: $\{1, 2, 4, 8\}$; CelebA: $\{1, 2, 4, 8, 16\}$ |
| Learning Rate Decay | $(1 + x)^{-0.005}$ |
| Optimizer | RMSprop |
| RMSprop Smoothing Constant $\alpha$ | 0.99 |
| RMSprop $\epsilon$ | $10^{-8}$ |
| Regularization $\mu$ | $\{1, 10\}$ |
| EMA Parameter $\beta$ | $\{0.99, 0.999, 0.9999\}$ |

Figure 18: Hyperparameters for GAN experiments on CIFAR-10 and CelebA

# L  ALTERNATIVE PROOF OF THEOREM 3 VIA $\tau^*$ CONSTRUCTION FROM THEOREM 1

In order to highlight the utility of Theorem 1 along with future directions of obtaining values of $\tau^*$ for structured games, we revisit the proof of Theorem 3 and derive the result directly from the construction. The purpose of this section is to illustrate that the structure of equilibria considered in Theorem 3 can be exploited to obtain the value of $\tau^*$ for the entire class of games using properties of the Kronecker product and sum.

Consider the generative adversarial network example under Assumption 1. Then, we can show precise from the construction that for all games of this class that $\tau^* = 0$ as long as $\mu > 0$. Fix a

regularization parameter $\mu > 0$. Then,

$$-J_{\tau,\mu}(x^*) = \begin{bmatrix} 0 & -B \\ \tau B^\top & \tau(\mu R + C) \end{bmatrix} = \begin{bmatrix} A_{11} & A_{12} \\ -\tau A_{12}^\top & \tau A_{22} \end{bmatrix}$$

Note that $A_{22} < 0$ and $A_{11}$ is the zero matrix and

$$S_1 = B(\mu R + C)^{-1} B^\top = A_{11} - A_{12} A_{22}^{-1} A_{12}^\top < 0$$

by Assumption 1. Now, we can conclude for this entire class of games that $\tau^* = 0$ precisely because $Q$ has no positive real roots. Indeed, to start recall from the proof that we need to find

$$\tau^* = \lambda_{\max}^+ (\underbrace{-M_2(A_{11} \otimes I_{n_2} + M_1)}_{Q}))$$

where

$$M_1 = -2H_{n_2}^+(-A_{12}^\top \otimes I_{n_2})(A_{22} \boxplus A_{22})^{-1}(A_{12} \otimes I_{n_2})H_{n_2}$$

and

$$M_2 = I_{n_1} \otimes A_{22}^{-1} - 2(I_{n_1} \otimes A_{22}^{-1} A_{12}^\top)H_{n_1}(S_1 \boxplus S_1)^{-1}H_{n_1}^+(I_{n_1} \otimes A_{12} A_{22}^{-1})$$

Plugging in the generative adversarial network parameters we see that we need

$$\tau^* = \lambda_{\max}^+(-M_2 M_1)$$

where

$$M_1 = 2H_{n_2}^+(B \otimes I_{n_2})((\mu R + C) \boxplus (\mu R + C))^{-1}(B^\top \otimes I_{n_2})H_{n_2} < 0$$

and

$$M_2 = I_{n_1} \otimes (\mu R + C)^{-1} - 2(I_{n_1} \otimes (\mu R + C)^{-1} B^\top)H_{n_1}$$
$$\cdot (B(\mu R + C)^{-1} B^\top \boxplus B(\mu R + C)^{-1} B^\top)^{-1}H_{n_1}^+(I_{n_1} \otimes B(\mu R + C)^{-1})$$

For later use, let us define

$$M_3 = -2(I_{n_1} \otimes (\mu R + C)^{-1} B^\top)H_{n_1}$$
$$\cdot (B(\mu R + C)^{-1} B^\top \boxplus B(\mu R + C)^{-1} B^\top)^{-1}H_{n_1}^+(I_{n_1} \otimes B(\mu R + C)^{-1})$$
$$= -2(I_{n_1} \otimes (\mu R + C)^{-1} B^\top)H_{n_1}(H_{n_1}^+(B(\mu R + C)^{-1} B^\top \otimes I_{n_1}$$
$$+ I_{n_1} \otimes B(\mu R + C)^{-1} B^\top)H_{n_1})^{-1}H_{n_1}^+(I_{n_1} \otimes B(\mu R + C)^{-1}).$$

Hence, as long as $M_2 \geq 0$ we know that $\tau^* = 0$. Here is where we can exploit the properties from Magnus (1988) on the $\boxplus$ and $\otimes$ operators. Specifically, let us use the property that $H_{n_1}^+(G \otimes I_{n_1})H_{n_1} = H_{n_1}^+(I_{n_1} \otimes G)H_{n_1} = \frac{1}{2}H_{n_1}^+(I_{n_1} \otimes G + G \otimes I_{n_1})H_{n_1}$ for a matrix $G$ which in turn implies that

$$(H_{n_1}^+(I_{n_1} \otimes G)H_{n_1})^{-1} = 2H_{n_1}^+(I_n \otimes G + G \otimes I_{n_1})^{-1}H_{n_1}$$

to get the following:

$$2H_{n_1}(B(\mu R + C)^{-1} B^\top \boxplus B(\mu R + C)^{-1} B^\top)^{-1}H_{n_1}^+ = (H_{n_1}^+(I_{n_1} \otimes B(\mu R + C)^{-1} B^\top)H_{n_1})^{-1}$$

Hence,

$$M_3 = -(I_{n_1} \otimes (\mu R + C)^{-1} B^\top)(I_{n_1} \otimes B(\mu R + C)^{-1} B^\top)^{-1}(I_{n_1} \otimes B(\mu R + C)^{-1})$$
$$= -(I_{n_1} \otimes (\mu R + C)^{-1} B^\top)(I_{n_1} \otimes (B(\mu R + C)^{-1} B^\top)^{-1})(I_{n_1} \otimes B(\mu R + C)^{-1})$$
$$= -(I_{n_1} \otimes (\mu R + C)^{-1} B^\top(B(\mu R + C)^{-1} B^\top)^{-1})(I_{n_1} \otimes B(\mu R + C)^{-1})$$
$$= -(I_{n_1} \otimes B^+)(I_{n_1} \otimes B(\mu R + C)^{-1})$$
$$= -(I_{n_1} \otimes (\mu R + C)^{-1})$$

which clearly holds since $B$ is full rank in general by Assumption 1, for which it is necessary that $n_2 \geq n_1/2$ by Proposition E.1.

