# OpenReview forum: "Local Convergence Analysis of Gradient Descent Ascent with Finite Timescale Separation"
_ICLR.cc/2021/Conference — ICLR 2021 Poster_

### Official Review · AnonReviewer4 · 2020-10-13
**Finite Timescale Separation for Gradient Descent Ascent**

**Rating:** 6
**Confidence:** 3

**Review:**

Motivated by the many applications of min-max optimization problems in Machine Learning, the authors examine the effect of using different learning rates for each player in Gradient Descent Ascent (GDA) for non-convex non-concave optimization problems. Prior work has already established that making the learning rate of one player infinitely larger than other player's learning rate alleviates the cycling problems of GDA and makes game-theoretically meaningful equilibria the only asymptotically stable fixed points. The main contribution of this work is that it proves that we can get the same stability guarantees while keeping the learning rates of both players finite. This is crucial for practical applications where using unbounded learning rates in not an option. The authors employ this result to prove a variety of local convergence results in both deterministic ans stochastic settings.

Pros:
1) The finite time scale separation is necessary in order to make the theoretical intuitions in prior work applicable for practical problems like trainings GANs.
2) The proof techniques used for this separation results are, to the best of my knowledge, significantly different and elaborate than the ones used in prior work (Jin et al., 2020).
3) The theoretical findings are complemented with empirical evaluations both on small min-max problems and on complex ones like training GAN architectures.

Cons:
Theorem 28 in the arxiv version of Jin et al. 2020 does not explicitly reference the existence of a finite time scale that satisfies their inclusion results. However,  it is clear from the proof of Theorem 28 on page 24 that such a finite time-scale separation exists even though they do not provide an explicit formula for it.  At least for one of the inclusion statements they explicitly mention that it holds for $\epsilon < \epsilon_0$  for some $\epsilon_0$ where $\epsilon$ corresponds to $1/\tau$.

Of course the result of the authors gives a more direct construction of the threshold $\tau^*$ by reducing the search for it to an eignenvalue problem. From a practical standpoint though, both results are existential. Neither proof approach gives particular intuition on how this time scale can be found in a computationally efficient way.  The added value of leveraging an array of mathematical tools to provide this explicit construction is unclear to me.

Given the above concern and that the convergence results essentially leverage the asymptotic stability properties provided by Theorems 1 and 2,  I am assigning a weak reject score. However, I am willing to substantially increase my score if the authors address the above concern.

---

> ### Author Response · Authors · 2020-11-12
> **Response to Reviewer 4**
>
> Thanks for your review. Please see the common response to you and R3 for a detailed comment regarding the primary con you have listed. We hope this can clear up your concern and make it clear that the results you refer to from past work can only be thought of in the asymptotic sense and we make a significant advance by providing a tight non-asymptotic construction. As you mentioned in the beginning of your review, this is extremely important for practical applications where using unbounded learning rates is not an option.
>
> Below we address more specific aspects of your review to hopefully provide further clarification.
>
> In your review you state:
>
> "Of course the result of the authors gives a more direct construction of the threshold $\tau^{\ast}$ by reducing the search for it to an eigenvalue problem. From a practical standpoint though, both results are existential."
>
> The construction of $\tau^{\ast}$ we provide is in fact not comparable to anything from [1]. This is because there is no construction given in [1] since it is an asymptotic result and is only stating local stability for GDA when $\tau\rightarrow \infty$. Please refer back to the general response to you and reviewer 3 for more details.
>
> Moreover, our result is not only existential, but it is also constructive. From a practical standpoint, this is major difference and is important. From an existential point of view alone, our result is useful since it is saying that a finite timescale separation, which importantly results in an implementable algorithm as opposed to a timescale separation approaching infinity, can guarantee the type of convergence we would desire. Then, from a constructive point of view, since we can exactly compute $\tau^{\ast}$, we can explore many types of problems to gain insights into what we can generally expect as necessary or sufficient for the desired behavior. We in fact did this for the many simpler experiments we ran, and found that the theoretical construction of $\tau^{\ast}$ was also tight as the theory would suggest and furthermore $\tau^{\ast}$ is often a reasonable finite value that allows for a range of learning for the discrete time update to converge to local minmax (which would not be feasible if the timescale separation needed to actually approach infinity). This is helpful because it can inform heuristics for more complicated problems such as training generative adversarial networks as well as explain the success of heuristics that are already frequently deployed. To give a concrete example, we found that the insights we gained from the simple Dirac-GAN example turned out to effectively match up with what we observed in the performance of the generative adversarial network training experiments for various hyperparameter settings.
>
> Regarding “The added value of leveraging an array of mathematical tools to provide this explicit construction is unclear to me.”
>
> As noted in our general response to you and reviewer 3, the new tools in our paper expose a number of opportunities for analyzing multiple hyperparameters as we demonstrate with Theorem 3 on generative adversarial networks. Moreover, also as noted above, because of the explicit construction allowed by these tools, we are able to see that 1) in practice the value of $\tau^\ast$ is generally not large and 2) we can optimize for convergence rate by exploring the tradeoff between the choice of finite $\tau$ and the learning rate $\gamma_1$.
>
> The review says:
>
> “The convergence results essentially leverage the asymptotic stability properties provided by Theorems 1 and 2.”
>
> We believe that our convergence results are actually more meaningful than just thinking of them as asymptotic convergence in the neighborhood of a asymptotically stable equilibrium. To get this result, there are in fact several steps and it does not follow trivially from Theorems 1 and 2. This is because $J_{\tau}(x^{\ast})$ has complex eigenvalues, so to obtain rates that do not assume $J_{\tau}(x^{\ast})$ has a positive definite symmetric part (which is required to use the normal 2 norm convergence rate analysis techniques), we formulate an optimization problem for finding the stepsize given $\tau>\tau^{\ast}$ that maximizes the spectral radius of the local linearization $I-\gamma_1 J_{\tau}(x*)$. Furthermore, this analysis also includes an estimate on the region of convergence.
>
> Finally, we note that we have a number of other results including specializing to generative adversarial networks in Theorem 3, extensive numerical experiments that range from interpretable and informative to large-scale training, results on the stochastic setting where we get almost sure asymptotic convergence using less restrictive stepsize assumptions than is typical since we do not need a two-timescale sequence given $\tau^{\ast}$.
>
> [1] Jin et al. Arxiv 1902.00618 (published in ICML 2020). What is Local Optimality in Nonconvex-Nonconcave Minimax Optimization.

---

### Official Review · AnonReviewer3 · 2020-10-28
**Part of the main result might already be known (concern clarified by revision)**

**Rating:** 6
**Confidence:** 4

**Review:**

The main result of the paper states that a strict local minmax point is a stable critical point of t-GDA for some large enough t, and that any non-strict local minmax can be made unstable by s-GDA if we choose s large enough.


-Major issues

My greatest concern is that the first part of the main result, that a strict local minmax is stable for t-GDA with all large, but finite t, is already known (Jin et al. 2020). Specifically, the proof of Lemma 40 in (Jin et al. 2020) shows that for all large enough finite t, the Jacobian of t-GDA only has eigenvalues whose real part is smaller than 0, which then implies the stability of t-GDA for a finite t.

From what I can see, the reason why Jin et al. 2020 stated their results in terms of infinite timescale separation is because they did not have a uniform bound on how large the timescale t should be, and therefore in general it can be made as large as possible (but finite). The proof in the current submission has exactly the same feature: for every game, there is a finite t that makes t-GDA stable, but in general this t can be made arbitrarily large. It seems to me that the authors have not qualitatively improved over (Jin et al. 2020) (although I believe the bounds in this paper are tighter).

In the same vein, the converse statement also more or less appeared in (Jin et al. 2020); see the proof of Theorem 28, p.24-25 therein.

Due to the above, I cannot see the claimed novelty of providing the first finite timescale separation for GDA, hence my rating. I'm willing to change my score should the authors convince me that I have misunderstand something.


-Minor issues
1. The authors claimed that "On the empirical side, it has been widely demonstrated that timescale separation in gradient descent-ascent is crucial to improving the solution quality when training generative adversarial networks." I believe this is an overstatement of what we currently know about GANs; see

https://arxiv.org/pdf/1711.10337.pdf

for a comprehensive empirical study of the effects on the timescale for GDA, which is not as conclusive as the authors stated. I would therefore suggest to tone down the sentence.

2. Appendix B.3 is quite weird. F(x_k) here should be a vector-valued mapping but the authors seem to view it as a function. Also, by F(k) = O(M^k), did the authors mean ||x_{k+1} − x*|| = O(M^k)?


----

Post-revision evaluation:

The authors have modified the statements of the main theorems as well as including a more detailed comparison to previous works, which clarifies my concern. I have thus increased my score.

The technical contributions bring new insight into the studying of scale separation of GDA, and enables a tight characterization of many toy examples. I believe these are solid contributions and should be valued.

On the down side, I'd like to point out that the "practical implication" in this paper is a bit of stretch since the ImageNet experiments are run with RMSprop, whereas the analysis of this paper is highly specialized to GDA.

Of course, studying adaptive algorithms in min-max games is exceedingly hard and well beyond the scope of this paper. What I recommend the authors is then:

1. Explicitly notify the readers of the difference between RMSprop and GDA.

2. Find a nontrivial but simple example where 1-GDA provides an okay baseline (say 7-layer CNNs for mixture of Gaussians or MNIST). Increase the time scale to show if it exhibits a similar behavior that a small $\tau$ gives the best result. This is directly verifying what the theory is saying, and hence feels more valuable to me.

---

> ### Author Response · Authors · 2020-11-12
> **Response to Reviewer 3 (2/2)**
>
> Regarding minor point on empirical section: While we are happy to tone down the sentence as suggested, we would like to point out that in the cited paper it appears they always have the same learning rate and explore 1 versus 5 rollouts for the discriminator, e.g. which isn’t necessarily a contradiction of our claim. We are more suggesting that there has been numerous works touting the empirical success of different learning rates between the generator and discriminator and hence our work provides some theoretical backing for this heuristic. This is not to suggest or imply that timescale separation is strictly necessary for all GANs (as the reference points out, not all GANs are created equal).
>
> Regarding appendix B.3: Fair point. There is a conflation of the use of F here which we should not have done. On the one hand $x_{k+1}=F(x_k)$ where F defines the dynamics and then in the next sentence F is used as a real valued function. We will change this so that $x_{k+1}=h(x_k)$ some different function and then use F(k) to be a scalar value function such as the norm $\|x_k-x*\|$, e.g. The point was to define how big O, is being used in our convergence rates, but perhaps this is common enough that a reference would suffice.
>
> [1] Jin et al. arxiv 1902.00618 (published ICML 2020). What is Local Optimality in Nonconvex-Nonconcave Minimax Optimization.
> [2] Kokotovic et al (see, Chapter 3). 1986. Singular Perturbation Methods in Control Analysis and Design.

---

> ### Author Response · Authors · 2020-11-12
> **Response to Reviewer 3 (1/2)**
>
> We thank you for your review. Regarding your `major concern’, see the general response to you and reviewer 4. Here, we now address remaining specific concerns that you had.
>
> In the review, you said:
>
> “From what I can see, the reason why [1] stated their results in terms of infinite timescale separation is because they did not have a uniform bound on how large the timescale $\tau$ should be, and therefore in general it can be made as large as possible (but finite). The proof in the current submission has exactly the same feature: for every game, there is a finite $\tau$ that makes $\tau$-GDA stable, but in general this $\tau$ can be made arbitrarily large. It seems to me that the authors have not qualitatively improved over [1] (although I believe the bounds in this paper are tighter).”
>
> To begin, it is extremely important to note that as described in our general response to you and reviewer 4, and as Theorem 28 is stated in [1], that the result of [1] is only for $\tau\rightarrow \infty$ and it is not finite. To reiterate, the part of the proof of Theorem 28 in [1] you are referencing never said a local minmax is guaranteed to be stable for a finite $\tau$, but rather said that the eigenvalues of $-J_{\tau}(x^{\ast})$ are asymptotically close to the open left-hand plane, but [1] only guarantee they are actually in the open left-half plane in the limit.
>
> We would respectively like to strongly push back at this assumption you have made about why [1] did not obtain/write down a finite $\tau$ result and provide evidence for why this is clearly not the case. The tools you need to get a construction (non-asymptotic) of $\tau^\ast$ are completely different and totally unrelated to the asymptotic construction in either [1] or [2, Chapter 3]. Hence, it is not just because [1] did not have a uniform bound. It is rather because it is a highly non-trivial problem requiring a totally different set of analysis tools. The construction by no means follows from the results in [1]. It is a non-trivial problem to construct in a non-asymptotic manner this finite value of $\tau$ as demonstrated in our work. Indeed, from a historical perspective, there is a long history/body of work from the 1980’s on asymptotic stability of a closely related dynamical system to $\tau$-GDA known as singularly perturbed systems (which we bring to light and has not been talked about before in the context of learning in games as applied to ML). Following this body of work, there was at least ten years of extensive work trying to characterize stability for finite singularly perturbation parameters, and it is this work that we build on to give our novel construction.
>
> For the final part of your comment, see the general response to you and reviewer 4, where we have noted that the $\tau^{\ast}$ we construct is often a small reasonable value, and furthermore, we have a method to compute exactly what it is. Moreover, it is not just that our result is tighter, it is a completely different result since there is no non-asymptotic result in [1].
>
> ”In the same vein, the converse statement also more or less appeared in [1]; see the proof of Theorem 28, page 24-25 therein.”
>
> If by converse you mean the following direction in Theorem 1:  $\exists\  \tau^\ast<\infty$ s.t. $\mathrm{spec}(-J_\tau(x^*)) \subset \mathbb{C}_-^\circ$ for all $\tau>\tau^\ast\ \implies\ x^\ast$ is a strict local minmax, then again in [1] they only provide an asymptotic result while we provide an  argument by contradiction which does not depend on asymptotic analysis. In particular, the result in [1] depends on the asymptotic expansion in equation (7) as detailed in the general response to you and reviewer 4.
>
> If by converse you mean the instability result (Theorem 2), then there is no analogous result in [1.]
> Theorem 2 states that if a critical point is not a strict local minmax then there exists a finite $\tau_0$ such that for all $\tau>\tau_0$, $\mathrm{spec}(-J_\tau(x^\ast)) \not\subset \mathbb{C}_-^\circ$ (i.e. its not asymptotically stable).  Theorem 2 gives a way to construct $\tau_0<\infty$ such that any non-strict local minmax critical point becomes unstable for all $\tau>\tau_0$. This is not implied by the asymptotic result in Theorem 28 of [1] or even our Theorem 1. Indeed, the relevant direction from our Theorem 1 is as follows: $\exists\ \tau^{\ast}<\infty$ s.t. $\mathrm{spec}(-J_\tau(x^\ast)) \subset \mathbb{C}_{-}^{\circ}$ for all $\tau>\tau^\ast\ \implies \  x^*$ is a strict local minmax.  This is not a constructive theorem for finding $\tau_0$ that makes non-strict local minmax critical points become unstable. This is to say that Theorem 2 is showing something different.

---

> ### Author Response · Authors · 2020-11-23
> **Thanks for Updated Review: Some Further Response to Your Suggestions**
>
> Dear reviewer 3, we want to thank you for the effort and care you have put into reviewing our paper. Your comments were very important in improving how the paper is positioned and making clear our contributions in the context of the literature.
>
> We saw in your post revision evaluation which you included in your edited review some additional comments that did not come up in our previous discussions regarding suggestions to boost the practical relevance in your point of view.
>
> We have edited the experiments section with the image datasets to point out more clearly the differences between RMSprop and GDA. We appreciate you acknowledging that studying adaptive algorithms in min-max games is exceedingly hard and beyond the scope of this paper.
> We do note that, as you are probably aware, it is pretty standard in the literature for learning rules in min-max problems to be analyzed and then in the implementations for large scale image generative adversarial networks to use adaptive methods on top of those algorithms. We still find the insights about the effect of timescale separation useful when considering these variations and complimentary to the extensive set of experiments we present in the appendix with vanilla GDA with timescale separation.
>
> This being said, we think your suggestion about some training some simpler generative adversarial networks that include neural networks using vanilla GDA with timescale separation is reasonable and could strengthen the paper. Given that this was suggestion close to the end of the revision period, we had limited time to explore this in depth. However, we did follow your suggestion and trained some generative adversarial networks with a 2 hidden layer generator and 1 hidden layer discriminator with 32 neurons in each layer to learn a 2d mixture of 8 Gaussians arranged in a circle. For this scale of problem, which falls into what you suggested, it is not necessary to use much larger networks. We did the experiments with a batch size of 512 and 1024 and varied $\tau$ with a fixed learning rate of $\gamma_1=5e-3$ and ran for 40k updates. Each parameter configuration was simulated 8 times and every 2k iterations we computed the KL divergence between the real data and the generated data. The results are in this repo https://anonymous.4open.science/r/f3f459fc-0a2e-4b41-b5ad-29acf39cb043/ in the folder called MoG_GAN_Figs. We plot the mean KL divergence across the runs along the learning path and shade in the standard error of the mean. The figures in the folder are labeled by the batch size and if zoom is in the name of the figure, we simply zoomed in on a smaller range for the KL divergence on the y-axis to make the differences between the parameter settings more clear.
>
> We find that the insights we draw on this example are consistent with the many simpler, interpretable examples we presented including the Dirac-GAN and in the several found in appendix, as well as the image dataset generative adversarial network training where we used RMSprop. Notably, when there is not any or minimal timescale separation there is cycling. Then, as we increase the timescale separation up to a point the performance continuously improves, before beginning to degrade as the timescale separation gets much larger. We are open to including such experiments of this form in the appendix before the revision period ends shortly if it would strengthen your view of the paper.
>
> Thanks again for the many helpful comments and feedback.

---

### Official Review · AnonReviewer2 · 2020-10-28
**An very interesting theoretical result that could be backed-up with experiments**

**Rating:** 7
**Confidence:** 4

**Review:**

## Summary
This paper studies the stable points of gradient descent ascent with different step-sizes.
Roughly, this paper's main result is that for any fixed point for the GDA dynamics, that point is stable for GDA with a large enough timescale separation if and only if it is a local Stackelberg equilibrium.

This result is quite intuitive since a similar result has been proved by Jin et al. 2020. (the timescale had to go to infinity). Though, proving such a result for a *finite* timescale separation is a major improvement and is related to practical significant considerations.

## Pros and cons
Strengths:
the work is well motivated, and overall the paper is well written. Though if I think some of the technical aspects could be improved.
The results may interest the community.
The theoretical tools introduced could be used in other theoretical work.

weaknesses:
Some of the theory sections could be more developed to build intuitions on the phenomenon going on. It seems really hard for me to follow the current proof sketches since many notations are not properly introduced (or at least any intuition about what they mean is proposed). (see my section on questions/ comments)
The experiments are not really related to the theory. (see my section about experiments)
I have some technical concerns and questions (I would be glad if the authors can answer them; see my section on questions)
The conclusion is overstating the results of the paper.

## Overall review
Overall, this paper is a good paper that should be accepted if the authors fix some statements in the conclusion section and answer my questions about the theory. Also, the experimental section could be improved (not by running more large scale experiments but by more related to the theory presented)

## Questions and comments (by decreasing order of importance):

### technical question
The most important question I have regards the guardian map you use. In the proof sketch of Theorem 1 where you state that the map $\nu(\tau) := det(-(J_\tau \oplus J_\tau))$ is a guardian map of $\mathcal{S}(\mathbb{C_-^\circ})$ but if we look at lemma C.1 the result boils down to the fact that the eigenvalues of $A \oplus A$ are $(\lambda_i + \lambda_j)$ with $\lambda_i, \lambda_j \in Sp(A))$.
However, it means that if $A$ has a single eigenvalue in $\mathbb{C_-^\circ}$, then $\nu(A)$ is non zero. Some arguments should be added (since $A$ is a real matrix, the non-real eigenvalues are complex conjugates) to justify that the only problematic case is when $A$ has $0$ as an eigenvalue. One quick fix would be to consider $\nu(A) := det(A)det(-(A\oplus A))$ as guardian map$. But I do not know how it would change the derivations in the proof of claim C.1.
I am quite novel with these notions, so my questions are: Am I missing something? If yes, what? If not, would the fix work, and will it change the results of Theorem 1 and 2?

### Conclusion
In the conclusion section, you write, “We proves gradient descent-ascent converges to a critical point for a range of finite learning rate ratios if and only if the critical point is a differential Stackelberg equilibrium. This answers a standing open question about the convergence of first-order methods to local minimax equilibria.” These two sentences may be misleading for the following reasons:
- you only prove a *local* convergence result
- The local distinction is important because the method can still cycle outside of these neighborhoods. (see for instance [Letcher 2020])
- Also, the value of $\tau$ depends on the neighborhood. So it seems that you may have an infinite number of critical points, and the value of \tau to globally (max of the \tau for each critical point) only has local convergence to local minimax may be infinite. Do you agree with that statement?

### Experiments
Your theory is about the nature of the stationary points found by the training dynamics  (are they theoretically meaningful), but you do not verify that training with different learning rates actually finds local minimax. Moreover, it is known that using different learning rates is necessary to get better empirical performances (see, for instance [Brock et al. 2019, Jolicoeur-Martineau et al. 2020] or most of the SOTA results on https://paperswithcode.com/paper/adversarial-score-matching-and-improved). I think that one experiment that would support your theory would be about looking at the eigenvalues of J_\tau around the “practical equilibria” for different values of $\tau$ and see if you only get local minimax for large enough $\tau$.
In Assumption 1, you suppose that $(w,\theta)$ is an equilibrium. But what kind of equilibrium? Nash? What do you need? That it is a stationary point of the dynamics?


### About guardian maps
Shouldn’t you add in the definition of a guardian map that the map is continuous?

### Related work
Your examples 1 and 2 look very similar to the ones proposed by Zhang et al. 2020 in section 5.2. Can you comment on that? (I guess there is a difference, but I think this is related work that should be addressed, particularly Section 5.2)

### Questions/comments on the appendix
I think there is a typo that should be fixed on page 22: you recall that $A \oplus B = A \otimes B + B \otimes A$, which is, I think, incorrect. (and it is the only place where I found a definition for $\oplus$)
On page 24, you mention more elegant constructions (what are these construction?).
You also claim that you get the tightest bounds for $\tau$. Can you compare these bounds? Can you prove that you are tighter?
I do not understand the sentence "we use $A \boxplus A$ because of its computational advantages." Do you mean algebraic computations? Computational advantage usually refers to the algorithmic complexity to compute these quantities, but I guess this is not what you are talking about in the sentence.

### Minor comments:
Page 2 maybe define Schur complement
Page 3 The notations $vec$ is not introduced.
Page 4 The Kronecker product and sum are not defined
Proof of Lemma C.1 there is no n_1 and n_2


Refs:

Brock, Andrew, Jeff Donahue, and Karen Simonyan. "Large scale gan training for high fidelity natural image synthesis." arXiv preprint arXiv:1809.11096 (2018).

Zhang, Guojun, Pascal Poupart, and Yaoliang Yu. "Optimality and Stability in Non-Convex-Non-Concave Min-Max Optimization." arXiv preprint arXiv:2002.11875 (2020).


Letcher, Alistair. "On the Impossibility of Global Convergence in Multi-Loss Optimization." arXiv preprint arXiv:2005.12649 (2020).

Jolicoeur-Martineau, Alexia, et al. "Adversarial score matching and improved sampling for image generation." arXiv preprint arXiv:2009.05475 (2020).

---

> ### Author Response · Authors · 2020-11-11
> **Response to Reviewer 2 (2/2)**
>
> Experiments:
>
> We do verify local minmax for a number of examples including the Dirac GAN. Moreover, we explored this in the Appendix since this is important with a number of examples which we felt were interpretable such as quadratic, polynomial, and torus games along with a simple generative adversarial network for learning a covariance matrix. Furthermore, for such examples, we confirmed our construction of $\tau^{\ast}$ was tight and in the supplementary material we included code to compute $\tau^{\ast}$. For the generative adversarial networks with image datasets, we did not do this as they are much more computationally intensive and at present there are not great methods for this scale of problem to estimate the eigenvalues. Hence, we focused on performance for those experiments and found we could draw similar insights as from our simple examples as discussed in the experiments section. For the last point, we meant critical point, not equilibrium. We will change this in assumption 1 and in the Theorem 3 statement.
>
> Guard map:
>
> The definition [5] does not include this, however, the guard map we construct is continuous. We can add a comment about continuity.
>
> Related Work:
>
> These are just illustrative examples and not results. They have appeared in various forms across the literature including in [1,2,3,4].  We will add the citation to [4] in the paper as well.
>
> Questions on Appendix:
>
> Yes, that is a typo. It should be $A\oplus B=A\otimes I+I\otimes B$. Thanks for the catch! We will also move this definition up to the main, along with the definition of the vec operator.
>
> Regarding the more elegant constructions: We simply meant that you could directly compute $\tau^{\ast}$ via $\text{det}(-(J_{\tau}(x^{\ast})\oplus J_{\tau}(x^{\ast})))$ or even via a Lyapunov construction analogous to the approach in the instability proof (Theorem 2) and these are both more “interpretable” than using $\text{det}(-(J_{\tau}(x^{\ast})\boxplus J_{\tau}(x^{\ast})))$, since $\boxplus$ is not a common operator given that it uses duplication matrices to reduce the dimension of the problem. Specifically on this point, using $\boxplus$ actually reduces the computational complexity of finding $\tau^{\ast}$ since it removes redundancies. That is, the use of $\boxplus$ lets us reduce the computational complexity because it takes symmetric operators of say dimension $n^2$ and reduces to dimension $n(n+1)/2$.  Specifically, the size of the matrix for which you need to solve the eigenvalue problem is of much lower dimension for $\boxplus$ versus $\oplus$.
> The bound is tight because all guard maps by definition vanish for the same real values of the parameter $\tau$ (see [6], remark 2), and hence $\tau^{\ast}$ is independent of the guard map and in our construction there is no approximation introduced.
> More specific to the question of comparison, as noted all guards by definition produces the same real roots so they would all lead to a tight construction of $\tau^{\ast}$ but potentially more computationally expensive. We will clarify/formalize this statement as it is an important point. Thanks for highlighting that.
>
> Minor comments: Thanks for the pointers. We will address each of these in our revision.
>
> [1] Fiez et al. ICML 2020, Arxiv 2019. Implicit Learning Dynamics in Stackelberg Games: Equilibria Characterization, Convergence Analysis, and Empirical Study, Convergence of Learning Dynamics in Stackelberg Games.
> [2] Jin et al. ICML 2020. What is Local Optimality in Nonconvex-Nonconcave Minimax Optimization.
> [3] Zhang et al. Arxiv 2020. Optimality and Stability in Non-Convex-NonConcave Min-Max Optimization.
> [4] Mazumdar et al. SIMODS 2020, Arxiv 2018. On Gradient-Based Learning in Continuous Games.
> [5] Saydy. New stability/performance results for singularly perturbed systems.
> [6] Saydy et al. Guardian maps and the generalized stability of parametrized families of matrices and polynomials.
> [7] Ratliff et al. 2014  Genericity and structural stability of non-degenerate differential Nash equilibria. Allerton.
> [8] Mazumdar et al. 2019 IEEE CDC. ‘Local Nash Equilibria are Isolated, Strict Local Nash Equilibria in ‘Almost All’ Zero-Sum Continuous Games’

---

> > ### Comment · AnonReviewer2 · 2020-11-16
> > **Thank you**
> >
> > Thank you for your detailed answer.
> > Regarding the continuity of the guard map I was initially surprised but after reading [5] it now makes complete sense.

---

> > > ### Author Response · Authors · 2020-11-18
> > > **Thanks**
> > >
> > > Thanks for getting back to us. We agree that there is certainly some interesting tools in the old papers by Saydy.

---

> ### Author Response · Authors · 2020-11-12
> **Response to Reviewer 2 (1/2)**
>
> Thank you for the very detailed review and insightful comments. We appreciate how in depth you went into the paper and the appendix, it is very helpful and constructive!
>
> We hope to clear up your questions in the following response.
>
> Technical question:
>
> We need to clear up some points to answer your question.
>
> First, the guard map for the matrices with eigenvalues in the open left-hand plane, which we will denote by $\mathcal{S}(\mathbb{C}_{-}^{\circ})$, is in fact $\nu(A)=\text{det}(A)\text{det}(A\oplus A)$ in general as you suggest.
>
> However, when A is non-singular, $\nu(A)=\text{det}(A\oplus A)$ is a guard map for matrices with eigenvalues in the open left-hand plane. In our case, we only use the guard map on $\mathcal{S}(\mathbb{C}_{-}^{\circ})$ as motivation/intuition for how we end up using it. Specifically, we don’t care about all possible matrices $A$ with eigenvalues in the open left-hand plane.
>
> Instead, we care about the parameterized family of matrices $-J_{\tau}(x^{\ast})$ parameterized by $\tau$. Looking only at these matrices, we reduce $\nu(A)$ to $\nu(tau)=\text{det}(-(J_{\tau}(x^{\ast})\oplus J_{\tau}(x^{\ast})))$ which is now a polynomial in $\tau$, and we can do this because by assumption $\text{det}(\text{Schur}(J_{\tau}(x^{\ast}))\neq 0$ and $\text{det}(D_2^2f(x^{\ast}))\neq0$ so that $\text{det}(J_{\tau}(x^{\ast}))=\tau^{n_2}\text{det}(\text{Schur}(J_1(x^{\ast}))\text{det}(-D_2^2f(x^{\ast})) \neq 0$. That is, $J_\tau(x^\ast)$ is non-singular for all $\tau>0$. We mentioned this in the proof sketch of Theorem 1 where we connect to the 2x2 case. Specifically, we know in the 2x2 case that $\nu(A)=\text{det}(A)\text{Tr}(A)$ is a guard map and we argue that for $J_{\tau}(x^{\ast})$, $\text{det}(J_{\tau}(x^{\ast})) \neq 0$ since $\text{det}(\text{Schur}(J_{\tau}(x^{\ast}))\neq 0$ and $\text{det}(D_2^2f(x^{\ast}))\neq0$ by assumption. We will amend Lemma C.1 to add that $\nu(A)=\text{det}(A\boxplus A)$ is a guard map for non-singular Hurwtiz stable matrices. Thanks for catching that!
>
> Conclusion:
>
> We will change that sentence to say:
>
> “We prove gradient descent-ascent locally converges to a critical point for a range of finite learning rate ratios if and only if the critical point is a differential Stackelberg equilibrium. This answers a standing open question about the local convergence of first-order methods to local minmax equilibria.”
>
> For the final point, the value of $\tau^{\ast}$ will be finite for any (isolated) critical point as shown in Theorem 1 (isolated because of the non-degeneracy assumptions, and a generic property of functions $f\in C^2$. While there may be an infinite number of (isolated) critical points, each of these still has a finite value of $\tau^{\ast}$ and the supremum over this set could conceivably be taken to get the guarantee you refer to and $\tau^{\ast}$ will not infinite even in this case since it is finite for all critical points. Our results do not say anything about the case where there is a continuum of critical points, however, this is known to be a non-generic condition and a non-structurally stable condition (any smooth perturbation will remove the continuum — see e.g., [1,7,8] for precise statements on genericity and structural stability in games).  In the conclusion we will add some discussion of non-convergence results (as requested by several reviewers) which do not contradict our results but are rather complementary to them.

---

### Official Review · AnonReviewer1 · 2020-10-31
**Local analysis of Gradient Descent-Ascent with a finite timescale separation**

**Rating:** 6
**Confidence:** 4

**Review:**

The paper studies  the local asymptotic stability of a specific class of solutions points, referred to as strict local minmax equilibria (or differential Stackelberg equilibria), in the case of Gradient Descent-Ascent Dynamics with a finite time-scale separation. The time-scale separation (\tau) is being captured by the ratio of the step-sizes between the min and max agents respectively. Recently, Jin et al. showed the set of asymptotically stable critical points of gradient descent-ascent coincide with the set of differential Stackelberg equilibrium as the time separation goes to infinity. The paper shows that an infinitely large separation is not needed and some finite but large enough separation suffices.  The paper provides a close analogue of another previous result by Mescheder about local stability of gradient descent dynamics in GANs under strong technical assumptions. The paper ends with GAN experiments where \tau=1, 2, 4, 8 are tested and the performance seems to peak at 4.

The paper performs a detailed theoretical analysis of the coupling between GDA and diff Stackelberg equilibria. Although this is positive, the results are not particularly surprising given the prior work. The writing of the paper could also be significantly improved.

One issue that I had reading the paper is that at times and especially in the introduction the
treatment of (asymptotically stable), stable, unstable fixed points seem to be a little ambiguous.The paper only formally defines locally exponentially stable equilibrium in the preliminaries which is a notion that is not used in the introduction. I think it is important to set early on a clear terminology that is consistent throughout the whole paper.
I am also a bit confused about some statements in the paper about which type of solution concepts are game theoretically meaningful. The paper seems to state that any critical point that does not satisfy the definitions of differential Stackelberg equilibria lack game theoretic meaning.  From the paper

the stable critical points of gradient descent-ascent coincide with the set of differential Stackelberg equilibrium as \tau goes to infinity. All ‘bad critical points’ (critical points lacking game-theoretic meaning) become unstable and all ‘good critical points’ (game-theoretically meaningful equilibria) remain or become stable as \tau goes to infinity.

This seems like a strong statement. It seems to me that min-max solutions of bilinear zero-sum games does not satisfy the differential Stackelberg definition. Such statements would imply that min-max solutions are 1) bad critical points and 2) lack game theoretic meaning despite being the golden standard of a solution concept in game theory.
Maybe I am missing something here?

[1] has recently shown that alternating GDA with fixed time-separation does not converge in the case of bilinear zero-sum games but is instead recurrent with the min-max equilibrium being stable but not asymptotically stable. This seems to be exactly the setting that you are studying. How are the results of [1] connected to yours? I think that due to the tight match between the two settings a thorough discussion is needed.

The definitions of differential/strict Nash/Stackelberg equilibria are two of several alternative definition/solution concepts that have only been recently introduced in the context of non-convex non-concave games. The paper should compare and contrast to other notions ideas  (e.g. proximal equilibria are only mentioned briefly [2], see also [3], [4]).

 Although one should of course not expect a global convergence as such a result would be too ambitious, the title could be interpreted as such a result by non-specialists. I think it might be better if the term local analysis is used instead. A more thorough discussion about non-convergence results for GDA and variants in zero-sum games could be also helpful [1,3,5-8] to dispel any possible confusion.

In terms of the experimental results why do the simulations stop with \tau=8? The theoretical results as well as the prior work by Jin et. al are supportive of arbitrary large \tau. What happens e.g. for tau= 2^4, 2^8, .... It already seems that performance starts dropping for \tau>4. Does this trend continue? Does the performance have a unique peak? or does it fluctuate?
The regularization results (Theorem 3) remain true even for \tau<<1 e.g. \tau = 2^{-4}, 2^{-8}, ...
What would experiments show for such \tau under regularization?


[1] Bailey et al. Finite Regret and Cycles with Fixed Step-Sizevia Alternating Gradient Descent-Ascent. COLT 2020
[2] Farnia et. al Do GANs always have Nash equilibria? ICML 2020
[3] Vlatakis-Gkaragkounis et al. Poincaré Recurrence, Cycles and Spurious Equilibria in Gradient-Descent-Ascent for Non-Convex Non-Concave Zero-Sum Games. NeurIPS 2019.
[4] Zhang, et al. Optimality and Stability in Non-Convex-NonConcave Min-Max Optimization. arXiv e-prints, art. arXiv:2002.11875, February 2020.
[5] Mertikopoulos et al. "Cycles in adversarial regularized learning." Proceedings of the Twenty-Ninth Annual ACM-SIAM Symposium on Discrete Algorithms. Society for Industrial and Applied Mathematics, 2018.
[6] Cheung et al. "Vortices instead of equilibria in minmax optimization: Chaos and butterfly effects of online learning in zero-sum games." arXiv preprint arXiv:1905.08396 (2019).
[7] Letcher "On the Impossibility of Global Convergence in Multi-Loss Optimization." arXiv preprint arXiv:2005.12649 (2020).
[8] Hsieh, et. al. "The limits of min-max optimization algorithms: convergence to spurious non-critical sets." arXiv preprint arXiv:2006.09065 (2020).

---

> ### Author Response · Authors · 2020-11-18
> **Do you have any response?**
>
> Dear Reviewer 1,
>
> We just wanted to comment in hopes that you have seen our response to your review, and we would greatly appreciate further comments from you.

---

### Author Response · Authors · 2020-11-12
**Common Response to Reviewers 3 and 4 (3/3)**

Importance for and Impact on Future Research in ML/AI:

The impact and implications of this work are significant. They are important for the development of provably convergent (to meaningful equilibria) first order algorithms for AI/ML applications that can be formulated as minmax optimization problems or zero-sum games.

Zero-th Order/Black-Box Online Learning:

A number of recent works have focused on minmax optimization problems in settings where agents receive noisy gradients and/or only function evaluations. In the non-convex-non-concave setting (even in the non-convex-strongly concave setting) at best it is known that first order methods can achieve $\varepsilon$-first order stationary points, and second order methods are highly computationally expensive requiring numerous samples/environment interactions to obtain reasonable gradient estimates. Leveraging the insights in this paper, there are numerous opportunities to improve existing results along this direction toward reaching only game-theoretically meaningful equilibria.

Adversarial Learning:

Hyperparameter tuning and developing well-founded, practically implementable heuristics for adversarial learning are one of the major challenges. Our work exposes a new set of tools (e.g., the guard map) which can be used to give theoretical guarantees (and hint at ways of constructing well-founded heuristics) on multiple parameters such as learning rate and regularization which play key roles in the practical implementation of, e.g., GANs. This set of tools is amenable to multiple parameters and could, e.g., give guarantees on learning rate parameters, regularization parameters, and exponential averaging parameters simultaneously. This is a direction of future work that comes directly from the contributions in our paper.

[1] Jin et al. arxiv 1902.00618. What is Local Optimality in Nonconvex-Nonconcave Minimax Optimization. (published in ICML 2020).
[2] Kokotovic et al (see, Chapter 3). 1986. Singular Perturbation Methods in Control Analysis and Design.

---

### Author Response · Authors · 2020-11-12
**Common Response to Reviewers 3 and 4 (2/3)**

This type of result is fundamentally different than our statement which for clarity says the following:  $\exists\ \tau^\ast<\infty \ :\ \mathrm{spec}(-J_\tau(x^\ast))\subset \mathbb{C}_-^\circ, \ \forall \tau>\tau^\ast \ \Longleftrightarrow\ x^\ast$ is a strict local minmax. That means, we achieve the stability guarantee for any strict local minmax for a finite value of $\tau$, not just in the limit. Indeed, we do not show just that the eigenvalues are arbitrarily close to eigenvalues having a stability property as [1] does, but actually show that the eigenvalues lie in the open left-half plane for all $\tau$ greater than some finite $\tau^\ast$ thereby giving the stability guarantee for finite $\tau$. Moreover, we provide an explicit non-asymptotic construction of the $\tau^\ast$ that achieves this (hence can be seen as its own contribution beyond the existential result), while the asymptotic result of [1] does not lead to any implementable version of $\tau$-GDA with convergence guarantees.

To give some further idea of how the proof techniques compare, the known asymptotic results work by comparing the eigenvalues of $\text{Schur}(-J_\tau(x^\ast))$ and those of $\tau D_2^2f(x^\ast)$ to those of $-J_\tau(x^\ast)$. This result relies on the fact that you expect asymptotically the spectrum of $-J_\tau(x^\ast)$ to split into two components (not necessarily with non-empty intersection), namely the spectrum of $\text{Schur}(-J_\tau(x^\ast))$ and the spectrum of $\tau D_2^2f(x^\ast)$. To give an intuitive explanation of the difference between the asymptotic techniques and ours, if you imagine the eigenvalues of $\mathrm{spec}(\text{Schur}(-J(x^\ast)))$ being negative but arbitrarily close to the imaginary axis, our result is saying that $n_1$ eigenvalues of $\text{spec}(-J_\tau(x^\ast))$ will approach $\mathrm{spec}(\text{Schur}(-J(x^\ast)))$ from the left hand side, where as the asymptotic proof technique is considering that $n_1$ eigenvalues of $\text{spec}(-J_\tau(x^\ast))$ are approaching $\mathrm{spec}(\text{Schur}(-J(x^\ast)))$ from any direction, and this direction could be the right-hand side. This means that the approximation must be arbitrarily small since it needs to be at least as close as the the distance between $\mathrm{spec}(\text{Schur}(-J(x^\ast)))$ and the imaginary axis. Our approach does not rely on this spectrum splitting or comparing the spectrum of $-J_\tau(x^\ast)$ to $\mathrm{spec}(\text{Schur}(-J(x^\ast)))\cup \mathrm{spec}(\tau D_2^2f(x^\ast))$. Instead, we show exactly when the eigenvalues of $-J_\tau(x^\ast)$ cross into the open left-hand plane by using the guard map, and this actually occurs before the asymptotic splitting begins. This gives a much stronger result as a consequence, and provides a new set of analysis tools which may be useful for other analysis in this area (Theorem 3 in our paper is demonstrative where we give guarantees for multiple hyperparameters).

We will amend the statement of Theorem 1 to include the non-asymptotic construction of $\tau^\ast$ versus simply having the existence statement. This is a very important point and contribution, so it should be featured in the result. This will also help to highlight the contribution over the purely asymptotic results of [1] for which no explicit learning rate ratio can be given. Indeed, there is no existing result on a non-asymptotic construction of $\tau^\ast$ for $\tau$-GDA. We acknowledge in the paper the asymptotic result of [1] when we compare their result to the analogous known result from the text book by [2]. We will further comment on the difference between the two results in the related work.

Moreover, our examples illustrate that $\tau^\ast$ in practice is not arbitrarily large and the observation about the spectral splitting (specifically, when the eigenvalues collapse to the real line) occurs at very modest values of $\tau$. These are important points because they lead to practical implementation. As pointed out to reviewer 1, large values of $\tau$ actually lead to poor conditioning. This can be seen even in our toy examples in the root locus plots (i.e., the Re Vs Im eigenvalue plots). The general structure of the root locus plot in fact always looks like what is seen in these toy examples; essentially the eigenvalues start from zero (i.e, when $\tau<<0$, and their imaginary components increase more rapidly than the real component until a point where they decrease and then asymptotically approach the eigenvalues of $\mathrm{spec}(\text{Schur}(-J(x^\ast)))\cup \mathrm{spec}(\tau D_2^2f(x^\ast))$. Our examples highlight that this collapsing to the real line in fact occurs at very modest values of $\tau$. And, our theoretical results give a precise way to construct $\tau^\ast$ in a non-asymptotic manner. Our results also provide an explanation of practical successes of smaller timescale separation and lead to directions for theoretical research on:

---

### Author Response · Authors · 2020-11-12
**Common Response to Reviewers 3 and 4 (1/3)**

In what follows,  we respond to reviewers 3 and 4, who shared a common concern regarding the connection between our result, which provides a non-asymptotic, tight construction of $\tau^\ast$ such for all $\tau>\tau^{\ast}$, $\tau$-GDA locally converges to a local minmax equilibria,  and the result of [1], which only gives asymptotic guarantees ($\tau \rightarrow \infty$) on the local stability of  $\tau$-GDA. Moreover, we will highlight why our theoretical advance is highly significant from both a theoretical and practical perspective and opens up many future research directions.

High level of our response:

First, we respectfully would like to point out that the reviewers are seemingly extrapolating something from [1] that is never implied, which is clear from the theorem statements and proofs in in the paper after careful inspection. In fact, arguably by Proposition 27 of [1], [1] is trying to imply that there does not exists a finite $\tau$. However, we show in Appendix J in great detail that this is not a conclusion one can draw from the examples in Proposition 27 of [1]. Moreover, we use similar examples (Example 1 and 2) to motivate our Theorems 1 and 2. Finally, in Appendix I we provided a proof of the asymptotic result based on the content of a known singular perturbation result [2, Chapter 3] that we found to be arguably more intuitive than proof from [1] to avoid questions of this nature and bring to light a well-established line of literature with analogous results .

To clarify the distinction between what Theorem 28 of [1] says and our results, observe that the result of [1] only holds in the limit which is precisely what their theorem says.  More specifically,  stated in our notation with $\tau$ instead of $\varepsilon$, their result implies there exists a $\tau_1$ such that for all $\tau>\tau_1$, the eigenvalues of $-J_\tau(x^\ast)$ are “close to” those of $\text{Schur}(-J_\tau(x^\ast))$ and $\tau D_2^2f(x^\ast)$ where “close to” is defined by the asymptotic expansion given in their equation 7 of [1] (and also described in our Appendix I where we compare the results of [1] to those of [2, Chapter 3]). Indeed, again in our notation, $\lambda_i=\lambda_i(\mathrm{Schur}(-J(x^\ast)))+O(1/\tau)$ for $i=1,\ldots, n_1$ and $\lambda_{i+n_1}=\tau (\lambda_j(D_2^2f(x^\ast)+O(1/\tau))$ as $\tau\to\infty$. In light of this expansion, [1] can only guarantee that the eigenvalues of $-J_\tau(x^{\ast})$ get close asymptotically to values in the open left-hand plane, but cannot guarantee for a finite $\tau$ they will be in the open left-hand plane. Their proof does not suggest this either. It simply states what we have recalled above: the eigenvalues of $-J_\tau(x^\ast)$ get close to those of the second order conditions for the minmax which are $\text{Schur}(-J_\tau(x^\ast))<0$ and $\tau D_2^2f(x^\ast)<0$. To get the conclusion that the local minmax point is stable — i.e. the eigenvalues of $-J_\tau(x^\ast)$ are in the open left-hand plane -- you have to pass to the limit as $\tau\to\infty$. That means that when they say the local minmax is stable for $1/\varepsilon$-GDA, this is when $\varepsilon \rightarrow 0$ as their precise theorem statement suggests. In fact, their proof technique cannot be extended to the type of finite $\tau$ result we have. This is because their proof technique — like the proof technique of [2] and other asymptotic results — is to show that the eigenvalues of $-J_\tau(x^\ast)$ asymptotically approach those of $\mathrm{Schur}(-J(x^\ast))$ and $\tau D_2^2f(x^\ast)$ which are guaranteed to be in the open left-hand plane because $x^\ast$ is a strict local minmax. Again, to make sure this is clear, for any finite value of $\tau$ their result only guarantees that the eigenvalues of $-J_\tau(x^\ast)$ are close to those of $\mathrm{Schur}(-J(x^\ast))$ and $\tau D_2^2f(x^\ast)$ which is no the same as guaranteeing $\mathrm{spec}(-J_\tau(x^\ast))\subset \mathbb{C}_-^\circ$.

---

> ### Comment · AnonReviewer4 · 2020-11-12
> **Follow up Question**
>
> I would first like to thank the authors for all the effort of writing their very thorough responses.  I think I am close to understanding the point that the authors are trying to make.
>
> The authors are saying that even if we have for $\tau \to \infty$ that $\lambda_i(J_{\tau}(x^*)) \to \lambda_i(\textrm{Shur}(-J(x^*)))$, this still does not necessarily imply that there is a finite $\tau^*$ such that for all $\tau>\tau^*$, $\lambda_i(J_{\tau}(x^*))$ is in the open left-hand plane, even if $\lambda_i(\textrm{Shur}(-J(x^*)))$ is in the open left hand plane. But I have the following objection with respect to this statement.
>
> Let us take a ball $B$ around $\lambda_i(\textrm{Shur}(-J(x^*)))$ that is small enough to be contained fully in the open left hand plane.  We can always create such a ball since $\lambda_i(\textrm{Shur}(-J(x^*)))$ is in the open left hand plane. Now by the limit definition there exists always a $\tau_i^* \in \mathbb{R}$ such that for $\tau>\tau_i^*$ we have that $\lambda_i(J_{\tau}(x^*))$ is inside the ball $B$. And since $B$ is inside the open left hand plane then $\lambda_i(J_{\tau}(x^*))$ is also in the open left hand plane.
>
> We can then use a similar argument for all the $\lambda_{i+n_1}$ as well. Since we only have a finite number of eigenvalues, we can take the maximum of all the $\tau_i^*$ to get a finite $\tau^*$. For this $\tau^*$ then, we get that all eignevalues of $J_{\tau}(x^*)$ are in the open left hand plane.
>
> Maybe I missunderstood the explanation of the authors or there is a missconception in what I have stated above.
> If the authors could point out what is wrong with the reasoning above, then I would be willining to increase my score.

---

> > ### Author Response · Authors · 2020-11-12
> > **Thanks for the Follow-Up: Our Response**
> >
> > Thanks for your fast response and follow-up question. Let us try and be more clear about what we are saying and we hope it will answer your question and help you see our point of view.
> >
> > We agree that if you fix a game and a strict local minmax equilibrium $x^\ast$, then there are balls $\{B_1,\ldots, B_{n_1}\}$ (arbitrarily small potentially) around  $\{\lambda_{1}\big(\text{Schur}(-J_{\tau}(x^{\ast}))\big),\ldots, \lambda_{n_1}\big(\text{Schur}(-J_{\tau}(x^{\ast}))\big)\}$ that lie in the open left-hand plane. Moreover, as you suggested, there is then some $\tau_1$ such that for $\tau>\tau_1$, for all $i$, $\lambda_{i}\big(-J_{\tau}(x^{\ast})\big) \in B_j$ where $j\in \{1,\dots, n_1\}$. The result of [1] is regarding the class of all games, so it each $B_j$ must be arbitrarily small and $\tau_1$ must be arbitrarily large. Our result is not of this nature and even in the scenario of a fixed game and equilibrium, their result does not give a constructive proof and still suffers from the approach of depending on the ball size that is a direct result of comparing the eigenvalues of $-J_{\tau}(x^{\ast})$ to that of $\text{Schur}(-J_{\tau}(x^{\ast}))$ and $\tau D_2^2f(x^{\ast})$.
> >
> > To illustrate why this is important with a simple example from the paper (Fig 1c-d), consider the Dirac-GAN example with regularization. The eigenvalue of $\text{Schur}(J(x^{\ast}))\propto 1/\mu$ whereas the eigenvalues of $-\tau D_2^2f(x^{\ast})\propto \tau\mu$ where $\mu>0$ is the regularization in the cost. Note that we flipped the sign to $\text{Schur}(J(x^{\ast}))$ and $-\tau D_2^2f(x^{\ast})$ to be consistent with the plot in the paper. Furthermore, recall that $x^{\ast}=(0, 0)$ is the unique strict local minmax equilibria for any regularization. For any choice of $\mu>0$, the $\tau^{\ast}$ we obtain with our method is always $\tau^{\ast}=0$. However, clearly if we take the game so that $\mu\rightarrow \infty$, the ball that can be created around the eigenvalue of $\text{Schur}(J(x^{\ast}))$ becomes arbitrarily small so that $\tau_1\rightarrow \infty$ even for this fixed game and equilibrium.  This to say our results are showing that any positive timescale separation works for this entire class of games, whereas [1] requires the timescale separation to go to infinity for the class of games.
> >
> > This example is not a corner case, this is because it is known that strict local minmax are structurally stable [2], which means that the equilibria persists under smooth perturbations to the game (the structure is maintained). In other words, we could take any arbitrary game and smoothly perturb it so that the eigenvalues of a persistent equilibrium continuously move toward the imaginary axis, creating exactly the same phenomena from the simple example above. Our results do not suffer from this problem exactly because they are non-asymptotic and do not try and compare to the limiting case. Since [1] compares to the limiting case, they can only state their result in an asymptotic sense, and that is exactly do in Theorem 28 of [1].
> >
> > [1] Jin et al. ICML 2020, Arxiv 2019. What is Local Optimality in Nonconvex-Nonconcave Minimax Optimization.
> > [2] Fiez et al. ICML 2020, Arxiv 2019. Implicit Learning Dynamics in Stackelberg Games: Equilibria Characterization, Convergence Analysis, and Empirical Study, Convergence of Learning Dynamics in Stackelberg Games.

---

> > > ### Comment · AnonReviewer4 · 2020-11-12
> > > **Value of explicit construction**
> > >
> > > Based on the authors' comments, I undesrstand that they agree that for a fixed game and equilibrium the results of Jin et al already give a finite time scale seperation. I also understand that they are not satisfied by their result because Jin et al's  $\tau^*$ can be arbitrarily large and because Jin et al's argument is existential a.k.a. Jin et al do not give an explicit construction.
> > >
> > > Regarding the claim that  Jin et al's  $\tau^*$ can be arbitrarily large:  Given that Jin et al's argument proves that for a fixed game and equilibrium  there is a finite $\tau^*$ that satisfies the desired property, one already knows that there is a finite infimum across all positive $\tau$ that satisfy the property.  It is clear that the optimal $\tau^*$ corresponds to the last eigenvalue crossing for the final time the boundary of the open left hand plane. Of course since the result is purely existential we get no insight about the the optimal $\tau^*$.
> > >
> > > In my opionion, this reduces the discussion to understanding what is the added value of the explicit construction provided by the authors. I understand that the formula of Claim C.1 on page 24 is way more concrete than the existential result of Jin et al. But I fail to get some novel intruition about the minimal $\tau^*$ just by looking at it. From a theoretical perspective, I do not see how it influences our understanding on how high level properties of the game/equilibrium affect time scale seperation. From a practical perspective, one would need a point close to $x^*$ to even attempt to use the formula which most likely defeats the purpose for most applications.
> > >
> > > This brings the discussion full circle, since I already asked the authors about the added value of the explicit construction in my original review. Although I remain to be convinced about this point, I appreciate once again the thorough responses provied by the authors.

---

> > > > ### Comment · AnonReviewer3 · 2020-11-12
> > > > **My novelty concern remains**
> > > >
> > > > First, I would like to thank the authors for the thorough replies, and Reviewer4's crystal clear arguments which precisely express my confusion.
> > > >
> > > > After carefully reading the replies, Reviewer4's comments, and inspecting the key reference (Jin et al. 2020), I came to the conclusion that ***Theorem 1 of this paper is already contained in (Jin et al. 2020).***
> > > >
> > > > The only missing step from (Jin et al. 2020) to **Theorem 1** is the simple continuity argument sketched by Reviewer4 above, which I deem immediate. As a result, I disagree with the authors claim that "*the reviewers are seemingly extrapolating something from [1] that is never implied, which is clear from the theorem statements and proofs in in the paper after careful inspection.*"
> > > >
> > > > Furthermore, I would like to point out that the authors have misread (Jin et al. 2020) in the following claim:
> > > >
> > > > "*In fact, arguably by Proposition 27 of [1], [1] is trying to imply that there does not exists a finite $\tau^*$. However, we show in Appendix J in great detail that this is not a conclusion one can draw from the examples in Proposition 27 of [1].*"
> > > >
> > > > Proposition 27 of (Jin et al. 2020) aims to show that for every **fixed** $\tau$, there exists a game $f$ such that $\tau$-GDA is not locally stable for $f$ at some local min-max. On the other hand, both the main theorems in (Jin et al. 2020) and this paper state that for every local min-max, there exists a $\tau^*$ such that $\tau^*$-GDA is locally stable. These two do not contradict each other, and hence I find it odd that the authors use it as a counterevidence to the common concern of Reviewer4 and myself.
> > > >
> > > > In conclusion, this paper has serious novelty issues in the theory, although the technical tools seem to be interesting. I have hence increased my confidence in recommending rejection.
> > > >
> > > > I thank again the authors and Reviewer4 for the very illuminating discussions. I remain at your disposal should there remain any question.

---

> > > > > ### Author Response · Authors · 2020-11-12
> > > > > **Reviewer has come to 100% False Conclusion: Hope our Response Clears up the Mistake**
> > > > >
> > > > > The conclusion you have come to about Theorem 1 is 100% wrong. You are unfairly discounting our work and contributions, when it is clear you haven’t fully understood the results and furthermore from the conclusion you have drawn we would be surprised if you have even looked at the analysis in the paper since if you did there would be no way you could come to this conclusion.  Theorem 1 is about a tight construction of $\tau^\ast$ in a non-asymptotic manner that is not implied by [1], meaning we actually can get an exact number for $\tau^\ast$. This goes well beyond existence. Thus, the simple continuity argument provided not by [1] but by reviewer 4 is not enough to get the result of Theorem 1. It appears clear that you did not look at the proof of Theorem 1 in Appendix C if you think that the argument in the discussion with Reviewer 4 is the argument we use to prove our result. Moreover, it is our understanding that even Reviewer 4 is not of this opinion, and that they were simply asking for clarifications on the existential aspects of Theorem 1 in our paper relative to Theorem 28 in [1]. This is only one aspect of Theorem 1, which it self is only one contribution of our paper. The proof of Theorem 1 in Appendix C is a entirely different proof technique that does not rely on drawing balls around eigenvalues and looking at the asymptotic limit a technique which has been known in the singular perturbation literature since the 1980s if not before (see [2, Chapter 3]) for showing this exact result for a class of systems of which $\tau$-GDA is a subset. The technique we use does not require this asymptotic analysis. It finds the exact value of $\tau$ for which the spectrum of $-J_\tau(x^\ast)$ cross into the open left hand plane and stays there.  We show the difference between these two results in a very concrete way with the Dirac-GAN example in our previous response to reviewer 4. Specifically, we showed that $\tau^\ast$ via our method is exactly equal to zero while $\tau^\ast$ for the method in [1] (and [2,Chapter 3]) must approach infinity to work for the class of regularized Dirac-GANs. And even for a fixed regularization parameter, $\tau^\ast$ can be arbitrarily large via the asymptotic technique in [1] and our methods always give the same $\tau^\ast$.
> > > > >
> > > > > We have not misread Proposition 27 of [1], we understand that given a fixed $\tau$, one can construct a game such that $\tau$-GDA is not stable around a strict local minmax. As mentioned in our previous response, we actually provide a whole discussion around this in great detail in Appendix J so we are fully aware of what Proposition 27 in [1] is saying. We did not mean to use it as “counterevidence”, we were actually trying to articulate the difference in perspectives of the papers.  In [1], the perspective is trying to understand for a fixed parameter $\tau$ what is the class of games for which you get convergence guarantees to local minmax,  while our perspective which we are trying to use the discussion of Proposition 27 to highlight is for a given game what are the parameters $\tau$ for which you have guaranteed convergence to game-theoretically meaningful equilibria. Just to be clear, we are not saying in any way that Proposition 27 contradicts any of Theorem 28 in [1] or our Theorem 1, in fact this is exactly why we included a detailed discussion in Appendix J to avoid that exact potential confusion. Nonetheless, the discussion around this is really peripheral and really has nothing to do with our contributions or the discussion thereof.
> > > > >
> > > > > Again finally just to reiterate, we have many more results and experimental analysis beyond the novel result in Theorem 1.
> > > > >
> > > > >
> > > > > [1] Jin et al. arxiv 1902.00618. What is Local Optimality in Nonconvex-Nonconcave Minimax Optimization. (published in ICML 2020).
> > > > > [2] Kokotovic et al (see, Chapter 3). 1986. Singular Perturbation Methods in Control Analysis and Design.

---

> > > > > > ### Comment · AnonReviewer3 · 2020-11-12
> > > > > > **Serious misunderstanding?**
> > > > > >
> > > > > > Dear authors,
> > > > > >
> > > > > > I think there is a serious misunderstanding here. I never thought that "*the argument in the discussion with Reviewer 4 is the argument we use to prove our result*." I was also aware that the technical tools are different (I mentioned this in my comments). Finally, I understand that the techniques in this paper provide further insight into particularized examples such as Dirac-GANs, where as (Jin et al. 2020) could not.
> > > > > >
> > > > > > What I claim is that the **Theorem 1** in this paper, which states the **existence** of a finite time-scale separation (the authors wrote in **Theorem 1** "... There exists a $\tau^* \in (0,\infty)$ such that..."), was already shown in (Jin et al. 2020). This is also what I understood from your discussion with Reviewer4.
> > > > > >
> > > > > > I have tried my best to understand what the authors mean by "asymptotic" for (Jin et al. 2020), and what is the difference between **Theorem 1** of this paper and **Proposition 28** of (Jin et al. 2020). So far the best argument I see from the authors is that equation (7) in (Jin et al. 2020) involves asymptotic expansions. However, these asymptotic expansions came from applying Lemma 41, which is an existential result (that is, there exists a small enough number $\delta$ such that some nice things happen). If one keeps track of this $\delta$ and propagates it into the proof of **Proposition 28**, then one can come up with a finite time-scale $\tau^*$ such that $\tau^*$-GDA is locally stable for $x^*$, which is the major direction of **Theorem 1** in this paper.
> > > > > >
> > > > > > Please let me know what is wrong in the above argument. Many thanks in advance.

---

> > > > > > > ### Author Response · Authors · 2020-11-13
> > > > > > > **Hopefully Clearing up Misunderstanding (2/2)**
> > > > > > >
> > > > > > > In particular, Theorem 1 is not only about the existence of a finite $\tau^{\ast}$, but primarily about actually constructing in a tight way exactly what the value of $\tau^{\ast}$ is.
> > > > > > >
> > > > > > > It has became clear this is not coming across because of the way we stated Theorem 1, despite it being explained in the proof sketch directly following Theorem 1. For this reason, we are going to restate Theorem 1 and similarly Theorem 2 to make this abundantly clear. Moreover, we are going to include further discussion of the result from chapter 2 of [2], where the existence result is obtained from the asymptotic analysis, in comparison to Theorem 1. To be clear and reiterate, what we are doing in Theorem 1 is not just showing the existence of a $\tau^{\ast}$, but providing an exact construction for the value of $\tau^{\ast}$. Moreover, there is no analogue of Theorem 2 in either [1] or [2] as we explained in our initial response directly to you (not in the common response to you and reviewer 4), where we are explicitly constructing a $\tau_0$ such that for all $\tau>\tau_0$, $\tau$-GDA avoids a critical point that is not a strict local minmax equilibrium. Further, we consider the setting from [3], and prove that with any regularization parameter under their assumptions, any $\tau>0$, guarantees the local stability of a strict local minmax equilibria. Then, we use the result of Theorem 1, along with some novel ways to optimize the learning rate of player 1, to give fast explicit convergence rates. Finally, we explore many many informative examples in simulation which support the insights of our theory and elucidate the importance of the results along with an in-depth analysis of training generative adversarial networks and how multiple parameters interplay with each other to dictate performance in a manner that is consistent with the simpler, interpretable experiments we present. Together, we feel that Theorem 1 is being deeply misinterpreted and discounted, and the several other contributions outlined above are not being considered at all.
> > > > > > >
> > > > > > > To come back full circle to your comment, in our discussion with Reviewer 4, we were making the point that one could (although [1] does not), extrapolate from the proof of Theorem 28 in [1] to get the statement in Corollary 3.1 in Chapter 2 of [2], but as we were demonstrating with the dirac-GAN example in the discussion with them, such a method would lead to a $\tau^{\ast}$ estimate approaching infinity, whereas our method exactly recovers the exact value of $\tau^{\ast}=0$ for which the system is always stable given $\tau>\tau^{\ast}$. Finally, we reiterate that as we were explaining to Reviewer 4, such an example is not a corner case because of the structural stability properties of strict local minmax.  From a practical perspective, a tight characterization of $\tau^{\ast}$ is important since it allows for a range of learning rates to be chosen for the discrete time update with convergence guarantees which has a significant impact on the rate of convergence and numerical stability (this is what the linked figures were illustrating).
> > > > > > >
> > > > > > > Let us now explicitly say here how we will revise Theorems 1 and 2 in a way that we hope will clear up the misunderstandings you are mentioning.
> > > > > > >
> > > > > > > Theorem 1 Revised: [A Non-Asymptotic Construction of $\tau^\ast$]
> > > > > > > Given a zero-sum game defined by $f\in C^2$ and a critical point such that $\text{det}(\text{Schur}(-J_1(x^\ast)))\neq 0$ and $\text{det}(D_2^2f(x^\ast))\neq 0$, then the following hold: (1) if $x^\ast$ is a strict local minmax, then for all $\tau>\tau^{\ast}$,
> > > > > > > $\text{spec}(-J_{\tau}(x^*)$ in the open left hand plane
> > > > > > > where $\tau^{*}=\lambda_{\max}^{+}(Q)<\infty$ with $\lambda_{\max}^{+}(\cdot)$ defined as the largest positive real root if it exists and otherwise zero, and
> > > > > > > where $Q$ is defined in Eqn (14) [currently in the Appendix but we will bring it to the main.]
> > > > > > > (2) if $\text{spec}(-J_{\tau}(x^{\ast}))\subset \mathbb{C}_{-}^\circ$ for all $\tau>\tau^{\ast}$ for some $\tau^{\ast}<\infty$, then $x^{\ast}$ is a strict local minmax.
> > > > > > >
> > > > > > > We will revise Theorem 2 along a similar line to highlight the construction. Explicitly, we will bring the eigenvalue problem formulated at the end of the proof in Section D of the Appendix for $\tau_0$ into the main body.
> > > > > > >
> > > > > > > [1] Jin et al. arxiv 1902.00618. What is Local Optimality in Nonconvex-Nonconcave Minimax Optimization. (published in ICML 2020).
> > > > > > > [2] Kokotovic et al (see, Chapter 2). 1986. Singular Perturbation Methods in Control Analysis and Design.
> > > > > > > [3] Mescheder et al. ICML 2018. Which Training Methods for GANs do actually Converge?

---

> > > > > > > ### Author Response · Authors · 2020-11-13
> > > > > > > **Hopefully Clearing up Misunderstanding (1/2)**
> > > > > > >
> > > > > > > There is indeed a serious misunderstanding that is being used to discount Theorem 1 from our paper (which is again just 1 of the many contributions of this work and it is coming off that the rest of the paper is not being considered). Please let us try to explain the error in the conclusions you are drawing, and thank you for appearing to be willing to let us clear this up. To be clear, we do not mean to come off as obstinate, we just honestly think that our results are being misinterpreted and consequently are being discounted so we really want to make sure they are being understood properly and our results in sum to be acknowledged. We believe some of the edits we are proposing to make to the statements as detailed below will help in this direction from our side.
> > > > > > >
> > > > > > > At a high level regarding Theorem 1, when you say:
> > > > > > >
> > > > > > > “What I claim is that the Theorem 1 in this paper, which states the existence of a finite time-scale separation (the authors wrote in Theorem 1 "... There exists a $\tau^{\ast}\in (0,\infty)$
> > > > > > >  such that..."), was already shown in [1]. This is also what I understood from your discussion with Reviewer 4.”
> > > > > > >
> > > > > > >  The fundamental problem with your claim is that Theorem 1 is not only about existence (which as we detail below we will amend the statement in the paper to make this clear, because we acknowledge that one would have to read the proof sketch immediately to see this), but about actually constructing exactly $\tau^{\ast}$ in a tight way such that for all larger $\tau$ the Jacobian is stable. We now explain the use of “asymptotic” in connection with [1].
> > > > > > >
> > > > > > > For a given fixed game and equilibrium of that game, you can take the asymptotic expansion in [1] and use the limit definition (which means asymptotic since a limit is being passed towards $\tau\rightarrow\infty$) of the asymptotic expansion (as [1] does to translate between lemma 41 to eqn 7) to get an, albeit arbitrarily large, finite $\tau^{\ast}$ which is exactly what Reviewer 4 is saying. This it is not a theorem statement in [1] by the way since they are instead focusing on the entire class of zero-sum games and analyzing the stability of $\tau$-GDA as a function of a fixed $\tau$. Furthermore, the precise statement for a finite $\tau$ which is a corollary of an asymptotic expansion is actually in chapter 2 (we said chapter 3 in a couple of past responses, but it is actually chapter 2) of the textbook [2, Chapter 2, Corollary 3.1 pg 58]. We essentially gave proof in Appendix I translating from the more general setting of singularly perturbed linear systems to the stability of $\tau$-GDA in zero-sum games to bring this interesting body of literature to light. This is to say that we are fully aware that such a result exists, and it is in no way the result that we are giving in Theorem 1.

---

> > > > > > > > ### Comment · AnonReviewer3 · 2020-11-13
> > > > > > > > **My concern confirmed**
> > > > > > > >
> > > > > > > > Dear authors,
> > > > > > > >
> > > > > > > > I understand all of your claims and contributions (the added benefits of your tools over [1], examples demonstrating tight bounds, etc.). My evaluation is based on the following two concerns which I will make precise and falsifiable. As a result, I would appreciate an answer in the form of "Yes" or "No, because...", instead of stressing time and again your various other contributions or you knew this result etc. I acknowledge those, and they are insignificant if my concerns were true.
> > > > > > > >
> > > > > > > > ---
> > > > > > > > **Theoretical concern**: the most important theorem of this paper, that there exists a finite timescale separation for GDA (the major direction of Theorem 1), is known.
> > > > > > > >
> > > > > > > > My contention is that if you follow the proof of [1] **verbatim** and simply keep track of the $\delta, \epsilon$ in Lemma 41 and constants, then you get an (existential) finite timescale separation.
> > > > > > > >
> > > > > > > > This is a mathematical statement whose response can only be "Yes" or "No, the error occurs at..." Comments such as "This is not the the interpretation of [1]" is irrelevant. Since you have agreed to modify the statement of your Theorem 1, I  take it as you agree the answer is "Yes".
> > > > > > > >
> > > > > > > > ---
> > > > > > > > **Practical concern**: there is no additional benefit of the new tools in this paper for virtually all practical problems (such as GANs).
> > > > > > > >
> > > > > > > > My contention is that the "non-asymptotic" $\tau^*$ in this paper requires to know $x^*$, the very solution you are looking for. In practice, we never know $x^*$ (otherwise there is nothing to optimize over), and hence the "non-asymptotic" $\tau^*$ in this paper is just as good as [1].
> > > > > > > >
> > > > > > > > This is, again, a mathematically refutable sentence. I am fully aware of the benefits of your tools on toy examples (such as Dirac-GANs), but I fail to see what is being implied by your theorem other than:
> > > > > > > >
> > > > > > > > "there exists a finite $\tau^*$ such that $\tau^*$-GDA is locally stable at $x^*$"
> > > > > > > >
> > > > > > > > in your ResNet experiment on CIFAR-10 and CelebA. You can easily refute my contention by telling me how your tools provide insight into choosing the timescale, instead of heuristically doubling it as you did in the submission, before knowing $x^*$.
> > > > > > > >
> > > > > > > > Note that "we provide the first finite timescale separation where as [1] requires $\tau^* \rightarrow \infty$" is no longer valid as discussed in my theoretical concern.
> > > > > > > >
> > > > > > > >
> > > > > > > >
> > > > > > > > [1] Jin et al. arxiv 1902.00618. What is Local Optimality in Nonconvex-Nonconcave Minimax Optimization. (published in ICML 2020).

---

> > > > > > > > > ### Author Response · Authors · 2020-11-14
> > > > > > > > > **On Practical Concern of Reviewer 3**
> > > > > > > > >
> > > > > > > > > Practical Concern:
> > > > > > > > >
> > > > > > > > > You say:
> > > > > > > > >
> > > > > > > > > “There is no additional benefit of the new tools in this paper for virtually all practical problems (such as GANs).”
> > > > > > > > >
> > > > > > > > > We will show shortly this is false.
> > > > > > > > >
> > > > > > > > > Then you claim the following is the mathematical statement that is refutable.
> > > > > > > > >
> > > > > > > > > “My contention is that the non-asymptotic $\tau^{\ast}$ in this paper requires to know $x^{\ast}$, the very solution you are looking for. In practice, we never know $x^{\ast}$.., and hence the “non-asymptotic $\tau^{\ast}$ in this paper is just as good as [1]….You can easily refute my contention by telling me how your tools provide insights into choosing the timescale…”
> > > > > > > > >
> > > > > > > > > The answer again is no, and we can certainly mathematically refute this and provide a concrete example.
> > > > > > > > >
> > > > > > > > > We do not need to know $x^{\ast}$ to construct $\tau^{\ast}$ and we can indeed get insights about GANs from our construction. In fact, this was the purpose of Theorem 3 and that entire section. Similarly that is why we showed the Dirac-GAN with regularization and why we used the same framework for our generative adversarial network experiments.
> > > > > > > > >
> > > > > > > > > First of all, you say that there is no additional benefit of the new tools for all practically relevant problems. We previously tried to explain why this was not the case by showing how for the Dirac-GAN, we obtain $\tau^{\ast}=0$ whereas an asymptotic construction would result in $\tau$ bound that approaches infinity. We see that this was not enough to convince you. We find it relevant based on your comment to note that in the well-cite paper of [3], when introducing the Dirac-GAN as an important case study for understanding local convergence in generative adversarial networks, the authors of [3] referenced the following quote from Ali Rahimi’s Test of Time Award speech in NIPS 2017, which says that
> > > > > > > > >
> > > > > > > > > “Simple experiments, simple theorems are the building blocks that help us understand more complicated systems.”
> > > > > > > > >
> > > > > > > > > We wholeheartedly agree with this sentiment, and are surprised the reviewer does not feel the same way.
> > > > > > > > >
> > > > > > > > > So, to extrapolate from the Dirac-GAN example to satisfy your concern, in the revised version of the paper we have shown in Appendix L an alternative proof of Theorem 3 using our construction for $\tau^{\ast}$ in which we recover $\tau^{\ast}=0$ for the entire class of games without knowing $x^{\ast}$. We remark that this Theorem is under Assumption 1, which was previously considered by [3] and is arguably the most relaxed assumptions under which theoretical guarantees are known for local convergence in generative adversarial networks. It is not hard to generalize from the argument we used regarding the Dirac-GAN in a previous response, that an asymptotic construction would result in a $\tau$ bound that approaches infinity.
> > > > > > > > >
> > > > > > > > > This is again to say, that the insights that can be made from simple examples can translate to larger scale problems that you may be concerned about in practice. We did in fact choose our parameters for our generative adversarial networks based on these insights and as mentioned in the paper, we found the conclusions about the performance as a function of the timescale separation and the regularization were consistent with what we learned from the simple example.
> > > > > > > > >
> > > > > > > > > This example we have provided exactly illustrates the practical (and theoretical) use of the construction in Theorem 1 by applying it to an important class of games to infer exactly what the value of $\tau^{\ast}$ is for that entire class without knowing $x^\ast$ simply using the properties of the Kronecker sum and product, a litany of which can be found in the reference [4] which we have already cited in the original manuscript.
> > > > > > > > >
> > > > > > > > > We believe we have fully and concretely answered both your theoretical and practical concerns. Please let us know if any questions remain.
> > > > > > > > >
> > > > > > > > > [1] Jin et al. arxiv 1902.00618. What is Local Optimality in Nonconvex-Nonconcave Minimax Optimization. (published in ICML 2020).
> > > > > > > > > [2] Kokotovic et al (see, Chapter 3). 1986. Singular Perturbation Methods in Control Analysis and Design.
> > > > > > > > > [3] Mescheder, ICML 2018. Which Training Methods for GANs do actually Converge?
> > > > > > > > > [4] Magnus. 1988. Linear Structures.

---

> > > > > > > > > > ### Comment · AnonReviewer3 · 2020-11-14
> > > > > > > > > > **Followup; practical**
> > > > > > > > > >
> > > > > > > > > > I wholeheartedly agree with the statement:
> > > > > > > > > >
> > > > > > > > > > “Simple experiments, simple theorems are the building blocks that help us understand more complicated systems.”
> > > > > > > > > >
> > > > > > > > > > I also confirm that I understand your tools give better insight into Dirac-GANs. Now, please tell me what your tools can reveal while [1] cannot on your ResNet experiments, so as to help the readers "understand more complicated systems".

---

> > > > > > > > > ### Author Response · Authors · 2020-11-14
> > > > > > > > > **On Theoretical Concern of Reviewer 3**
> > > > > > > > >
> > > > > > > > > Thanks for the continued discussion. We hope we can clear up your concerns. Please also see the revised version of the paper that we have posted.
> > > > > > > > >
> > > > > > > > > Theoretical concern: The answer is no. The error in what you are saying happens exactly when you say:
> > > > > > > > >
> > > > > > > > > “The most important theorem in this paper, that there exists a finite timescale separation for GDA (the major direction of Theorem 1), is known.”
> > > > > > > > >
> > > > > > > > > The error you are making is that what you just stated is not the contribution of Theorem 1. The contribution of Theorem 1 is the precise construction of $\tau^{\ast}$. The tight construction of $\tau^{\ast}$ is a complimentary result to that from [1] and Corollary 3.1 of Chapter 2 from [2], which is why we have had trouble understanding the conflation that is happening. We have tried to make this clear repeatedly, but we see that it is not coming across yet. Can you please confirm that you understand that the contribution of Theorem 1 is about exactly constructing $\tau^{\ast}$ where the system becomes stable for all $\tau^{\ast}>\tau$ and not just about the existence of a finite, but potentially arbitrarily large timescale parameter that guarantees stability?
> > > > > > > > >
> > > > > > > > > You say
> > > > > > > > >
> > > > > > > > > “Since you have agreed to modify the statement of Theorem 1, I take it as you agree the answer is ‘yes’.”
> > > > > > > > >
> > > > > > > > > We are sorry that you misinterpreted what we were modifying. As we showed when we wrote it out for you, what we are modifying is to including the exact construction of $\tau^{\ast}$ in the theorem statement as opposed to leaving it in the appendix. Again, the modification we have made to Theorem 1 is not changing anything about it or its original intended purpose. It is simply to avoid confusion about the contribution if the reader did not examine the proof sketch immediately after. Thank you for improving our paper by making it clear that it is necessary to state the exact construction in the theorem to make this explicit.
> > > > > > > > >
> > > > > > > > > We hope it is now clear that Theorem 1 and its goal are complementary, yet entirely different than that of Theorem 28 of [1] or even Corollary 3.1 of Chapter 2 from [2].
> > > > > > > > >
> > > > > > > > > If you can acknowledge this answer, then that takes us to your final concern about the practical importance.

---

> > > > > > > > > > ### Comment · AnonReviewer3 · 2020-11-14
> > > > > > > > > > **Followup; theoretical**
> > > > > > > > > >
> > > > > > > > > > To answer the question:
> > > > > > > > > >
> > > > > > > > > > "Can you please confirm that you understand that the contribution of Theorem 1 is about exactly constructing $\tau^*$ where the system becomes stable for all $\tau > \tau^*$ (I assume there is a typo here) and not just about the existence of a finite, but potentially arbitrarily large timescale parameter that guarantees stability?"
> > > > > > > > > >
> > > > > > > > > > I confirm, and I further confirm that [1] established the existence of such $\tau^*$: page 25, 5th line,
> > > > > > > > > >
> > > > > > > > > > "*By Eq. (7), we know there exists sufficiently small $\epsilon_0$ such that for any $\epsilon < \epsilon_0$, the real part Re(λi) < 0; i.e., (x, y) is a strict linear stable point of $1/\epsilon$−GDA.*"
> > > > > > > > > >
> > > > > > > > > > As a result, I confirm that finite timescale separation is not novel.

---

> > > > > > > > > ### Author Response · Authors · 2020-11-14
> > > > > > > > > **Response to Reviewer 3 (previously said R4 but meant R3) Follow Up**
> > > > > > > > >
> > > > > > > > > You say:
> > > > > > > > >
> > > > > > > > > “I confirm and I further confirm that [1] established the existence of $\tau^{\ast}$: page 25, 5th line.”
> > > > > > > > >
> > > > > > > > > That is fine, we agree that [1] has such an existence result, and similarly that [2] much before (1961 to be precise) have an analogous result in the setting of singularly perturbed linear systems of which $\tau$-GDA is a subclass (in a trivial way) of (meaning result of [2] is more general).
> > > > > > > > >
> > > > > > > > > However, this doesn’t matter at all concerning the results of our paper. You just said you confirm you understand that the contribution of Theorem 1 is about exactly constructing $\tau^{\ast}$ such that it is the minimal value of $\tau$ such that for all $\tau >\tau^{\ast}$ the system is stable (there is no typo in what we wrote). Then you immediately shift away and talk about existence, even though we have said many times our result is not focused existence, it is about constructing the minimal $\tau^{\ast}$. As we explained to you, if you used the method from [1] or [2] even on the simple Dirac-GAN, the bound on $\tau^{\ast}$ from the asymptotic approach of [1] or [2] results in a $\tau^{\ast}$ approaching infinity whereas our approach (which is completely different mathematically and conceptually to get around this) would get exactly $\tau^{\ast}=0$. Then again we reiterate this is not some kind of corner case and we are just using it as a concrete example to illustrate the difference to you.
> > > > > > > > >
> > > > > > > > > We honestly feel you are being unreasonable completely since we answered your question, explained the differences, and then when we asked you to confirm that you acknowledged that the contribution we are making with Theorem 1 is about exactly constructing the minimal $\tau^{\ast}$ and you shifted the conversation back to existence which we explicitly said is not the contribution we are making with Theorem 1 because this is previously known even before [1].
> > > > > > > > >
> > > > > > > > > So we would like to ask you again, do you understand that the contribution of Theorem 1 we are making is about constructing exactly the minimal $\tau^{\ast}$ and not about existence?
> > > > > > > > >
> > > > > > > > > We would like to not talk about the existence result anymore because we have said so many times that is not the contribution we are making with Theorem 1, and to be honest is distracting away from what we are contributing when it is peripheral or perhaps complimentary to what we are doing and we have said so many times that the existence result is known and not what we are doing.
> > > > > > > > >
> > > > > > > > > We want to recenter the discussion on the contribution we are making which is about constructing exactly the minimal $\tau^{\ast}$ which is completely novel. Whether or not you think that this novel result is practically useful is a different discussion, but there is no reasonable you can deny that the construction of the minimal $\tau^{\ast}$ is not novel. If this is still not clear to you, we would ask that you look at the proof of Theorem 1.
> > > > > > > > >
> > > > > > > > > So now, back to the practical usefulness. Again regarding the practical question, much like the theoretical question you had, we answered your ‘yes’ or ‘no’ questions precisely with evidence to back them up and you completely ignored them.
> > > > > > > > >
> > > > > > > > > So for your followup on the practical concern, we ask that you please actually acknowledge where we explained the answer to this question. It might help also if you looked at the Appendix L in the paper as we suggested. Specifically, as we just explained our insight would be that under the proper assumptions for the GANs as considered in [3], we find that $\tau^{\ast}=0$ whereas the asymptotic analysis method of [1, 2] would result in a bound of $\tau^{\ast}=\infty$. This provides explanation for satisfying performance for small, reasonable values of $\tau$ on ResNet, whereas without our insights you would expect that $\tau$ must be approaching infinity and the algorithm would have numerical issues since the learning rate would have to be arbitrarily small or the update would diverge.
> > > > > > > > >
> > > > > > > > > It really seems like you ignored our whole response on the practical consideration, which was one of your specific questions, that we answered very precisely and so we just ask that you try to be reasonable and actually read and acknowledge that we are answering exactly your questions.
> > > > > > > > >
> > > > > > > > > [1] Jin et al. arxiv 1902.00618. What is Local Optimality in Nonconvex-Nonconcave Minimax Optimization. (published in ICML 2020). [2] Kokotovic et al (see, Chapter 3). 1986. Singular Perturbation Methods in Control Analysis and Design. [3] Mescheder, ICML 2018. Which Training Methods for GANs do actually Converge?

---

> > > > > > > > > > ### Author Response · Authors · 2020-11-18
> > > > > > > > > > **Response to Reviewer 3: Another Concrete Example**
> > > > > > > > > >
> > > > > > > > > >
> > > > > > > > > > Dear reviewer 3, we wanted to bring to light another concrete example of how we can use our tools to get a finite $\tau^{\ast}$ for entire classes of games. This we mentioned in a comment to reviewer 4 and thought it would be worth pointing out to you also with some more detail.
> > > > > > > > > >
> > > > > > > > > > Please also see our last response to you above, which we mistakenly titled as to reviewer 4 where we meant reviewer 3, and potentially that caused you to miss it. Just to remind you the order since it is hard to track, you posted 'my concern confirmed', we posted both 'on theoretical concern of reviewer 3' and 'on practical concern of reviewer 3', then you posted 'followup; theoretical' and 'followup; practical', then we posted 'response to reviewer 3 follow up' and now this post.
> > > > > > > > > >
> > > > > > > > > > Using the tools in our paper, we can show the following. For non-convex strongly-concave games where $-J_{1}(x^\ast)$ is stable and $\text{det}(\text{Schur}(-J_1(x^\ast)))\neq 0$, we can obtain a bound of $\tau^{\ast}\leq 1$.
> > > > > > > > > >
> > > > > > > > > > In other words, in the non-convex strongly concave setting, any stable point of 1-GDA is also stable for $\tau$-GDA for all $\tau\geq 1$. This has an important implication. Precisely, the upper bound of $\tau^{\ast}\leq 1$ given that $-J_{1}(x^\ast)$ is stable implies that any stable point of 1-GDA in this class of games is a strict local minmax by Theorem 1 of our paper.
> > > > > > > > > >
> > > > > > > > > > The significance of this is that there has been recent work on analyzing various first order methods in nonconvex strongly-concave games. However, to our knowledge, these works almost exclusively only consider convergence to stationary points and do not consider the classification of them as equilibrium. The result we can obtain using the tools in our paper implies that in this class of games, if GDA converges to a stable point, then that stable point is in fact a strict local minmax.
> > > > > > > > > >
> > > > > > > > > > We will now explain how we can get this using the methods in our paper.
> > > > > > > > > >
> > > > > > > > > > To get a sense of why this is the case, consider Theorem 1 in [1] which says that a family of matrices is stable for a range of parameters $U$  if and only if (1) it is stable for some nominal parameter $\tau_{\ell}$ and (2) the guard map is non-zero for all $\tau\in U$. In our case, $U=[1,\infty)$ and the nominal parameter is $\tau_{\ell}=1$. To argue (2), we show that the guard map is non-zero for all $\tau\in [1,\infty)$ simply by examining the structure of the guard map and again reduce it to an eigenvalue problem in similar manner to Theorem 5 in [1], and in the proofs of our Theorems 1 and 2. Then, our Theorem 1, can be applied (since $\tau$-GDA is stable for all $\tau\geq 1$) to conclude that $x^\ast$ is a strict local minmax.
> > > > > > > > > >
> > > > > > > > > > Moreover, if one can choose a nominal value of $\tau$ and show that $-J_\tau(x^\ast)$ is stable, then the same argument shows that this nominal $\tau$ is an upper bound for $\tau^\ast$ for this class of games.
> > > > > > > > > >
> > > > > > > > > > We did not include this result in the paper because we have included it in another of our papers.
> > > > > > > > > >
> > > > > > > > > > Thanks again for your time and responsiveness.
> > > > > > > > > >
> > > > > > > > > > [1] Saydy, New Stability/Performance Results for Singularly Perturbed Systems, 1996.

---

> > > > ### Author Response · Authors · 2020-11-12
> > > > **Another Response to Reviewer 4 (2/2)**
> > > >
> > > > We have in our last response and in the paper discussed a concrete example of the Dirac-GAN, which actually we found the insights to carry over to training generative adversarial networks as discussed in our previous responses and in the paper.
> > > >
> > > > The finite value construction in a non-asymptotic way is extremely useful both for setting up new directions for theoretical results as well as opening up avenues for practical development of heuristics by studying theoretically similar smaller scale problems expected to be in the same class as the larger problem one wants to solve in practice — something we do in our experiments.
> > > >
> > > > First, even in gradient descent, optimal rates for local convergence are most often given in terms of the stepsize choice which is in turn defined in terms of the spectral properties of the local linearization (or Hessian in gradient descent) around a stable critical point. Our result let's us do exactly this for zero-sum games and $\tau$-GDA. This is something the result of [1] does not allow for by any means. Specifically, we have an expression for $\tau^\ast$ in terms of the spectral properties of the block components of $J_\tau$.  This allows for rates to be given in terms of this parameter $\tau^\ast$ and the choice of stepsize $\gamma_1$ such that both depend on the sub-blocks of $J_\tau$. In a sense, our results add an extra parameter which is explicitly defined in terms of the spectral properties of the Jacobian. Again, this cannot be done with the results of [1], nor any other existing result. Further work is required to have a more “interpretable” or “intuitive” understanding of $\tau^\ast$, but nonetheless our work is progress beyond [1].
> > > >
> > > > Practical Development of Theoretical Grounded Heuristics: While this reviewer might not find the expression for $\tau^\ast$ intuitive, it is entirely computable. And hence, it can be used to explore classes of zero sum games with specific structure (e.g., like we do with the GANs under Assumption 1) in order to determine exact values of $\tau^\ast$ on smaller scale problems (Dirac-GAN and Covariance Estimation with WGAN) which give insights about reasonable values of $\tau^\ast$ for much larger scale problems (CIFAR-10). These things we explore in our experiments. To expand on the hyper-parameter exploration problem, consider, for example, the root locus plot (eigenvalue plot) around any critical point will always the same general structure you see in our eigenvalue plots. This is because we are looking at eigenvalues of local linearizations with a very specific structure [A, B; -B^T, -D]. Hence, by exploring the value of tau on classes of zero sum games (e..g, classes that often show up in GAN such as the Dirac GAN) we can understand better ways of choosing heuristics (other than blindly choosing the roll outs or timescale separation). The Dirac GAN is also a perfect example of how exploring $\tau$ on smaller examples gives insights about more complex problems.  It is what we do in going from our smaller scale experiments (including the covariance GAN and Dirac GAN) to the large scale GAN where we have no handle on the critical points a priori but none the less expect similar behavior due to the structure of the problem; indeed, this is how we determined which values of $\tau$ to explore, and we show in doing so we can improve upon (while maintaining the theoretical convergence guarantees) naive uses of GDA with a simple change that doesn’t require the introduction of higher order terms. If we followed a result of the nature provided in [1], it would suggest that $\tau$ needed to be arbitrarily large. We observe in experiments in fact that scaling $\tau$ in this way leads to divergence very rapidly (e.g., at values of $\tau=16$ even, which would force us to reduce the stepsize so that the timescale separation would work, but then with small learning rate there is no progress).
> > > >
> > > > Optimizing Rates for Stochastic and Online Settings and Obtaining First-Order Convergence Guarantees to Minmax: It also opens up new opportunities to understand convergence properties and optimal rates for first order methods even in stochastic and online settings, something that a non-constructive, asymptotic result cannot be used for! For instance, existing results in online learning via stochastic or zero-th order and other black box methods based on GDA for minmax problems can only guarantee convergence to first order stationary points. By using the results we have, it is possible to not only ensure that these approaches with finite timescale separation achieve strict local minmax and not just first order points, but also to optimize the rate in a similar way as is done in the literature by choosing the stepsize.
> > > >
> > > > [1] Jin et al. arxiv 1902.00618. What is Local Optimality in Nonconvex-Nonconcave Minimax Optimization. (published in ICML 2020).
> > > > [2] Kokotovic et al (see, Chapter 3). 1986. Singular Perturbation Methods in Control Analysis and Design.

---

> > > > ### Author Response · Authors · 2020-11-12
> > > > **Another Response to Reviewer 4 (1/2)**
> > > >
> > > > Thanks for the response and continued discussion. Prior to answering your questions about the value of an explicit construction, we want to reiterate that you are extrapolating from the results in [1] to draw further unstated conclusions and using them to limit the value you see in one of our contributions, which again go far beyond existence of a $\tau^{\ast}$.
> > > >
> > > > Theorem 28 of [1] and Theorem 1 in our paper are completely different in the goal and perspective.  The analysis of [1] is across all games where as our analysis is on a game by game basis. Across of all games, [1] is showing that if you want a fixed $\tau^{\ast}$ then $\tau^{\ast}\rightarrow \infty$. Now, you on your own are extrapolating from the analysis in the proof of Theorem 28, to conclude a game by game result. As we just discussed in our previous response, applying this type of asymptotic analysis to obtain a game by game result, can be arbitrary far from the optimal $\tau^{\ast}$ or even any reasonable $\tau$ that could result in an implementable algorithm. Furthermore, using asymptotic expansion to infer existence of a finite value a known result for singular perturbed systems, a class of which contains $\tau$-GDA,  and contained in references such as [2, Chapter 3] and therein. To reiterate, as we showed, in the simple Dirac-GAN example, applying the asymptotic analysis would result in finding $\tau^{\ast}\rightarrow \infty$ whereas our approach independent of the value of $\mu$ would always produce the $\tau^{\ast}=0$ as is correct.
> > > >
> > > > To make this fully concrete since it did not seemed to be acknowledged, and to demonstrate why this is important, we took the simple Dirac-GAN and increased the regularization to $\mu=75$. We begin with a fixed choice of $\gamma_1$ and then for that choice increase $\tau$ by factors of 2 until the dynamics diverged. We then plot the convergence of the final choice of $\tau$ that did not diverge for that learning rate, and then decrease the learning rate and repeat the process of increasing $\tau$. We show the results in this anonymous link with the Figures: https://anonymous.4open.science/repository/f3f459fc-0a2e-4b41-b5ad-29acf39cb043/DiracGAN-convergenceRates.png. As shown in the figure called DiracGAN-convergenceRates.png if we forced to choose a massive $\tau$ for stability purposes, then to get convergence we have to choose a vanishing learning rate $\gamma_1$ so then it is extremely slow to converge. On the other hand, if we have flexibility in the choice of $\tau$ because of a tight construction, then we can implement a learning rule that actually converges fast. We also show how in DiracGAN-divergence.png of the link how with the largest learning rate the dynamics with larger $\tau$ diverge in just a couple steps.
> > > >
> > > > Now we would like to address the importance of a non-asymptotic construction, and our other contributions which seemed to be overlooked. Also just to reiterate, we will be sure to include the constructions of $\tau^{\ast}$ and $\tau^{0}$ in the statement of Theorems 1 and 2 versus just having them in the appendix since these are important contributions that seem to be getting lost in the reviews.
> > > >
> > > > First, the statement that it is clear that $\tau^\ast$ is the one such that the eigenvalues of $-J_\tau(x^\ast)$ crosses into the open left hand plane we disagree with. Specifically, we disagree that it is clear or obvious how to construct such a $\tau$ or show that it exists. Moreover, this concept of determining what value of $\tau$ the eigenvalues cross in to the open left hand plane is not what [1] does at all.  This idea cannot be attributed to them. They compare to balls around the eigenvalues of $\text{Schur}(-J_\tau(x^\ast))$ and $\tau D_2^2f(x^\ast)$ which could be any where in the open left hand plane, unrelated to the imaginary axis. It is a contribution of our work to actually look at the problem from this perspective that we need to 1) find a certificate for when the eigenvalues cross the boundary and 2) ensure that remain there for all larger such $\tau$.
> > > >
> > > > Questioning the value of Non-Asymptotic Results?:
> > > > To reiterate, the result you are comparing Theorem 1 in ours to is based on an asymptotic expansion and our is not. So they are two different types of results. Is this reviewer suggesting that non-asymptotic work is not worthwhile? and has no value? We whole heartedly disagree with this sentiment. Even if [1] had explicitly included some finite $\tau$ result it would be an asymptotic one. Ours is non-asymptotic. We also disagree that removing the issue by avoiding having to compare to the eigenvalues of matrices which could lead to arbitrarily large constructions of $\tau$ and hence serious numerical challenges is not a contribution. We get rid of this problem completely by avoiding the asymptotic expansion all together. This is a significant contribution, and certainly does expand our understanding of this class of games which are of practical importance in ML.

---

### Author Response · Authors · 2020-11-12
**Message to Reviewers: We Posted Responses and Look Forward to Discussing**

Dear reviewers,

Thank you for your time and effort spent looking over our paper and providing helpful feedback.

We have finished posting responses to the initial reviewers. We will be working to update the paper to reflect what is included in our responses. You should be able to find the responses to your review directly under them. Reviewers 3 and 4 had a similar question, so in addition we have posted a common response to that question for them at the top.

In the meantime, we are looking forward to having a dialogue about the paper and any questions or concerns that remain. We hope to hear back from you as soon as possible!

---

### Author Response · Authors · 2020-11-14
**Revisions to Paper Uploaded: Request for Response Inside**

Dear Reviewers:
We have updated the draft based on the discussions here. Updates are in blue so that they are easier to find.

The main changes are adding to Theorem 1 and 2 the explicit construction of $\tau^\ast$ and $\tau_0$, respectively. In addition, we have added an Appendix L that contains an example of how to apply our construction from Theorem 1 to an entire class of games satisfying Assumption 1 in order to construct $\tau^\ast$ without knowing $x^\ast$.  The example illustrates how the construction can be used more generally by simply leveraging the structure of the class of games and the properties of the Kronecker sum and products.

Reviewer 4: We would like to call your attention specifically to the updates on the bottom of page 2 and on page 3. We have added quite a bit of new text comparing our work to that of Jin et al and other existing results that rely on an asymptotic expansion around the eigenvalues of the Schur complement and the second players Hessian to obtain a finite value of $\tau$. We have also added to the theorem statements of 1 and 2  the construction of the values of $\tau$. Furthermore, we have added text after Theorem 1 contextualizing the construction and highlighting the importance of such a construction for both practical and theoretical reasons. And to address practical considerations on how our construction informs our knowledge about parameters such as $\tau$ in games applied to ML we have added an Appendix L as noted above.

Can you please let us know your thoughts on these changes and other questions we can answer?

Reviewer 2: we have made the changes we stated we would in our response to you. Can you please let us know if this is sufficient?

Reviewer 1: We have added text on bilinear games in the related work section Appendix A. We would like to hear from you regarding our prior response and to know whether you still feel that we should change the title?

We look forward to hearing from you all. We do appreciate the feedback so far and we think that it has helped greatly improve the paper in the sense that the results are now placed more clearly in the context of the existing singular perturbation theory and the ML+games literature, as well as highlight how to use the results for choosing $\tau$ in practice.

---

> ### Comment · AnonReviewer4 · 2020-11-14
> **Updates have improved the paper**
>
> The discussion in page 2 and page 3 presents the context of existing results and their limitations in a more explicit and understandable way. I think this has improved the paper substantially.
>
> Additionally, Appendix I discusses some potential applications of the explicit construction without knowing $x^*$.  I think this argument is even more straightforward compared to the argument used in Appendix E.  My only concern is that equilbrium agnostic statements, due to the complexity of the formula of Q, will only be able to give results like $\tau^*=0$.  I would like to ask the authors if they are aware of tools that would provide upper bounds on the eigenvalues of Q based on the smoothness of $f$ and its first and second order derivatives (or other game properties).
>
> Based on the current version of the paper I have decided to increase my score to a 6.

---

> > ### Author Response · Authors · 2020-11-14
> > **Thanks you for your response and feedback**
> >
> > Thanks for your feedback and taking the time to look at the updated paper, it is really appreciated. Your comments have been really insightful and helpful, and helped us see how we need to position our results so that they come across in the way we intend them.
> >
> > You bring up a really good question. From our point of view, we think that these tools really open up the possibility to explore the kinds of questions that you have brought up. It may indeed be easiest to give equilibrium agnostic guarantees when $\tau^{\ast}=0$, which itself is useful and in some cases could potentially not be possible or be much harder without the tools we have presented. We do think that it will be important in future to explore further the types of structures where we can get such results for $\tau^{\ast}$ and because of our construction of $\tau^{\ast}$ we have method to try and do this which is useful. It really boils down to better understanding the operators that are present in the block matrix that characterizes the game, and really what we are trying to do in Appendix L is illustrate how it can be potentially used as you acknowledged.
> >
> > In response about tools that provide upper bounds on the eigenvalues of $Q$, we don’t have a super concrete response right now since we had not thought about that before. However, we think that if from the smoothness parameters you could bound the operators in the game matrix (even locally), you could potentially use that information to translate to a bound on $\tau^{\ast}$.
> >
> > We can give another concrete example that we are currently working on that is a direct result of the analysis methods in this paper. Using analogous tools but for a different purpose, we can show that if $-J(x^{\ast})$ (meaning $\tau=1$) is stable, and the game is nonconvex-strongly-concave, then it must be that $x^{\ast}$ is a strict local minmax and conversely that $\tau^{\ast}=1$. This is something we had tried a very very long time, but we could not prove it until discovering the tools we use in this work. For us, we think this is a very nice follow-up because it implies that if you do GDA without timescale separation in a nonconvex-strongly-concave game and converge to a stable point then it must be a strict local minmax, whereas to our knowledge all the work recently on nonconvex-strongly-concave games only consider convergence to a first order stationary point.

---

### Author Response · Authors · 2020-11-23
**Note to Reviewers: Thanks for Detailed Review Process, Also we changed title at the request of a reviewer**

Dear reviewers, we have posted another revision of the paper in which we believe we have addressed the vast majority of questions and comments you have raised. Note that we have changed the title, as this was requested by reviewer 1.

We would like to sincerely thank each of you for the amount of time and effort you have devoted to reviewing our paper. The feedback you have provided has been invaluable to improving our paper and we are very satisfied with the current version that incorporates your suggestions.

---

### Decision · Program_Chairs · 2021-01-07
**Final Decision**

**Decision:**

Accept (Poster)

**Comment:**

This paper treats the problem of running gradient descent-ascent (GDA) in min-max games with a different step-size for the two players. Earlier work by Jin et al. has shown that, when the ratio of the step-sizes is large enough, the stable fixed points of GDA coincide with the game's strict local min-max equilibria. The main contribution of this paper is an explicit characterization of a threshold value $\tau^*$ of this ratio as the maximum eigenvalue of a specific matrix that involves the second derivatives of the game's min-max objective at each (strict local) equilibrium.

This paper generated a fairly intense discussion, and the reviewers showed extraordinary diligence in assessing the authors' work. Specifically, the reviewers raised a fair number of concerns concerning the initial write-up of the paper, but these concerns were mostly addressed by the authors in their revision and replies. As a result, all reviewers are now in favor of acceptance.

After my own reading of both versions of the paper and the corresponding discussion, I concur with the reviewers' view and I am recommending acceptance subject to the following revisions for the final version of the paper:
1. Follow the explicit recommendations of AnonReviewer3 regarding the numerical simulations (or, failing that, remove them altogether). [The authors' phrase that "The theory we provide also does not strictly apply to using RMSprop" does not suffice in this regard]
2. Avoid vague statements like $\tau \to \infty$ in the introduction regarding the work of Jin et al. and state precisely their contributions in this context. In the current version of the paper, a version of this is done in page 4, but the introduction is painting a different picture, so this discussion should be transferred there.
3. A persisting concern is that the authors' characterization of $\tau^*$ cannot inform a practical choice of step-size scaling (because the value of $\tau^*$ derived by the authors depends on quantities that cannot be known to the optimizer). Neither the reviewers nor myself were particularly convinced by the authors' reply on this point. However, this can also be seen as an "equilibrium refinement" result, i.e., for a given value of $\tau$ only certain equilibria can be stable. I believe this can be of interest to the community, even though the authors' characterization cannot directly inform the choice of $\gamma_1$ and $\gamma_2$ (or their ratio).

Modulo the above remarks (which the authors should incorporate in their paper), I am recommending acceptance.